# Calibrating Generative Models to Distributional Constraints

**Henry D. Smith** [1]   **Nathaniel L. Diamant** [1]   **Brian L. Trippe** [1]

## Abstract

Generative models frequently suffer miscalibration, wherein statistics of the sampling distribution—such as the fraction of generations in a given class—deviate from desired values. We frame calibration as a constrained optimization problem and seek the closest model in Kullback-Leibler divergence satisfying a calibration constraint. To address the intractability of imposing these constraints exactly, we introduce two surrogate objectives for fine-tuning: (1) the relax loss, which replaces the constraint with a miscalibration penalty, and (2) the reward loss, which converts calibration into a reward fine-tuning problem. We demonstrate that these approaches substantially reduce calibration error across hundreds of simultaneous constraints and models with up to nine billion parameters, spanning applications in protein design, image generation, and language modeling. Code is available at https://github.com/smithhenryd/cgm.

## 1. Introduction

Generative models commonly produce samples whose statistics deviate systematically from desired values. Such *miscalibration* occurs across many domains. Image models, such as GANs and diffusion models, exhibit mode collapse, producing images that cover only a subset of the training distribution (Arora & Zhang, 2017; Qin et al., 2023). Language models represent gender, race, religion, and age in ways that reinforce societal biases (Gallegos et al., 2024). In synthetic biology applications, protein structure models produce samples that have alpha-helical and beta-strand substructures at frequencies atypical of proteins found in nature (Lu et al., 2025), and DNA models generate samples that contain subsequences at frequencies that differ from those in human DNA (Sarkar et al., 2024). These calibration errors

arise from many sources including dataset imbalances, suboptimal training dynamics, and post-hoc adjustments such as low-temperature sampling or preference fine-tuning.

We frame calibration as a constrained optimization problem: find the distribution closest in Kullback-Leibler (KL) divergence to the base model that satisfies a set of expectation constraints. We introduce two fine-tuning algorithms, **CGM-relax** and **CGM-reward** ("calibrating generative models"), that approximately solve the calibration problem by stochastic optimization. We demonstrate across three applications that CGM effectively calibrates high-dimensional generative models to meet hundreds of simultaneous constraints.

**Problem statement.** Consider a trained "base" generative model $p_{\theta_{\text{base}}}(\boldsymbol{x})$ with parameters $\theta_{\text{base}}$, a statistic $\boldsymbol{h}(\boldsymbol{x})$, and an expectation value desired for the statistic $\boldsymbol{h}^*$. We say $p_{\theta_{\text{base}}}$ is *calibrated* if $\mathbb{E}_{p_{\theta_{\text{base}}}}[\boldsymbol{h}(\boldsymbol{x})] = \boldsymbol{h}^*$ and *miscalibrated* if $\mathbb{E}_{p_{\theta_{\text{base}}}}[\boldsymbol{h}(\boldsymbol{x})] \neq \boldsymbol{h}^*$. In the case that $p_{\theta_{\text{base}}}$ is miscalibrated, our goal is to fine-tune its parameters $\theta_{\text{base}}$ to some $\theta$ such that $p_\theta$ is calibrated.

For example, if $\boldsymbol{h}(\boldsymbol{x}) = \mathbb{1}\{\boldsymbol{x} \in C\}$ is the 0-1 function indicating whether $\boldsymbol{x}$ belongs to class $C$, then $\mathbb{E}_{p_{\theta_{\text{base}}}}[\boldsymbol{h}(\boldsymbol{x})] = p_{\theta_{\text{base}}}(\boldsymbol{x} \in C)$ is the probability that $p_{\theta_{\text{base}}}$ generates a member of class $C$. When $\boldsymbol{h}^* > \mathbb{E}_{p_{\theta_{\text{base}}}}[\boldsymbol{h}(\boldsymbol{x})]$, calibration corresponds to increasing the probability of class $C$.

For a given $\boldsymbol{h}(\cdot)$ and $\boldsymbol{h}^*$, many calibrated models may exist. Provided a calibrated model exists, we seek the one that is closest to the base model in KL divergence,

$$p_{\theta^*} = \arg \min_{p_\theta} \mathrm{D_{KL}}\left(p_\theta \parallel p_{\theta_{\text{base}}}\right) \text{ s.t. } \mathbb{E}_{p_\theta}[\boldsymbol{h}(\boldsymbol{x})] = \boldsymbol{h}^*, \tag{1}$$

where $\mathrm{D_{KL}}\left(p' \parallel p\right) = \mathbb{E}_{p'}[\log p'(\boldsymbol{x})/p(\boldsymbol{x})]$ for $p'$ with a probability density with respect to $p$. Out of many possible notions of distance we choose $\mathrm{D_{KL}}$ because it is tractable for several classes of generative models.

**Related work.** Beyond the context of generative modeling, calibration is a major topic in supervised machine-learning (e.g., Lichtenstein et al., 1977; Dawid, 1982; Naeini et al., 2015; Guo et al., 2017; Vaicenavicius et al., 2019). In this literature, a model is calibrated if predicted probabilities match empirical frequencies on future data; this definition can be expressed as an expectation constraint. A variety of post-hoc methods enforce this property on the outputs

[1]Stanford University, Palo Alto, CA USA. Correspondence to: Henry Smith <smithhd@stanford.edu>, Nate Diamant <diamant@stanford.edu>, Brian Trippe <btrippe@stanford.edu>.

*Proceedings of the 43rd International Conference on Machine Learning*, Seoul, South Korea. PMLR 306, 2026. Copyright 2026 by the author(s).

of a predictive model, including Platt scaling (Platt, 1999) and conformal prediction (Shafer & Vovk, 2008). These approaches do not apply our setting, where the goal is to alter the distribution of the generative model.

Within the generative modeling community, there are a wealth of fine-tuning methods that incorporate preferences at the level of individual samples through a user-specified reward (Christiano et al., 2017; Rafailov et al., 2023; Uehara et al., 2024; Domingo-Enrich et al., 2025). These methods do not address calibration, for which the goal is to impose a constraint on the distribution $p_\theta$ rather than its samples $\boldsymbol{x}$.

Two prior works (Khalifa et al., 2021; Shen et al., 2024) and one concurrent work (Khalafi et al., 2026) propose fine-tuning procedures for distribution-level constraints. Khalifa et al. (2021) fine-tunes autoregressive language models to match distributional constraints with an algorithm similar to CGM-reward. Shen et al. (2024) propose a method for balancing class proportions in diffusion models that relies upon optimal transport. Khalafi et al. (2026) fine-tunes diffusion models with a primal-dual algorithm that is closely related to CGM-relax. However, all methods apply only to a single model class and are validated only on low-dimensional constraints. We show that CGM applies broadly to generative models with tractable likelihoods and reduces a majority of miscalibration for high-dimensional constraints.

Appendix A provides extended discussion of related work.

## 2. Calibrating Generative Models with CGM-relax and CGM-reward

The calibration problem is challenging in general because both the objective and calibration constraint in Equation (1) are defined by intractable expectations. To address this problem, we propose two alternative objectives whose *unconstrained* optima approximate the solution to (1). These objectives still involve expectations under $p_\theta$, but we show how to compute unbiased estimates of their gradients, which permits their minimization by stochastic optimization.

We call our algorithms optimizing the two surrogate loss functions CGM-relax and CGM-reward (Algorithms 1 and 2, respectively). These algorithms require only that one can draw samples $\boldsymbol{x} \sim p_\theta$ and compute $p_\theta(\boldsymbol{x})$ and $\nabla_\theta \log p_\theta(\boldsymbol{x})$; importantly, the constraint $\boldsymbol{h}$ need not be differentiable.

### 2.1. The Relax Loss

The relax loss avoids the intractability of imposing the calibration constraint exactly by replacing it with a constraint violation penalty

$$\mathcal{L}^{\text{relax}}(\theta) := \underbrace{\| \mathbb{E}_{p_\theta}[\boldsymbol{h}(\boldsymbol{x})] - \boldsymbol{h}^* \|^2}_{\mathcal{L}^{\text{viol}}} + \lambda \underbrace{\mathrm{D}_{\text{KL}}(p_\theta \| p_{\theta_{\text{base}}})}_{\mathcal{L}^{\text{KL}}}, \quad (2)$$

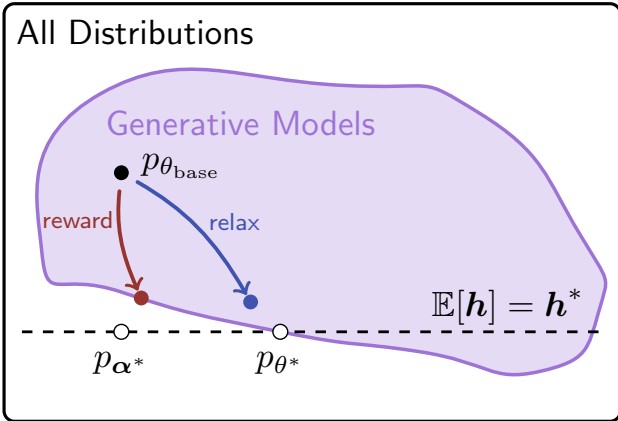

*Figure 1.* CGM aims to identify the generative model $p_{\theta^*}$, among those that satisfy the moment constraint, that is closest to $p_{\theta_{\text{base}}}$. $p_{\theta^*}$ is related to the maximum entropy distribution $p_{\boldsymbol{\alpha}^*}$.

where $\lambda > 0$ is a hyperparameter. $\mathcal{L}^{\text{viol}}$ is the constraint violation of the fine-tuned model, and $\mathcal{L}^{\text{KL}}$ is the KL divergence of the fine-tuned model to the base model. In the limit as $\lambda \to 0$, $\mathcal{L}^{\text{viol}}$ is the dominant term in the relax loss, and we expect the minimizer of (2) to approach the solution of the calibration problem (1). In practice, finite $\lambda$ must be chosen, and in our experiments we find $\lambda$ trades off between satisfying the calibration constraint and remaining close to $p_{\theta_{\text{base}}}$. Section 3 provides a heuristic for the choice of $\lambda$.

We choose the constraint penalty $\mathcal{L}^{\text{viol}}$ to be a squared $\ell_2$ norm because it is amenable to unbiased estimation. Suppose we have $M$ independent samples $\{\boldsymbol{x}_m\}_{m=1}^M$ from $p_\theta$. An unbiased estimate of $\mathcal{L}^{\text{viol}}$ is then

$$\widehat{\mathcal{L}}^{\text{viol}} := \left\| \frac{1}{M} \sum_{m=1}^M \boldsymbol{h}(\boldsymbol{x}_m) - \boldsymbol{h}^* \right\|^2$$
$$- \frac{1}{M(M-1)} \sum_{m=1}^M \left\| \boldsymbol{h}(\boldsymbol{x}_m) - \frac{1}{M} \sum_{m'=1}^M \boldsymbol{h}(\boldsymbol{x}_{m'}) \right\|^2, \quad (3)$$

where the second term is a bias correction. An unbiased estimate $\widehat{\mathcal{L}}^{\text{KL}}$ to the KL divergence can likewise be obtained as a Monte Carlo average from the same $M$ samples.

Combining these estimators yields our overall estimator for the relax objective, $\widehat{\mathcal{L}}^{\text{relax}} = \widehat{\mathcal{L}}^{\text{viol}} + \lambda \widehat{\mathcal{L}}^{\text{KL}}$. Appendix B.1 shows $\widehat{\mathcal{L}}^{\text{relax}}$ is unbiased for $\mathcal{L}^{\text{relax}}$.

### 2.2. The Reward Loss

The reward loss avoids the intractability of imposing the calibration constraint exactly through a connection between the calibration problem and the *maximum entropy problem* (Jaynes, 1957; Kullback, 1959; Csiszár, 1975). We first introduce the maximum entropy problem and show how to approximate its solution with samples from $p_{\theta_{\text{base}}}$. We then propose the reward loss as a divergence to this approximate solution and describe connections to reward fine-tuning.

**Algorithm 1** CGM-relax fine-tuning

**Input**: $p_{\theta_{\text{base}}}, \boldsymbol{h}(\cdot), \boldsymbol{h}^*, M$, and $\lambda$

▷ Initialize and optimize
$p_\theta \leftarrow p_{\theta_{\text{base}}}$
**while** not converged **do**
  ▷ Sample and compute weights
  $\boldsymbol{x}_1, \ldots, \boldsymbol{x}_M \overset{i.i.d.}{\sim} p_{\text{stop-grad}(\theta)}$
  $w_m \leftarrow p_\theta(\boldsymbol{x}_m)/p_{\text{stop-grad}(\theta)}(\boldsymbol{x}_m)$

  ▷ KL loss with LOO baseline
  $l_m \leftarrow \log p_{\text{stop-grad}(\theta)}(\boldsymbol{x}_m)/p_{\theta_{\text{base}}}(\boldsymbol{x}_m)$
  $l_m^{\text{LOO}} \leftarrow l_m - \frac{1}{M-1}\sum_{m'\neq m} l_{m'}$
  $\widehat{\mathcal{L}}^{\text{KL}} \leftarrow \frac{1}{M}\sum w_m l_m^{\text{LOO}}$

  ▷ Constraint violation loss
  $\boldsymbol{h}_m \leftarrow w_m(\boldsymbol{h}(\boldsymbol{x}_m) - \boldsymbol{h}^*)$
  $\widehat{\mathcal{L}}^{\text{viol}} \leftarrow \|\frac{1}{M}\sum \boldsymbol{h}_m\|^2 - \frac{1}{M}\widehat{\text{Var}}[\boldsymbol{h}_{1:M}],$
    $\widehat{\text{Var}}[\boldsymbol{h}_{1:M}] = \frac{1}{M-1}\sum\|\boldsymbol{h}_m - \frac{1}{M}\sum \boldsymbol{h}_{m'}\|^2$

  ▷ Total loss and update
  $\widehat{\mathcal{L}}^{\text{relax}} = \lambda\widehat{\mathcal{L}}^{\text{KL}} + \widehat{\mathcal{L}}^{\text{viol}}$
  $\theta \leftarrow \text{gradient-step}(\theta, \nabla_\theta \widehat{\mathcal{L}}^{\text{relax}})$
**end while**

---

**Algorithm 2** CGM-reward fine-tuning

**Input**: $p_{\theta_{\text{base}}}, \boldsymbol{h}(\cdot), \boldsymbol{h}^*, M, N$
▷ Estimate $\boldsymbol{\alpha}^*$ for reward
$\boldsymbol{x}_1, \ldots, \boldsymbol{x}_N \overset{i.i.d.}{\sim} p_{\theta_{\text{base}}}$
$\widehat{\boldsymbol{\alpha}}_N \leftarrow \arg\max \boldsymbol{\alpha}^\top \boldsymbol{h}^* - \log \sum \exp\{r_{\boldsymbol{\alpha}}(\boldsymbol{x}_n)\}$

▷ Initialize and optimize
$p_\theta \leftarrow p_{\theta_{\text{base}}}$
**while** not converged **do**
  ▷ Sample and compute weights
  $\boldsymbol{x}_1, \ldots, \boldsymbol{x}_M \overset{i.i.d.}{\sim} p_{\text{stop-grad}(\theta)}$
  $w_m \leftarrow p_\theta(\boldsymbol{x}_m)/p_{\text{stop-grad}(\theta)}(\boldsymbol{x}_m)$

  ▷ KL loss with LOO baseline
  $l_m \leftarrow \log p_{\text{stop-grad}(\theta)}(\boldsymbol{x}_m)/p_{\theta_{\text{base}}}(\boldsymbol{x}_m)$
  $l_m^{\text{LOO}} \leftarrow l_m - \frac{1}{M-1}\sum_{m'\neq m} l_{m'}$
  $\widehat{\mathcal{L}}^{\text{KL}} \leftarrow \frac{1}{M}\sum w_m l_m^{\text{LOO}}$

  ▷ Negative reward with LOO baseline
  $r_m^{\text{LOO}} \leftarrow r_{\widehat{\boldsymbol{\alpha}}}(\boldsymbol{x}_m) - \frac{1}{M-1}\sum_{m'\neq m} r_{\widehat{\boldsymbol{\alpha}}}(\boldsymbol{x}_{m'})$
  $\widehat{\mathcal{L}}^{\text{r}} \leftarrow -\frac{1}{M}\sum w_m r_m^{\text{LOO}}$

  ▷ Total loss and update
  $\widehat{\mathcal{L}}^{\text{reward}} = \widehat{\mathcal{L}}^{\text{KL}} + \widehat{\mathcal{L}}^{\text{r}}$
  $\theta \leftarrow \text{gradient-step}(\theta, \nabla_\theta \widehat{\mathcal{L}}^{\text{reward}})$
**end while**

---

**Maximum entropy problem.** The maximum entropy problem solves

$$\arg\min_{p\in\mathcal{P}(p_{\theta_{\text{base}}})} \text{D}_{\text{KL}}\left(p \parallel p_{\theta_{\text{base}}}\right) \text{ s.t. } \mathbb{E}_p[\boldsymbol{h}(\boldsymbol{x})] = \boldsymbol{h}^*, \quad (4)$$

where $\mathcal{P}(p)$ is the collection of probability distributions that have a density with respect to $p$. The calibration problem and the maximum entropy problem differ only in their domains: the domain of the calibration problem is generative models $p_\theta$ in the same parametric class as $p_{\theta_{\text{base}}}$, rather than the nonparametric set $\mathcal{P}(p_{\theta_{\text{base}}})$. Despite this difference, we obtain an alternative objective by studying the solution to (4). Theorem 2.1 characterizes this solution.

> **Theorem 2.1.** *Suppose $\boldsymbol{h}^*$ lies in the relative interior of the set of moments of $\boldsymbol{h}$ attainable by distributions in $\mathcal{P}(p_{\theta_{\text{base}}})$. Then there exists a vector $\boldsymbol{\alpha}^*$ for which*
>
> $$p_{\boldsymbol{\alpha}^*}(\boldsymbol{x}) \propto p_{\theta_{\text{base}}}(\boldsymbol{x})\exp\{r_{\boldsymbol{\alpha}^*}(\boldsymbol{x})\} \qquad (5)$$
>
> *with $r_{\boldsymbol{\alpha}}(\boldsymbol{x}) = \boldsymbol{\alpha}^\top \boldsymbol{h}(\boldsymbol{x})$ satisfies $\mathbb{E}_{p_{\boldsymbol{\alpha}^*}}[\boldsymbol{h}(\boldsymbol{x})] = \boldsymbol{h}^*$. $p_{\boldsymbol{\alpha}^*}$ is the solution to the maximum entropy problem.*

Appendix C provides additional background on the maximum entropy problem and a proof of Theorem 2.1.

The domain of the calibration problem may not contain $p_{\boldsymbol{\alpha}^*}$. However, if the class of generative models is expressive, its optimum $p_{\theta^*}$ will be close to $p_{\boldsymbol{\alpha}^*}$. This observation suggests a second way to remove the constraint in Equation (1): fine-tune $p_\theta$ to minimize a divergence to $p_{\boldsymbol{\alpha}^*}$.

**Estimating $p_{\boldsymbol{\alpha}^*}$.** Although Theorem 2.1 establishes the existence of some $\boldsymbol{\alpha}^*$ for which $p_{\boldsymbol{\alpha}^*}$ solves the maximum entropy problem, it does not tell us how to compute it. To accomplish this, we leverage Wainwright & Jordan (2008, Theorem 3.4), which states that when the assumption of Theorem 2.1 holds and there are no redundancies among the constraints $\boldsymbol{h}$,

$$\boldsymbol{\alpha}^* = \arg\max_{\boldsymbol{\alpha}} \boldsymbol{\alpha}^\top \boldsymbol{h}^* - \log\left(\mathbb{E}_{p_{\theta_{\text{base}}}}[\exp\{r_{\boldsymbol{\alpha}}(\boldsymbol{x})\}]\right). \quad (6)$$

In other words, by solving (6) one obtains the parameters $\boldsymbol{\alpha}^*$ of $r_{\boldsymbol{\alpha}}(\boldsymbol{x})$, which then determine the solution $p_{\boldsymbol{\alpha}^*}$ to the maximum entropy problem up to a normalizing constant.

A difficulty of solving (6) is that the expectation appearing in the second term is intractable for most generative models. We propose drawing $N$ independent samples $\{\boldsymbol{x}_n\}_{n=1}^N$ from $p_{\theta_{\text{base}}}$ and replacing the expectation with respect to $p_{\theta_{\text{base}}}$ by the expectation with respect to the empirical distribution that places probability $N^{-1}$ on each of the samples $\boldsymbol{x}_n$,

$$\widehat{\boldsymbol{\alpha}}_N = \arg\max_{\boldsymbol{\alpha}} \boldsymbol{\alpha}^\top \boldsymbol{h}^* - \log\left(\frac{1}{N}\sum_{n=1}^N \exp\{r_{\boldsymbol{\alpha}}(\boldsymbol{x}_n)\}\right). \quad (7)$$

Problem (7) is concave, and when $\widehat{\boldsymbol{\alpha}}_N$ is well-defined (see Appendix C.2), it can be found by convex solvers. Moreover, $\widehat{\boldsymbol{\alpha}}_N$ converges to $\boldsymbol{\alpha}^*$ in the limit of many samples $N$ at the parametric rate (Appendix C.4).

$\mathcal{L}^{\text{reward}}$ **and its estimation.** With $\widehat{\boldsymbol{\alpha}}_N$ in hand, we formulate our second loss as a divergence to $p_{\widehat{\boldsymbol{\alpha}}_N}$. For simplicity and

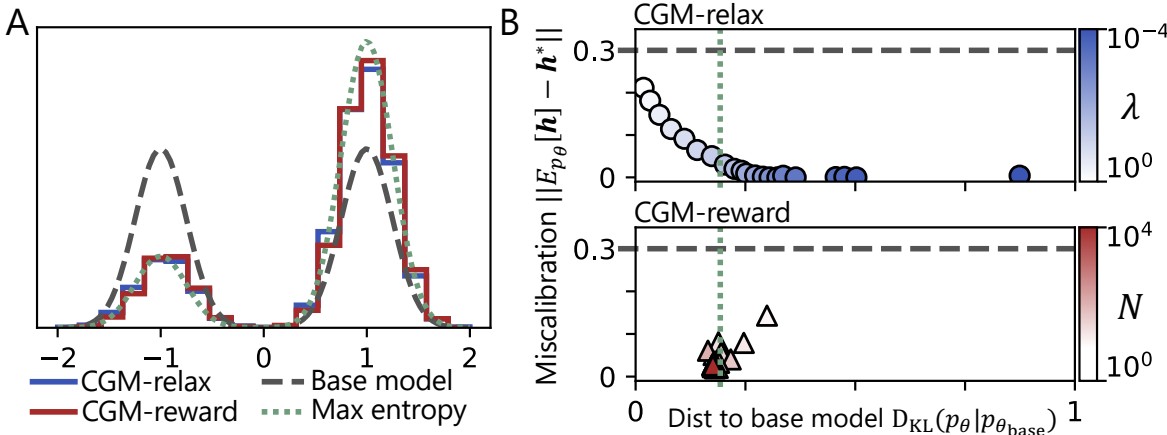

*Figure 2.* Calibrating mode proportions in a diffusion model targeting a 1D Gaussian mixture. **A**: Histograms of the CGM-relax and CGM-reward solutions, which closely approximate the maximum entropy solution. **B**: (top) The CGM-relax regularization parameter $\lambda$ trades off between constraint satisfaction and closeness to the base model (bottom) CGM-reward is accurate when enough samples $N$ are used to estimate $\boldsymbol{\alpha}^*$.

because it avoids the requirement to compute the normalizing constant of $p_{\widehat{\boldsymbol{\alpha}}_N}$, we again choose the KL divergence and define the reward loss as

$$\mathcal{L}^{\mathrm{reward}}(\theta) = \mathrm{D}_{\mathrm{KL}}\left(p_\theta \parallel p_{\widehat{\boldsymbol{\alpha}}_N}\right)$$
$$= \underbrace{\mathbb{E}_{p_\theta}[\log p_\theta(\boldsymbol{x})/p_{\theta_{\mathrm{base}}}(\boldsymbol{x})]}_{\mathcal{L}^{\mathrm{KL}}} + \underbrace{\mathbb{E}_{p_\theta}[-r_{\widehat{\boldsymbol{\alpha}}_N}(\boldsymbol{x})]}_{\mathcal{L}^{\mathrm{r}}} + C, \quad (8)$$

$\mathcal{L}^{\mathrm{KL}}$ is the KL divergence to the base model that appears in the relax loss (2), and $\mathcal{L}^{\mathrm{r}}$ is the expected negative reward under $p_\theta$. $C = \log \mathbb{E}_{p_{\theta_{\mathrm{base}}}}[\exp\{r_{\widehat{\boldsymbol{\alpha}}_N}(\boldsymbol{x})\}]$ is a log-normalizing constant that does not depend on $\theta$.

We call $r_{\boldsymbol{\alpha}}(\boldsymbol{x})$ the *reward* and $\mathcal{L}^{\mathrm{reward}}$ the reward loss because $\mathcal{L}^{\mathrm{reward}}$ coincides with the objective of reward fine-tuning algorithms. The goal of reward fine-tuning is to fine-tune the base generative model $p_{\theta_{\mathrm{base}}}$ to a tilted version of itself, where the tilt is determined by a reward $r(\boldsymbol{x})$.

Just as for $\mathcal{L}^{\mathrm{KL}}$, Monte Carlo sampling provides an unbiased estimate of $\mathcal{L}^{\mathrm{r}}$. This, in turn, gives us an unbiased estimate of the reward loss $\mathcal{L}^{\mathrm{reward}}$.

## 2.3. Gradient Estimation

We next describe our approach to computing unbiased estimates for the gradients of $\mathcal{L}^{\mathrm{relax}}(\theta)$ and $\mathcal{L}^{\mathrm{reward}}(\theta)$. This enables optimization of the relax and reward losses by stochastic optimization. We leverage the score function gradient estimator (Williams, 1992; Ranganath et al., 2014) for the reward loss and a similar importance sampling-based gradient estimator for the relax loss.

**Score function gradient estimation.** The primary challenge to computing gradients is the inability to directly exchange the order of the gradients and expectations taken with respect to $\theta$. That is, because $\nabla_\theta \mathbb{E}_{p_\theta}[f(\boldsymbol{x}, \theta)] \neq$

$\mathbb{E}_{p_\theta}[\nabla_\theta f(\boldsymbol{x}, \theta)]$, $\nabla_\theta \mathcal{L}(\theta)$ cannot in general be usefully approximated by $M^{-1} \sum \nabla_\theta f(\boldsymbol{x}_m, \theta)$ from samples $\boldsymbol{x}_m$ of $p_\theta$. To address this challenge, we observe

$$\mathcal{L}(\theta) = \mathcal{L}(\theta, \theta') := \mathbb{E}_{p_{\theta'}}\left[(p_\theta(\boldsymbol{x})/p_{\theta'}(\boldsymbol{x}))f(\boldsymbol{x}, \theta)\right], \quad (9)$$

for any set of model parameters $\theta'$. Since the expectation in Equation (9) does not depend on $\theta$, we *can* approximate its gradient with Monte Carlo samples from $p_{\theta'}$. The density ratio $p_\theta(\boldsymbol{x}_m)/p_{\theta'}(\boldsymbol{x}_m)$ in Equation (9) can be understood as the weights of an importance sampling estimate against target $p_\theta$ with proposal $p_{\theta'}$.

To estimate the gradients of the relax and reward losses, we choose proposal equal to the current model, i.e., $\theta'=\theta$. In this case, the importance weight is equal to 1 while its gradient is the "score" function $(\nabla_\theta p_\theta(\boldsymbol{x}_m))/p_\theta(\boldsymbol{x}_m) = \nabla \log p_\theta(\boldsymbol{x})$. Although the term $\mathcal{L}^{\mathrm{viol}}$ that appears in the relax loss is not of the form $\mathbb{E}_{p_\theta}[f(\boldsymbol{x}, \theta)]$, we can still construct an unbiased estimate of its gradient using importance sampling; see Appendix B.2. Combined with the variance-reduction strategies described in Appendix B.2, these gradient estimators perform well even in settings with high-dimensional latent variables, such as diffusion models and masked language models (Section 4.1).

## 2.4. Relationship Between Relax and Reward Losses

We showed in Section 2.2 that when the generative model $p_\theta$ is replaced by a probability distribution $p \in \mathcal{P}(p_{\theta_{\mathrm{base}}})$, the minimizer of the reward loss is the base model $p_{\theta_{\mathrm{base}}}$ tilted by the exponentiated reward with parameters $\boldsymbol{\alpha} = \boldsymbol{\alpha}^*$.

Less obviously, a similar statement holds for the relax loss: when $p_\theta$ is replaced by $p \in \mathcal{P}(p_{\theta_{\mathrm{base}}})$, the minimizer of the relax loss is equal to $p_\lambda(\boldsymbol{x}) \propto p_{\theta_{\mathrm{base}}}(\boldsymbol{x}) \exp\{r_{\boldsymbol{\alpha}_\lambda}(\boldsymbol{x})\}$, where the vector $\boldsymbol{\alpha}_\lambda$ depends on the regularization strength

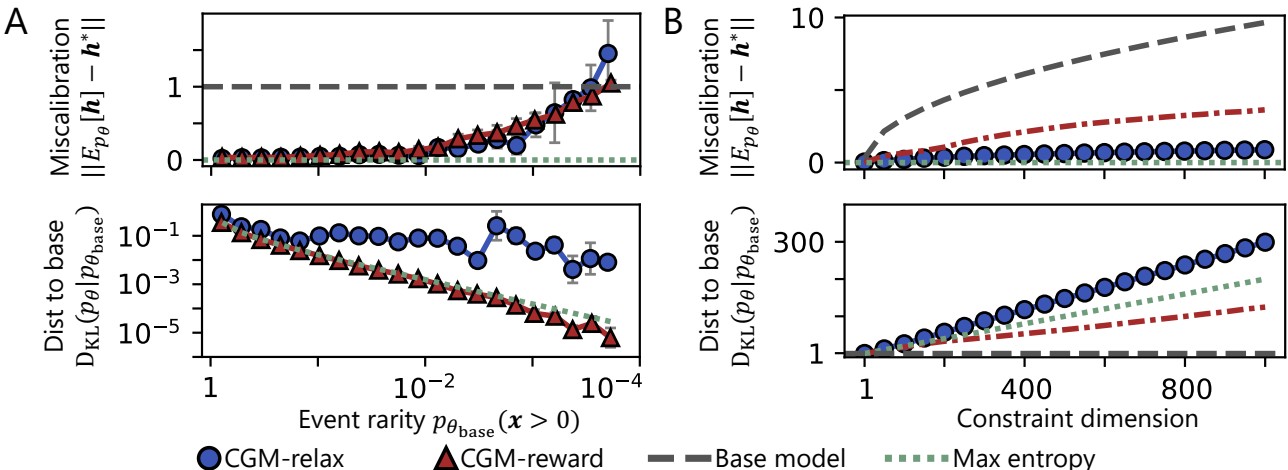

*Figure 3.* **A**: CGM successfully reweights the probability of a rare mode in a 1D Gaussian mixture. **B**: CGM-relax calibrates the base model to up to $10^3$ constraints, whereas CGM-reward is not well-defined for >30 constraints. CGM-relax outperforms CGM-reward even when $\widehat{\boldsymbol{\alpha}}_N$ is fixed to $\boldsymbol{\alpha}^*$ (red dashed line).

$\lambda$ and is not generally equal to $\boldsymbol{\alpha}^*$. However, both $\|\boldsymbol{\alpha}_\lambda - \boldsymbol{\alpha}^*\|$ and $\|\mathbb{E}_{q_\lambda}[\boldsymbol{h}(\boldsymbol{x})] - \boldsymbol{h}^*\|$ approach zero for small $\lambda$ at rate $\mathcal{O}(\lambda)$ (Hestenes, 1969). Appendix C.3 provides details.

Although the minimizer of the relax loss is an exponential tilt of the base model, minimizing the relax loss does *not* require estimating the parameters of the reward $\boldsymbol{\alpha}_\lambda$. In contrast, CGM-reward requires estimating $\boldsymbol{\alpha}^*$ through Equation (7). As we will see in Section 3, this distinction is important in settings with high-dimensional constraints, where Equation (7) can become infeasible and CGM-reward may therefore be inapplicable.

## 3. Simulations: Determining when CGM Thrives and Struggles

To understand the success and failure cases of CGM, we perform evaluations in a tractable setting. This setting allows us to understand the role of the CGM hyperparameters $\lambda$ and $N$, and to test CGM in challenging cases, including rare events and high-dimensional constraints.

We provide additional details on our simulation experiments in Appendices D to F. Appendix D provides background on continuous-time diffusion models, Appendix E compares CGM to the augmented Lagrangian algorithm (Hestenes, 1969; Khalafi et al., 2026) for these simulation experiments, and Appendix F discusses implementation.

**Simulation setup and evaluation.** We consider fine-tuning a diffusion generative model targeting a Gaussian mixture model (GMM) to reweight the mixture proportions of each mode. Here, $p_\theta(\boldsymbol{x})$ is a generative model of continuous paths $\boldsymbol{x} = (\boldsymbol{x}(t))_{t \in [0,1]}$, whose evolution is described by a stochastic differential equation (SDE).

Evaluating CGM on a diffusion model whose terminal distribution is a GMM has several advantages. First, we may choose the base diffusion model so that the final marginal $p_\theta(\boldsymbol{x}(1))$ exactly matches the target GMM (Anderson, 1982; Song et al., 2021); this enables us to focus solely on calibration rather than fitting the base model. And since the calibration constraint depends on the path only at time $t=1$, we can compute the KL divergence of the maximum entropy solution to the base model, and thereby measure the suboptimality of the solutions produced by CGM.

**Selecting hyperparameters for CGM-relax and CGM-reward.** We first initialize our base model $p_{\theta_{\text{base}}}$ such that $p_{\theta_{\text{base}}}(\boldsymbol{x}(1))$ is a one-dimensional Gaussian mixture with two equally-weighted, well-separated modes (Figure 2A). We define the calibration problem

$$\boldsymbol{h}(\boldsymbol{x}) = \mathbb{1}\{\boldsymbol{x}(1) > 0\}, \ \ \boldsymbol{h}^* = 0.8$$

to upweight the right mode from $\mathbb{E}_{p_{\theta_{\text{base}}}}[\boldsymbol{h}(\boldsymbol{x}(1))] = 0.5$.

For CGM-relax, the regularization parameter $\lambda$ trades off between satisfying the constraint (small $\lambda$) and deviating from the base model (large $\lambda$) (Figure 2B). For CGM-reward, increasing $N$ results in more accurate recovery of $\boldsymbol{\alpha}^*$ and thereby a better approximation to the maximum entropy solution. For appropriate hyperparameters, both solve the calibration problem to high accuracy.

In the remaining experiments, we select hyperparameters following a heuristic. For CGM-relax, we choose $\log(\lambda)$ on a linear grid and perform calibration for each value. For CGM-reward, we use $N = 10^5$ samples to estimate $\boldsymbol{\alpha}^*$.

**Calibrating rare events.** Calibrating the proportion of generations belonging to rare classes is central to applications including protein ensemble modeling (Lewis et al., 2025)

and reinforcement learning (O'Kelly et al., 2018). To assess the performance of CGM in this setting, we consider a variation of the GMM reweighting problem where the goal is to reduce an increasingly small probability of the right mode $p_{\theta_{\text{base}}}(\boldsymbol{x}(1) > 0)$ by a factor of $2\times$. So that the scale of the initial calibration error remains constant even the probabilities become very small, we choose

$$\boldsymbol{h}(\boldsymbol{x}) = 2\mathbb{1}\{\boldsymbol{x}(1) > 0\}/p_{\theta_{\text{base}}}(\boldsymbol{x}(1) > 0), \ \ \boldsymbol{h}^* = 1.$$

With this choice of $\boldsymbol{h}$ and $\boldsymbol{h}^*$ the violation loss measures the error in $p_\theta(\boldsymbol{x}(1) > 0)$ *relative to* the target proportion and is equal to 1 at initialization regardless of $p_{\theta_{\text{base}}}(\boldsymbol{x}(1) > 0)$. We vary $p_{\theta_{\text{base}}}(\boldsymbol{x}(1) > 0)$ from 0.8 to approximately $10^{-4}$ and use batch size $M = 10^2$. For CGM-relax, we select the value of $\lambda$ for which the calibration error is smallest.

Both algorithms reduce the calibration error by approximately 80-90% at base proportion $10^{-2}$ and by approximately 50% at $10^{-3}$ (Figure 3A). The nontrivial reduction in calibration error at $10^{-3}$ is surprising since, on average, fewer than one sample per batch is drawn from the rare mode throughout fine-tuning. Constraint satisfaction degrades past this threshold, but we suspect larger batch sizes would enable calibrating increasingly rare events. For events rarer than $10^{-1}$, CGM-relax deviates farther from the base model compared to CGM-reward and the maximum entropy solution.

In Appendix F.1, we consider a second rare event setting in which, rather than reweighting the rare mode by a constant factor, it is upweighted to the fixed proportion 0.8. Similarly, for $p_{\theta_{\text{base}}}(\boldsymbol{x}(1) > 0)$ as small as $10^{-3}$, CGM satisfies the calibration constraint to high accuracy.

**Scalability to high-dimensional models and constraints.** We next evaluate how performance depends on the dimensionality, $k$, of the GMM and the constraint. We take the base model to be a product of one-dimensional GMMs with marginals as in Figure 2A. For the constraint, we choose

$$\boldsymbol{h}(\boldsymbol{x}) = [\mathbb{1}\{\boldsymbol{x}(1)[1] > 0\}, \dots, \mathbb{1}\{\boldsymbol{x}(1)[k] > 0\}],$$
$$\boldsymbol{h}^* = [0.8, \dots, 0.8],$$

where $\boldsymbol{x}(1)[i]$ is the $i$th dimension of $\boldsymbol{x}(1)$. Since both the base model $p_{\theta_{\text{base}}}$ and maximum entropy solution $p_{\boldsymbol{\alpha}^*}$ are independent across dimension, the KL distance between these two distributions grows linearly in dimension. The multimodality of this model, with $2^k$ modes, mimics the multimodality of practical generative models. We perform CGM-relax and CGM-reward with batch size $M = 10^4$. For CGM-relax, we select the largest $\lambda$ for which miscalibration is reduced by 90%.

In this high-dimensional regime, significant discrepancies emerge between CGM-relax and CGM-reward (Figure 3B). CGM-relax consistently eliminates the majority of constraint violation up to $k=10^3$, albeit with a non-trivial excess

KL divergence to $p_{\theta_{\text{base}}}$ compared to the maximum entropy solution $p_{\boldsymbol{\alpha}^*}$ that increases linearly with dimension. Although CGM-reward performs well for low-dimensional constraints ($<10$), we find that the empirical maximum entropy problem (7) is infeasible with high probability for $k>30$. Even when $\widehat{\boldsymbol{\alpha}}_N$ is fixed to its oracle value $\boldsymbol{\alpha}^*$ (Figure 3B), CGM-relax still outperforms CGM-reward.

## 4. Case-studies with Diverse Models, Data, and Constraints

We evaluate the capacity of CGM-reward and CGM-relax to solve practical calibration problems through three applications involving diverse models, data, and constraint types. Section 4.1 calibrates a diffusion model (Lin et al., 2024a) and a masked language model (Hayes et al., 2025) of protein structure to more closely match statistics of natural proteins. Section 4.2 calibrates a normalizing flow model (Zhai et al., 2025) of images to reduce class imbalances on the basis of LLM image-to-text annotations. Lastly, Section 4.3 calibrates a large language model to eliminate gender bias in generated children's stories (Riviere et al., 2024).

**Baselines.** Only two prior works have proposed algorithms that intend to solve the calibration problem. Khalifa et al. (2021) propose a method for LLMs that we compare to in Section 4.3. Second, Shen et al. (2024) propose a method for class-balancing in diffusion models. However, their method assumes an existing probabilistic classifier and so is not applicable in our setting.

**Compute cost.** Each experiment is run on a single H100 GPU. Appendix F provides details on experimental setup.

**CGM-relax v. CGM-reward in practice.** Altogether, our results demonstrate that CGM-relax with optimally tuned $\lambda$ more robustly reduces miscalibration than CGM-reward and offers a navigable trade-off between calibration and fidelity to the base-model. As a practical matter, CGM-reward has the benefit of avoiding a hyperparameter search and often also performs well. Under compute constraints, we recommend first trying CGM-reward, and if results are not satisfactory then trying CGM-relax with a full grid search.

### 4.1. Calibrating Protein Design Models to Match Statistics of Natural Proteins

Diffusion generative models have become a central tool in protein design (Trippe et al., 2023; Watson et al., 2023). However, heuristics such as reduced noise during sampling (e.g., Yim et al., 2023) have been necessary to ensure a high proportion of the sampled structures are biophysically plausible. These heuristics substantially reduce the diversity of samples compared to proteins found in nature and thereby pose a trade-off between reliability and diversity. For two

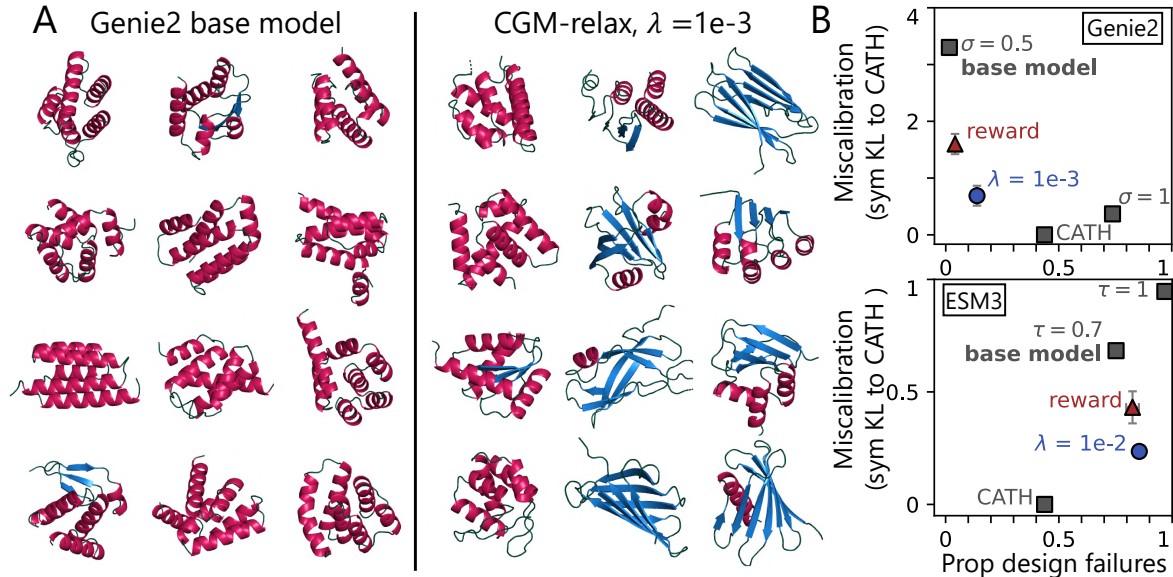

*Figure 4.* **A**: Samples from the Genie2 before and after calibration with CGM-relax ($\lambda=10^{-3}$). **B**: CGM-relax reduces the distance of secondary structure content to natural proteins by >4 times for Genie2 and >2 times for ESM3 while maintaining biophysical plausibility.

such protein design models, we investigate whether this trade-off can be mitigated by calibrating the models to match the secondary structure composition of natural proteins.

**Protein models Genie2 and ESM3 and their miscalibration.** The two protein design models we consider are (1) Genie2 (Lin et al., 2024a), a 15M parameter equivariant diffusion model, and (2) ESM3-open (Hayes et al., 2025), a 1.4B parameter masked language model on tokenized representations of protein backbones. For each model, we generate protein backbones consisting of 100 amino acids i.e., residues. Both Genie2 and ESM3-open suffer low diversity compared to natural protein domains in the CATH dataset (Sillitoe et al., 2021); specifically, they produce few generations with high beta-strand content (Figure 4A). Beta strands, along with alpha helices and loops, constitute what is known as a protein's secondary structure.

**Calibration constraints on secondary structure diversity.** To represent protein secondary structure as a calibration constraint, we use the empirical bivariate cumulative density function (CDF) of the fraction of residues in alpha-helical and beta-strand segments. We place up to $d = 99$ cutoff pairs $(\tau_{\alpha,i}, \tau_{\beta,i}) \in [0,1]^2$ and define a $d$-dimensional indicator vector $\boldsymbol{h}(\boldsymbol{x})$ with components

$$\boldsymbol{h}(\boldsymbol{x})[i] = \mathbb{1}\{f_\alpha(\boldsymbol{x}) \leq \tau_{\alpha,i},\ f_\beta(\boldsymbol{x}) \leq \tau_{\beta,i}\},\ i = 1,\ldots,d,$$

where $f_\alpha(\boldsymbol{x})$ and $f_\beta(\boldsymbol{x})$ are the secondary-structure fractions of protein structure $\boldsymbol{x}$. We set the calibration target $\boldsymbol{h}^*$ to the corresponding values of the CATH empirical bivariate CDF at these cutoffs.

**Results.** Calibration with CGM-relax yields a nearly five-

fold improvement in the diversity of sampled protein structures for Genie2 and a twofold improvement for ESM3-open, as quantified by the symmetrized KL distance between the secondary structure distributions of the generative models and CATH domains (Figure 4B). This improvement comes at the cost of an increased proportion of 'design failures', as defined in Appendix F.2. The ESM3-open base model generates a high proportion of design failures compared to Genie2 (consistent with e.g., Xiong et al. (2025)) and this fraction increases slightly upon calibration with CGM.

CGM-reward achieves more modest improvements in secondary structure diversity, which may in part be due to difficulty in computing $\widehat{\alpha}_N$. In order for Equation (7) to be feasible with $N = 2.5 \times 10^3$ samples, we need to reduce the number of cutoff pairs from 99 to 15. CGM-reward fine-tuning reduces the symmetrized KL distance to CATH by two times for Genie2 and 1.6 times for ESM3-open. However, for Genie2, CGM-reward also produces fewer design failures than CGM-relax.

The gains in secondary structure diversity achieved by CGM cannot be obtained by simply increasing the sampling noise of Genie2 or the sampling temperature of ESM3. In Figure 4B, we show that increasing the sampling noise of Genie2 from the base model value of $\sigma = 0.5$ to $\sigma = 1$ improves structure diversity, but at the cost of a much higher failure rate than CGM (74% vs 14%). The same is true for ESM3 with increased sampling temperature from $\tau = 0.7$ to $\tau = 1$, which yields a failure rate of 97%.

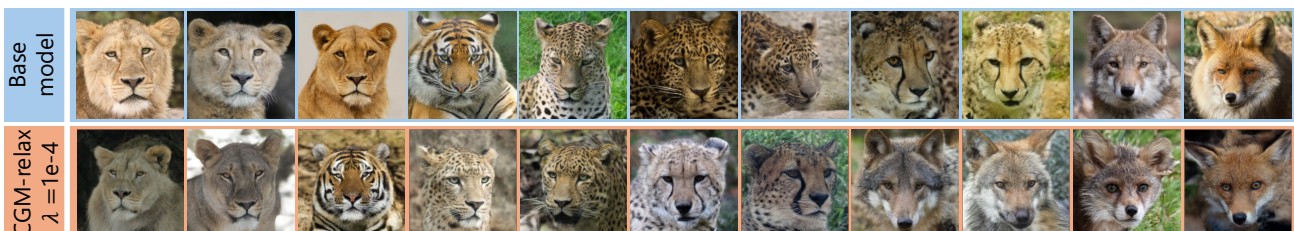

*Figure 5.* Generations from the conditional TarFlow model (Zhai et al., 2025) before and after calibration with CGM-relax ($\lambda = 10^{-4}$). CGM reweights the proportions of animals generated and produces realistic images. Some visual artifacts exist after calibration.

### 4.2. Calibrating a Conditional Flow Model to Balance Class Proportions

We next demonstrate that CGM is capable of effectively calibrating state-of-the-art normalizing flow models. Normalizing flows generate samples $x = f_\theta^{-1}(\epsilon)$, where $\epsilon \sim p_\epsilon$ is a distribution from which sampling is tractable and $f_\theta(x)$ is a map that is invertible in $x$ for each $\theta$ (Tabak & Vanden-Eijnden, 2010; Rezende & Mohamed, 2015). By the change-of-variable formula, the density of $x$ is $p_\epsilon(f_\theta(x))|\det(df_\theta(x)/dx)|$; this enables computation of exact likelihoods for training and calibration.

For our calibration problem, we consider the 463M-parameter TarFlow model (Zhai et al., 2025), which parameterizes $f_\theta$ as an autoregressive vision transformer (Dosovitskiy et al., 2021) that performs attention over a sequence of image patches. We examine the model trained conditionally on the $256 \times 256$ AFHQ dataset (Choi et al., 2020), which consists of images of animal faces belonging to one of three classes: {cat, dog, wildlife}. The wildlife class is further comprised of {lion, tiger, fox, wolf, cheetah, leopard}. We observe that, conditional on the wildlife class, approximately 36% of generations from the TarFlow model are lions and very few ($< 7\%$ total) are foxes or wolves. We apply CGM to calibrate the conditional TarFlow model to generate samples containing animals from the wildlife class with equal proportions. For $h$, we query GPT o5-mini to classify each image as containing one of the six animals or None.

**Results.** We find CGM-relax reduces miscalibration to the base TarFlow model with little visible degradation of sample realism (Figure 5). CGM-relax ($\lambda=10^{-4}$) reduces the total variation distance of animal proportions, as classified by an image-to-text model, to the uniform distribution from 0.306 to 0.101. However, the Fréchet inception distance (FID) to real images in the AFHQ wildlife class is larger for the calibrated model than for the base model (21.0 vs. 15.9). Since this metric is sensitive to class proportions, we evaluate the calibrated model on the training dataset after balancing classes. The discrepancy in FID can be explained by two types of visual artifacts introduced by calibration: some images depict animals outside the wildlife

class ($\sim 8\%$) and some "blend" multiple animals. Appendix Figure 11 shows random samples from both models. The model fine-tuned with CGM-reward remains close to the base model but fails to reduce constraint violation.

### 4.3. Calibrating an LLM to Reduce Gender-Profession Imbalance in Children's Stories

As a third example, we calibrate a large language model to generate short children's stories with reduced gender bias. Gemma-2-9B-IT is a nine-billion-parameter autoregressive transformer post-trained for instruction following Riviere et al. (2024). We find significant imbalances in prompt-conditional generations that introduce a character's profession. For example, only 18% of stories beginning "Once upon a time there was a lawyer" feature a female lawyer, whereas 41% of U.S. attorneys were women in 2024 (American Bar Association, 2024).

**Gender parity as a conditional calibration constraint.** We begin with a prompt from Eldan & Li (2023) to generate a short children's story and adapt it to specify the protagonist's profession. To quantify and correct gender imbalance, we define a statistic $h(x) \in \{-1, 0, 1\}$ encoding (male, ambiguous, female) and impose a profession-conditional parity constraint: for each profession $i$, we target $\mathbb{E}_{p_\theta}[h(x) \mid \text{prompt}_i] = 0$.

We first attempt to modify the prompt to request male and female characters with equal probability. This reduces imbalance but leaves substantial miscalibration (Figure 13), which we address with CGM. Rather than fine-tuning a separate model for each profession, we fine-tune a single LoRA adapter (Hu et al., 2022) by minimizing the sum of prompt-conditional CGM losses across all calibrated professions.

**Results on explicitly calibrated professions.** Both CGM-reward and CGM-relax reduce gender imbalance, measured as the average absolute per-profession frequency difference (Figure 6A) over five replicates per model. As expected, decreasing the regularization strength $\lambda$ improves constraint satisfaction at the cost of greater distance to the base model, as measured by symmetric KL. Notably, even the least-regularized model attains symmetric KL $<1.7$, correspond-

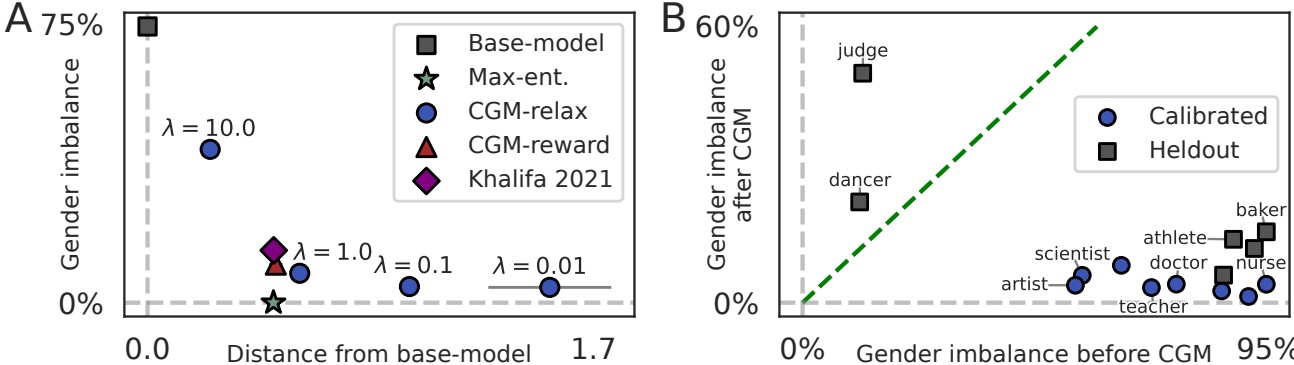

*Figure 6.* **A**: Gender imbalance and distance from base-model (symmetrized KL from pre-trained Gemma-2-9B-IT). **B**: Gender imbalance for professions included and heldout from calibration before and after CGM-relax ($\lambda = 0.01$). Points below the diagonal were improved.

ing to an average token log-probability difference of $<0.01$ nats/token. Appendix F.6.2 provides example generations before and after fine-tuning, showing no degradation in story quality.

Compared to Khalifa et al. (2021), CGM-reward yields a small but statistically significant (two-sided $t$-test) improvement in miscalibration at the same distance to the base model. CGM-relax reduces gender imbalance by more than threefold relative to the baseline but deviates further from the base model.

**Transference of calibration to held-out professions.** We evaluate how conditional calibration affects the calibration of "held-out" professions not considered during fine-tuning. This generalization is particularly valuable in applications where it is impractical to foresee and explicitly calibrate every possible prompt. To evaluate this, we consider six held-out professions; four are significantly improved, whereas two become more imbalanced (Figure 6B).

## 5. Conclusion

CGM-relax and CGM-reward provide practical approaches for calibrating generative models to satisfy distribution-level constraints. In applications to protein design, conditional image generation, and language modeling, CGM consistently reduces calibration error under hundreds of simultaneous constraints and in models with up to nine billion parameters while preserving generation quality.

Still, our results highlight that the calibration problem is not yet solved. Current objectives leave residual error, especially in the rare-event setting that is especially relevant to protein structure modeling, for example. More broadly, the CGM framework is tied to models with tractable likelihoods, raising the challenge of extending calibration to GANs and other implicit models. These open questions point to calibration as a fruitful research direction.

## Acknowledgments

We thank Julius Berner for his numerous helpful discussions about the connection between the calibration and reward fine-tuning problems as well as the inclusion of a baseline in the CGM-relax and CGM-reward gradient estimates; Tianyu Lu for sharing with us the dataset of secondary structure composition for CATH domains (Lu et al., 2025); Zhaoyang Li for his assistance with setting up the Genie2 and ESM3 evaluations; and Arthur Deng, Cole Citrenbaum, Nicholas Beltran, Rob Tibshirani, and Zhaoyang Li for providing feedback on our manuscript.

HDS is supported by the NSF Graduate Research Fellowship (DGE-2146755) and the Knight-Hennessy Graduate Fellowship. NLD is supported by the Stanford Graduate Fellowship (SGF).

## Author Contributions

BLT initiated the project, contributed algorithmic components, and provided supervision. HDS contributed the reward loss and its connections to maximum entropy and the relax loss, the diffusion and normalizing flow methodology and their implementations. NLD developed the discrete diffusion and LLM methods and implementation. All authors contributed to the codebase and writing.

## Impact Statement

The calibration methods developed in this work could offer potential ethical benefits including reducing biases (e.g., gender imbalance in text outputs) and improving scientific utility (e.g., protein structure design). However, as is the case for all works that fine-tune generative models, our methods could also in principle be misused, for example to enforce constraints that amplify discriminatory content. We emphasize that the choice of constraints should be made responsibly, with attention to societal and scientific impacts.

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

# Appendix Contents

# A. Extended Discussion of Related Work

In this section, we first discuss related works that address the generative model calibration problem for particular model classes. We then discuss a related class of algorithms that enforce constraints on a pre-trained model at sampling time, rather than by fine-tuning the model. Finally, we connect our work to reward fine-tuning and conditional generation.

**The calibration problem.** Several previous works have proposed algorithms whose goal it is to impose distributional constraints on generative models. However, each of these methods applies only to specific model classes and either suffers from poor empirical performance or imposes constraint satisfaction during training time (rather than fine-tuning).

Most closely related to the present work, Khalifa et al. (2021) fine-tune autoregressive language models to match distributional constraints. Like CGM-reward, their approach also targets the maximum entropy solution (5), but through a different divergence; they choose the KL divergence in the "forward" direction, $D_{KL}(p_{\alpha^*} \| p_\theta)$, rather than in the "reverse" direction, $D_{KL}(p_\theta \| p_{\alpha^*})$, as in CGM-reward.

Empirically, the approximate solutions to the calibration problem (1) found by Khalifa et al. (2021) fall short of constraint satisfaction compared to CGM, particularly CGM-relax. Khalifa et al. (2021) achieves comparable, albeit slightly worse, performance to CGM-reward in the Gemma 2 gender rebalancing experiment (Section 4.3), reducing miscalibration by roughly 81%. CGM-relax, on the other hand, reduces constraint violation up to 94%.

In follow-up work, Go et al. (2023) propose an algorithm for aligning language models to a specified target distribution by minimizing an arbitrary $f$-divergence (including the forward and reverse KL divergence). One example they consider is when the target distribution is the maximum entropy distribution corresponding to some constraint functions; the choice of forward KL then reduces to Khalifa et al. (2021). However, they obtain $< 50\%$ reduction in constraint violation.

Shen et al. (2024) propose a method for balancing class proportions in text-to-image diffusion models. They rely on an optimal transport objective that applies narrowly to diffusion models and find empirically their approach falls short of meeting desired class proportions.

Lewis et al. (2025) consider a fine-tuning diffusion generative model approximation of protein structure ensembles to protein stability measurements. They introduce an auxiliary fine-tuning loss that resembles the violation loss, but they do not demonstrate whether this approach leads to a significant reduction in calibration error.

A number of concurrent works also tackle the problem of generative model calibration problem for particular model classes. Khalafi et al. (2026) propose a primal-dual algorithm for fine-tuning diffusion models to a collection of expectation constraints. Their algorithm applies only to diffusion models and is validated on low-dimensional ($\leq 4$) constraints. Also concurrent to our work, Gutjahr et al. (2025) fine-tunes a diffusion generative model subject to inequality constraints on the expected value of a statistic to maximize an expected reward with a KL penalty to the base model. Their approach applies only to diffusion models and continuous normalizing flows.

**Incorporating distributional constraints during training.** Several other works have sought to impose distributional constraints during training time but differ from CGM in that they are not fine-tuning procedures and apply only to a specific model classes. Pitera & Chodera (2012) leverage the maximum entropy principle to derive a biasing potential that is applied to molecular simulations. This potential ensures that the ensemble average of an observable of interest matches a corresponding experimental measurement. Wu et al. (2020) propose a method for training generative adversarial networks (GANs) that includes a penalty term similar to $\mathcal{L}^{\text{viol}}$ that encourages agreement with statistics of the training data. Zhu et al. (2024) solve for the maximum entropy model of short (length 7) protein sequences with expected "fitness" surpassing a fixed threshold. Khalafi et al. (2024) propose a primal-dual algorithm to enforce distributional constraints on diffusion models; their constraints, however, are specified at the level of entire distributions, rather than their moments. Friedrich et al. (2023) develop a training procedure for diffusion models that balances the conditional distributions of samples, given some attribute e.g., gender.

**Distributional constraints beyond fine-tuning: density estimation, ensemble re-weighting, and inference time methods.** Beyond supervised learning, distributional constraints have appeared throughout computational statistics and generative modeling. In density estimation, Efron & Tibshirani (1996) suggest calibrating kernel density estimates by introducing an exponential tilt analogous to the maximum entropy tilt targeted by CGM-reward. In molecular simulation, similar exponential tilts are used to enforce agreement of ensembles with experimental measurements of ensemble observables (Pitera & Chodera, 2012; Różycki et al., 2011; Köfinger et al., 2019; Bottaro et al., 2020). And in the context of generative modeling *inference-time* methods have been introduced to bias sampling to produce an ensemble without altering model

weights. Inference-time methods have been applied to molecular modeling (Cardei et al., 2025; Kolloff et al., 2025) and image generation (Liu et al., 2021; Yao et al., 2024).

However these approaches do not inform how to fine-tune a generative model. Compared to sample reweighting and inference-time strategies, fine-tuning pays the compute cost up-front, and sampling from the fine-tuned model incurs the same cost as sampling from the base model. This tradeoff is advantageous when a model is to be used for downstream tasks such as conditional generation.

**Reward fine-tuning and conditional generation.** As we point out in Section 2.2, the idea of minimizing the KL divergence of the generative model to an exponential tilt of the base model (5) connects CGM to the rich research topic of reward fine-tuning. Reward fine-tuning algorithms, used in the contexts of reinforcement learning (Rafailov et al., 2023; Fan et al., 2023; Black et al., 2024; Wallace et al., 2024) and preference optimization (Tang, 2024; Uehara et al., 2024; Domingo-Enrich et al., 2025), minimize the same loss (8) as CGM-reward, but with $r_{\boldsymbol{\alpha}}(\boldsymbol{x})$ replaced by a user-specified "reward". Unlike reward fine-tuning algorithms, though, CGM does not require a reward; rather, the constraints themselves act as the reward.

Conditional generation (Dhariwal & Nichol, 2021; Ho & Salimans, 2021; Denker et al., 2024) can also be viewed through the lens of model calibration, where the calibration constraint is the indicator function of the set $C$ from which one would like to sample $\boldsymbol{h}(\boldsymbol{x}) = \mathbb{1}\{\boldsymbol{x} \in C\}$ and $\boldsymbol{h}^*$, the target proportion of samples that belong to $C$, approaches 1. In this case the optimal variational parameter $\boldsymbol{\alpha}^*$ approach infinity, and the maximum entropy solution approaches $p_{\theta_{\text{base}}}(\boldsymbol{x})\mathbb{1}\{\boldsymbol{x} \in C\}$.

# B. CGM-relax and CGM-reward Algorithms

In this section, we provide further detail on the CGM-relax and CGM-reward algorithms. First, we show in Appendix B.1 that our estimates for the relax and reward losses are unbiased. In Appendix B.2 we then discuss how to compute our gradient estimates for the relax and reward losses, and we show they are unbiased.

Throughout this section we will make the following regularity assumptions on the generative model $p_\theta$ and the constraint functions $\boldsymbol{h}$.

**Assumption B.1** (Regularity of $p_\theta$). The functions $p_{\tilde{\theta}}(\boldsymbol{x})/p_\theta(\boldsymbol{x})$, $\nabla_{\tilde{\theta}} p_{\tilde{\theta}}(\boldsymbol{x})/p_\theta(\boldsymbol{x})$, $\log p_{\tilde{\theta}}(\boldsymbol{x})$, $\nabla_{\tilde{\theta}} \log p_{\tilde{\theta}}(\boldsymbol{x})$ are uniformly dominated by a function that is square integrable with respect to $p_\theta(\boldsymbol{x})$, for all $\tilde{\theta}$ belonging to some neighborhood of $\theta$. Also, $\boldsymbol{h}(\boldsymbol{x})$, $\log p_{\theta_{\text{base}}}(\boldsymbol{x})$ have finite second moment under $p_\theta(\boldsymbol{x})$.

These assumptions are sufficient to exchange integration and differentiation in Appendix B.2 with dominated convergence.

## B.1. Loss Estimates

We begin by proving that our estimates $\widehat{\mathcal{L}}^{\text{relax}}$ and $\widehat{\mathcal{L}}^{\text{reward}}$ for $\mathcal{L}^{\text{relax}}$ and $\mathcal{L}^{\text{reward}}$, respectively, are, on average, correct.

**Proposition B.2.** $\widehat{\mathcal{L}}^{\text{relax}}$ *is unbiased for the relax loss* $\mathcal{L}^{\text{relax}}$.

*Proof.* We prove unbiasedness of $\widehat{\mathcal{L}}^{\text{relax}}$ by showing that $\widehat{\mathcal{L}}^{\text{KL}}$ is unbiased for $\mathcal{L}^{\text{KL}} = \mathrm{D}_{\text{KL}}\left(p_\theta \parallel p_{\theta_{\text{base}}}\right)$ and that $\widehat{\mathcal{L}}^{\text{viol}}$ is unbiased for $\mathcal{L}^{\text{viol}} = \|\mathbb{E}_{p_\theta}[\boldsymbol{h}(\boldsymbol{x})] - \boldsymbol{h}^*\|^2$.

As for $\widehat{\mathcal{L}}^{\text{KL}}$, its expectation is

$$\mathbb{E}_{p_\theta}\left[\widehat{\mathcal{L}}^{\text{KL}}\right] = \frac{1}{M}\sum_{m=1}^{M}\mathbb{E}_{p_\theta}\left[\log\frac{p_\theta(\boldsymbol{x}_m)}{p_{\theta_{\text{base}}}(\boldsymbol{x}_m)}\right] = \frac{1}{M}\sum_{m=1}^{M}\mathrm{D}_{\text{KL}}\left(p_\theta \parallel p_{\theta_{\text{base}}}\right) = \mathrm{D}_{\text{KL}}\left(p_\theta \parallel p_{\theta_{\text{base}}}\right).$$

In the first equality we invoke the linearity of expectation and in the second we use our assumption that $\{\boldsymbol{x}_m\}_{m=1}^{M}$ are sampled from $p_\theta$.

And for $\widehat{\mathcal{L}}^{\text{viol}}$, we recall that for a real-valued random variable $Z$, $\mathbb{E}[Z^2] = \mathbb{E}[Z]^2 + \mathrm{Var}(Z)$. Applying this to each dimension of $M^{-1}\sum_{m=1}^{M} \tilde{\boldsymbol{h}}_m = M^{-1}\sum_{m=1}^{M}(\boldsymbol{h}(\boldsymbol{x}_m) - \boldsymbol{h}^*)$, we obtain

$$\mathbb{E}_{p_\theta}\left\|\frac{1}{M}\sum_{m=1}^{M}\tilde{\boldsymbol{h}}_m\right\|^2 = \|\mathbb{E}_{p_\theta}[\tilde{\boldsymbol{h}}(\boldsymbol{x})]\|^2 + \frac{1}{M}\mathbb{E}_{p_\theta}\|\tilde{\boldsymbol{h}}(\boldsymbol{x}) - \mathbb{E}_{p_\theta}[\tilde{\boldsymbol{h}}(\boldsymbol{x})]\|^2, \tag{10}$$

where $\tilde{\boldsymbol{h}}(\boldsymbol{x}) = \boldsymbol{h}(\boldsymbol{x}) - \boldsymbol{h}^*$. Next, we replace the final term in (10) with $\mathbb{E}_{p_\theta}[M^{-1}(M-1)^{-1}\sum_m \|\tilde{\boldsymbol{h}}_m - M^{-1}\sum_{m'}\tilde{\boldsymbol{h}}_{m'}\|^2]$. The quantity $M^{-1}(M-1)^{-1}\sum_m \|\tilde{\boldsymbol{h}}_m - M^{-1}\sum_{m'}\tilde{\boldsymbol{h}}_{m'}\|^2$ is simply the trace of the sample covariance matrix of $\{\tilde{\boldsymbol{h}}_m\}_{m=1}^M$, scaled by $M^{-1}$. The sample covariance of $\{\tilde{\boldsymbol{h}}_m\}_{m=1}^M$ is unbiased for $\mathrm{Cov}[\tilde{\boldsymbol{h}}]$. Rearranging the above expression yields

$$\|\mathbb{E}_{p_\theta}[\tilde{\boldsymbol{h}}(\boldsymbol{x})]\|^2 = \mathbb{E}_{p_\theta}\left\|\frac{1}{M}\sum_{m=1}^M \tilde{\boldsymbol{h}}_m\right\|^2 - \frac{1}{M}\mathbb{E}_{p_\theta}\left[\frac{1}{(M-1)}\sum_{m=1}^M \left\|\tilde{\boldsymbol{h}}_m - \frac{1}{M}\sum_{m'=1}^M \tilde{\boldsymbol{h}}_{m'}\right\|^2\right]$$

$$= \mathbb{E}_{p_\theta}[\widehat{\mathcal{L}}^{\mathrm{viol}}]$$

This proves $\widehat{\mathcal{L}}^{\mathrm{viol}}$ is unbiased for $\|\mathbb{E}_{p_\theta}[\boldsymbol{h}(\boldsymbol{x})] - \boldsymbol{h}^*\|^2$. $\qquad\square$

Likewise, we demonstrate that our estimate for the reward loss is unbiased.

**Proposition B.3.** $\widehat{\mathcal{L}}^{\mathrm{reward}}$ *is unbiased for the reward loss* $\mathcal{L}^{\mathrm{reward}}$.

*Proof.* In the proof of Proposition B.2 we already demonstrated $M^{-1}\sum_{m=1}^M \log\frac{p_\theta(\boldsymbol{x}_m)}{p_{\theta_{\mathrm{base}}}(\boldsymbol{x}_m)}$ is unbiased for $\mathcal{L}^{\mathrm{KL}}$. By an identical argument, $-M^{-1}\sum_{m=1}^M r_{\widehat{\boldsymbol{\alpha}}_N}(\boldsymbol{x}_m)$ is unbiased for $\mathcal{L}^{\mathrm{r}} = \mathbb{E}_{p_\theta}[-r_{\widehat{\boldsymbol{\alpha}}_N}(\boldsymbol{x})]$ (again, it is a Monte Carlo estimate). $\qquad\square$

### B.2. Unbiased Gradient Estimates

As we detailed in Section 2.3, the naïve idea of taking the unbiased loss estimators $\widehat{\mathcal{L}}^{\mathrm{relax}}$, $\widehat{\mathcal{L}}^{\mathrm{reward}}$ and differentiating them with respect to $\theta$ will not yield unbiased estimates for the gradients of $\mathcal{L}^{\mathrm{relax}}$ and $\mathcal{L}^{\mathrm{reward}}$. This is because the probability distribution with respect to which the expectation is taken also depends on $\theta$, which needs to be taken into account in the gradient estimate.

For CGM-reward, we propose the gradient estimate

$$\widehat{G}^{\mathrm{reward}} = \frac{1}{M}\sum_{m=1}^M (\nabla_\theta w_m(\theta,\theta'))\left(l_m^{\mathrm{LOO}} - r_m^{\mathrm{LOO}}\right), \quad w_m(\theta,\theta') = \frac{p_\theta(\boldsymbol{x}_m)}{p_{\theta'}(\boldsymbol{x}_m)}$$

$$l_m^{\mathrm{LOO}} = l_m - \frac{1}{M-1}\sum_{m'\neq m} l_{m'}, \quad l_m = \log\frac{p_\theta(\boldsymbol{x}_m)}{p_{\theta_{\mathrm{base}}}(\boldsymbol{x}_m)} \tag{11}$$

$$r_m^{\mathrm{LOO}} = r_m - \frac{1}{M-1}\sum_{m'\neq m} r_{m'}, \quad r_m = r_{\widehat{\boldsymbol{\alpha}}_N}(\boldsymbol{x}_m)$$

As we explained in Section 2.3, $w_m(\theta,\theta')$ can be viewed as the weights of an *importance sampling* scheme, where $p_{\theta'}$ is the proposal distribution and $p_\theta$ is the target distribution. We choose $\theta' = \theta$ so that the proposal distribution is equal to the target distribution. For this choice of proposal, the weights $w_m$ are all equal to 1. However, their gradient with respect to $\theta$ is equal to the score of the calibrated model at $\boldsymbol{x}_m$, $\nabla_\theta \log p_\theta(\boldsymbol{x}_m)$. The expression (11), excluding the terms $(M-1)^{-1}\sum_{m'\neq m} l_{m'}$ and $(M-1)^{-1}\sum_{m'\neq m} r_{m'}$ is known as the score function gradient estimate or, in the terminology of reinforcement learning, the REINFORCE gradient estimate (Williams, 1992).

The terms $(M-1)^{-1}\sum_{m'\neq m} l_{m'}$ and $(M-1)^{-1}\sum_{m'\neq m} r_{m'}$ in (11) are known as leave-one-out baselines (Kool et al., 2019) corresponding to sample $\boldsymbol{x}_m$. Including these terms adds to the score function gradient estimate a *control variate*, which is a term that has expectation zero under $p_\theta$ but is correlated with each individual term in the estimate (Lavenberg & Welch, 1981; Ranganath et al., 2014; Mohamed et al., 2020). Indeed, we observe that by independence of the samples $\{\boldsymbol{x}_m\}_{m=1}^M$, it holds that for each $m \neq m'$,

$$\mathbb{E}_{p_\theta}\left[(\nabla_\theta \log p_\theta(\boldsymbol{x}_m))(l_{m'} - r_{m'})\right] = \mathbb{E}_{p_\theta}\left[\nabla_\theta \log p_\theta(\boldsymbol{x}_m)\right]\mathbb{E}_{p_\theta}\left[l_{m'} - r_{m'}\right] = 0.$$

Consequently, while the inclusion of the leave-one-out averages does not affect the unbiasedness of our gradient estimate, they can reduce its variance.

**Proposition B.4.** $\widehat{G}^{\mathrm{reward}}$ *is unbiased for the gradient of the reward loss,* $\nabla_\theta \mathcal{L}^{\mathrm{reward}}$.

*Proof.* We start by writing out the gradient of $\mathcal{L}^{\text{reward}}$ directly:

$$
\begin{aligned}
\nabla_\theta \mathcal{L}^{\text{reward}}(\theta) &= \nabla_\theta \mathbb{E}_{p_\theta} \left[ \log \frac{p_\theta(\boldsymbol{x})}{p_{\theta_{\text{base}}}(\boldsymbol{x})} - r_{\widehat{\boldsymbol{\alpha}}_N}(\boldsymbol{x}) \right] \\
&= \nabla_\theta \int \left\{ \log \frac{p_\theta(\boldsymbol{x})}{p_{\theta_{\text{base}}}(\boldsymbol{x})} - r_{\widehat{\boldsymbol{\alpha}}_N}(\boldsymbol{x}) \right\} p_\theta(d\boldsymbol{x}) \\
&= \nabla_\theta \int \frac{p_\theta(\boldsymbol{x})}{p_{\theta'}(\boldsymbol{x})} \left\{ \log \frac{p_\theta(\boldsymbol{x})}{p_{\theta_{\text{base}}}(\boldsymbol{x})} - r_{\widehat{\boldsymbol{\alpha}}_N}(\boldsymbol{x}) \right\} p_{\theta'}(d\boldsymbol{x}) \\
&\stackrel{(\star)}{=} \mathbb{E}_{p_\theta} \left[ \left( \nabla_\theta \frac{p_\theta(\boldsymbol{x})}{p_{\theta'}(\boldsymbol{x})} \right) \left\{ \log \frac{p_\theta(\boldsymbol{x})}{p_{\theta_{\text{base}}}(\boldsymbol{x})} - r_{\widehat{\boldsymbol{\alpha}}_N}(\boldsymbol{x}) \right\} \right] + \mathbb{E}_{p_\theta} \left[ \nabla_\theta \frac{p_\theta(\boldsymbol{x})}{p_{\theta'}(\boldsymbol{x})} \right] \quad (12)
\end{aligned}
$$

where $\theta' = \texttt{stop-grad}(\theta)$. In equality $(\star)$, exchange of the gradient and expectation is permissible as a consequence of dominated convergence and Assumption B.1. The second term is the expected score, which is zero. And so the gradient of the reward loss is

$$
\nabla_\theta \mathcal{L}^{\text{reward}}(\theta) = \mathbb{E}_{p_\theta} \left[ (\nabla_\theta \log p_\theta(\boldsymbol{x})) \left\{ \log \frac{p_\theta(\boldsymbol{x})}{p_{\theta_{\text{base}}}(\boldsymbol{x})} - r_{\widehat{\boldsymbol{\alpha}}_N}(\boldsymbol{x}) \right\} \right]. \quad (13)
$$

Looking at our gradient estimator $\widehat{G}^{\text{reward}}$ in (11) and ignoring the leave-one-out averages, we see that it is exactly the Monte Carlo estimate of the gradient of $\mathcal{L}^{\text{reward}}$ (13). $\qquad\square$

Dropping the potentially noisy expected score term in (12), as is done by Ranganath et al. (2014), also reduces variance of our gradient estimate.

Deriving an unbiased gradient estimate for the relax loss is more challenging, since the loss cannot be written as the expectation of some objective under $p_\theta$. Just as we did for the reward loss, we can compute an unbiased estimate for the gradient of $\mathcal{L}^{\text{KL}}$ in the relax loss by drawing independent samples $\boldsymbol{x}_m \sim p_{\theta'}$ and then differentiating the importance sampling weights $w_m(\theta, \theta')$

$$
\widehat{G}_{\text{KL}} = \frac{1}{M} \sum_{m=1}^{M} (\nabla_\theta w_m(\theta, \theta')) l_m^{\text{LOO}}.
$$

And so it remains to compute an unbiased gradient estimate for $\mathcal{L}^{\text{viol}}$. To do so, we first recall the unbiased estimate $\widehat{\mathcal{L}}^{\text{viol}}$ for $\mathcal{L}^{\text{viol}}$ that we introduced in Section 2.1

$$
\widehat{\mathcal{L}}^{\text{viol}}(\{\tilde{\boldsymbol{h}}_m\}_{m=1}^M) := \left\| \frac{1}{M} \sum_{m=1}^{M} \tilde{\boldsymbol{h}}_m \right\|^2 - \frac{1}{M(M-1)} \sum_{m=1}^{M} \left\| \tilde{\boldsymbol{h}}_m - \frac{1}{M} \sum_{m'=1}^{M} \tilde{\boldsymbol{h}}_{m'} \right\|^2,
$$

where $\boldsymbol{x}_m$ are independent samples from $p_\theta$ and $\tilde{\boldsymbol{h}}_m = \boldsymbol{h}(\boldsymbol{x}_m) - \boldsymbol{h}^*$. We propose a modification to this estimate wherein we draw independent samples $\boldsymbol{x}_m \sim p_{\theta'}$ and replace $\{\tilde{\boldsymbol{h}}_m\}_{m=1}^M$ by $\{w_m(\theta, \theta') \tilde{\boldsymbol{h}}_m\}_{m=1}^M$. To estimate the gradient of $\|\mathbb{E}_{p_\theta}[\boldsymbol{h}] - \boldsymbol{h}^*\|^2 = \|\mathbb{E}_{p_\theta}[\tilde{\boldsymbol{h}}]\|^2$, we compute the gradient of $\widehat{\mathcal{L}}^{\text{viol}}(\{w_m(\theta, \theta') \tilde{\boldsymbol{h}}_m\}_{m=1}^M)$ with respect to $\theta$ and then evaluate the result at $\theta' = \theta$. In Algorithms 1 and 2, we implement our gradient estimate for $\mathcal{L}^{\text{viol}}$ by sampling $\boldsymbol{x}_m$ independently from $p_{\texttt{stop-grad}(\theta)}$ and differentiating $\widehat{\mathcal{L}}^{\text{viol}}(\{w_m(\theta, \theta') \tilde{\boldsymbol{h}}_m\}_{m=1}^M)$ with $\theta' = \texttt{stop-grad}(\theta)$.

This yields the overall gradient estimator for the relax loss

$$
\widehat{G}^{\text{relax}} = \nabla_\theta \widehat{\mathcal{L}}^{\text{viol}} \left( \left\{ w_m(\theta, \theta') \tilde{\boldsymbol{h}}_m \right\}_{m=1}^M \right) + \lambda \widehat{G}_{\text{KL}}, \quad \boldsymbol{x}_m \stackrel{i.i.d.}{\sim} p_{\theta'}, \quad \theta' = \texttt{stop-grad}(\theta).
$$

In order to prove that $\widehat{G}^{\text{relax}}$ is unbiased for $\nabla_\theta \mathcal{L}^{\text{relax}}$, we need to show $\widehat{\mathcal{L}}^{\text{viol}}(\{w_m(\theta, \theta') \tilde{\boldsymbol{h}}_m\}_{m=1}^M)$ remains unbiased for $\mathcal{L}^{\text{viol}}$ when $\boldsymbol{x}_m$ are sampled independently from $p_{\theta'}$. Then, since the distribution from which $\boldsymbol{x}_m$ are sampled does not depend on $\theta$, it is allowable to exchange the gradient with the expectation.

**Proposition B.5.** $\widehat{G}^{\text{relax}}$ *is unbiased for the gradient of the relax loss,* $\nabla_\theta \mathcal{L}^{\text{relax}}$.

*Proof.* From Proposition B.2, we know that $\widehat{G}_{\mathrm{KL}}$ is unbiased for $\nabla_\theta \mathcal{L}^{\mathrm{KL}}$, and so it only remains to verify that the second term is unbiased for $\nabla_\theta \mathcal{L}^{\mathrm{viol}} = \nabla_\theta \|\mathbb{E}_{p_\theta}[\boldsymbol{h}] - \boldsymbol{h}^*\|^2$. To this end, by repeating the proof of Proposition B.2 (i.e., using the definition of the variance), it is straightforward to show

$$
\mathbb{E}_{p_{\theta'}}\left[\widehat{\mathcal{L}}^{\mathrm{viol}}\left(\left\{\frac{p_\theta(\boldsymbol{x}_m)}{p_{\theta'}(\boldsymbol{x}_m)}\tilde{\boldsymbol{h}}_m\right\}_{m=1}^M\right)\right] = \left\|\mathbb{E}_{p_{\theta'}}\left[\frac{p_\theta(\boldsymbol{x}_m)}{p_{\theta'}(\boldsymbol{x}_m)}\tilde{\boldsymbol{h}}_m\right]\right\|^2 = \|\mathbb{E}_{p_\theta}[\tilde{\boldsymbol{h}}]\|^2.
$$

In other words, $\widehat{\mathcal{L}}^{\mathrm{viol}}(\{w_m(\theta,\theta')\tilde{\boldsymbol{h}}_m\}_{m=1}^M)$ is unbiased for $\mathcal{L}^{\mathrm{viol}}$. However, since the samples $\{\boldsymbol{x}_m\}_{m=1}^M$ are drawn from $p_{\theta'}$, a probability distribution that does not depend on $\theta$, then we can exchange the gradient and expectation by appealing to dominated convergence and Assumption B.1. In particular, we have

$$
\mathbb{E}_{p_{\theta'}}\left[\nabla_\theta\widehat{\mathcal{L}}^{\mathrm{viol}}\left(\left\{\frac{p_\theta(\boldsymbol{x}_m)}{p_{\theta'}(\boldsymbol{x}_m)}\tilde{\boldsymbol{h}}_m\right\}_{m=1}^M\right)\right] = \nabla_\theta\mathbb{E}_{p_{\theta'}}\left[\widehat{\mathcal{L}}^{\mathrm{viol}}\left(\left\{\frac{p_\theta(\boldsymbol{x}_m)}{p_{\theta'}(\boldsymbol{x}_m)}\tilde{\boldsymbol{h}}_m\right\}_{m=1}^M\right)\right]
$$
$$
= \nabla_\theta\mathcal{L}^{\mathrm{viol}},
$$

where the final line follows from the unbiasedness of $\widehat{\mathcal{L}}^{\mathrm{viol}}(\{w_m\tilde{\boldsymbol{h}}_m\}_{m=1}^M)$ for $\mathcal{L}^{\mathrm{viol}}$. $\qquad\square$

As we discussed, the key insight from the proof of Proposition B.5 is that, by introducing importance weights, we can compute an unbiased estimate to $\|\mathbb{E}_{p_\theta}[\boldsymbol{h}] - \boldsymbol{h}^*\|^2 = \|\mathbb{E}_{p_\theta}[\tilde{\boldsymbol{h}}]\|$ without sampling directly from $p_\theta$.

## C. Maximum Entropy Principle

In this section, we provide an overview of the maximum entropy principle, which we use in Section 2.2 to define the reward loss $\mathcal{L}^{\mathrm{reward}}$. First, in Appendix C.1 we formally state and prove the maximum entropy principle. In Appendix C.2, we provide greater detail on our estimate $\widehat{\boldsymbol{\alpha}}_N$ for the parameters $\boldsymbol{\alpha}^*$ of the maximum entropy solution. In Appendix C.3, we characterize the relationship between the relax and reward losses by considering a problem whose solution is close to the optimum of the relax loss, and which resembles the maximum entropy problem. Lastly, in Appendix C.4, we study the behavior of the estimate $\widehat{\boldsymbol{\alpha}}_N$ in the limit as the number of samples $N$ becomes large.

Prior to jumping into the details of the maximum entropy principle, we work through an illustrative example that we discuss throughout this section.

**Example.** Suppose $\boldsymbol{x} \in \mathbb{R}$, $\boldsymbol{h}(\boldsymbol{x}) = \mathbb{1}\{\boldsymbol{x} > 0\}$, and $\boldsymbol{h}^* \in \mathbb{R}$. Also define $h_b = \mathbb{P}_{p_{\theta_{\mathrm{base}}}}(\boldsymbol{x} > 0)$, and assume $0 < h_b < 1$. In this example, the calibration problem amounts to either upweighting or downweighting the amount of probability mass $h_b$ that lies above $0$ under the base model $p_{\theta_{\mathrm{base}}}$. By Theorem 2.1, the maximum entropy solution has the form $p_{\boldsymbol{\alpha}^*} \propto p_{\theta_{\mathrm{base}}}(\boldsymbol{x})\exp\{\boldsymbol{\alpha}^*\boldsymbol{h}(\boldsymbol{x})\}$ for some $\boldsymbol{\alpha}^* \in \mathbb{R}$ that we need to determine. From this expression for $p_{\boldsymbol{\alpha}^*}$, we obtain

$$
1 - \boldsymbol{h}^* = \mathbb{E}_{p_{\boldsymbol{\alpha}^*}}[1 - \boldsymbol{h}(\boldsymbol{x})] = \frac{1}{h_b\exp(\boldsymbol{\alpha}^*) + (1 - h_b)}(1 - h_b),
$$
$$
\boldsymbol{h}^* = \mathbb{E}_{p_{\boldsymbol{\alpha}^*}}[\boldsymbol{h}(\boldsymbol{x})] = \frac{1}{h_b\exp(\boldsymbol{\alpha}^*) + (1 - h_b)}h_b\exp(\boldsymbol{\alpha}^*).
$$

Dividing the first equation by the second and rearranging yields $\boldsymbol{\alpha}^* = \log(\frac{\boldsymbol{h}^*(1-h_b)}{(1-\boldsymbol{h}^*)h_b})$. Following the same argument for the empirical distribution of $\{\boldsymbol{x}_n\}_{n=1}^N$, our estimator for $\boldsymbol{\alpha}^*$ is $\widehat{\boldsymbol{\alpha}}_N = \log(\frac{\boldsymbol{h}^*(1-\bar{\boldsymbol{y}}_N)}{(1-\boldsymbol{h}^*)\bar{\boldsymbol{y}}_N})$, where $\bar{\boldsymbol{y}}_N = \frac{1}{N}\sum_{n=1}^N \boldsymbol{y}_n$, $\boldsymbol{y}_n = \mathbb{1}\{\boldsymbol{x}_n > 0\} \stackrel{d}{=} \mathrm{Bernoulli}(h_b)$ for $\boldsymbol{x}_n \stackrel{i.i.d.}{\sim} p_{\theta_{\mathrm{base}}}$.

We point out that $\boldsymbol{\alpha}^*$ and $\widehat{\boldsymbol{\alpha}}_N$ can equivalently be derived by differentiating the objectives (6) and (7), respectively, and setting them equal to $0$.

### C.1. Precise Statement

Since the maximum entropy problem is not specific to generative model calibration, we present it in a more general setting. Our presentation builds on standard results from exponential families and convex analysis. We recommend Wainwright & Jordan (2008) for relevant background.

In particular, we consider $X := (X, \mathcal{X})$ a measurable space, $P$ a probability measure defined on $X$, $\boldsymbol{h} : X \to \mathbb{R}^d$ an $X$-measurable constraint function, and $\boldsymbol{h}^*$ a target value for the moment of $\boldsymbol{h}$. The maximum entropy problem corresponding to probability measure $P$, constraint $\boldsymbol{h}$, and target moment $\boldsymbol{h}^*$ is

$$\inf_{Q \in \mathcal{P}(P)} D_{\mathrm{KL}} \left(Q \parallel P\right), \text{ s.t. } \mathbb{E}_Q[\boldsymbol{h}(\boldsymbol{x})] = \boldsymbol{h}^*. \tag{14}$$

$\mathcal{P}(P)$ is the collection of all probability measures having a density with respect to $P$, which, by the Radon-Nikodym theorem, is equal to the collection of all absolutely continuous probability measures with respect to $P$. Choosing $P = p_{\theta_{\mathrm{base}}}$ yields the maximum entropy problem corresponding to the calibration problem.

As we mentioned in Section 2.2, we impose a condition on the target moment $\boldsymbol{h}^*$ to ensure (i) there exists a solution to the maximum entropy problem (ii) and this solution is an exponential tilt of $P$.

**Assumption C.1** (Interior moment condition). Define the subset $\mathcal{M}$ of $\mathbb{R}^d$ comprised of all possible moments of $\boldsymbol{h}$ attainable by probability distributions $Q$ having a density with respect to $P$

$$\mathcal{M} = \left\{ \mathbb{E}_Q[\boldsymbol{h}(\boldsymbol{x})] \ \middle| \ Q \in \mathcal{P}(P), \ \mathbb{E}_Q[\|\boldsymbol{h}(\boldsymbol{x})\|] = \int \|\boldsymbol{h}(\boldsymbol{x})\| Q(d\boldsymbol{x}) < \infty \right\}.$$

$\boldsymbol{h}^*$ lies in the *relative interior* of $\mathcal{M}$, written $\mathrm{relint}(\mathcal{M})$.

Since $\mathcal{M}$ is a convex set, the condition $\boldsymbol{h}^* \in \mathrm{relint}(\mathcal{M})$ can equivalently be stated as for every $\boldsymbol{y} \neq \boldsymbol{h}^*$ in $\mathcal{M}$, there exists some $\boldsymbol{z}$ in $\mathcal{M}$ and $\kappa \in (0, 1)$ for which $\boldsymbol{h}^* = \kappa \boldsymbol{z} + (1 - \kappa)\boldsymbol{y}$.

To see why Assumption C.1 is necessary for the solution to be an exponential tilt of $p_{\theta_{\mathrm{base}}}$, recall the example discussed at the beginning of Appendix C. In this case, $\mathrm{relint}(\mathcal{M}) = (0, 1)$. If $\boldsymbol{h}^* \notin [0, 1]$, then there does not exist any probability distribution $p$ having density with respect to $p_{\theta_{\mathrm{base}}}$ for which $\mathbb{E}_p[\boldsymbol{h}(\boldsymbol{x})] = \boldsymbol{h}^*$. And if $\boldsymbol{h}^*$ is either 0 or 1, then the solution to the maximum entropy problem is proportional to $p_{\theta_{\mathrm{base}}}(\boldsymbol{x})\mathbb{1}\{\boldsymbol{x} \leq 0\}$ or $p_{\theta_{\mathrm{base}}}(\boldsymbol{x})\mathbb{1}\{\boldsymbol{x} > 0\}$, respectively. Neither of these solutions is an exponential tilt of $p_{\theta_{\mathrm{base}}}$, Equation (5).

Our proof of the maximum entropy principle leverages classical convex duality (Rockafellar, 1970) by showing that (14) is a convex problem, defined on the infinite-dimensional space of all probability densities for which $\boldsymbol{h}$ has a finite moment. The corresponding *dual problem* is

$$\sup_{\boldsymbol{\alpha} \in \mathbb{R}^d} \boldsymbol{\alpha}^\top \boldsymbol{h}^* - A_P(\boldsymbol{\alpha}), \quad A_P(\boldsymbol{\alpha}) := \log \left( \mathbb{E}_P[\exp\{r_{\boldsymbol{\alpha}}(\boldsymbol{x})\}] \right), \quad r_{\boldsymbol{\alpha}}(\boldsymbol{x}) = \boldsymbol{\alpha}^\top \boldsymbol{h}(\boldsymbol{x}), \tag{15}$$

which is concave. $A_P : \mathbb{R}^d \to \mathbb{R} \cup \{+\infty\}$ is known as the *log-normalizer* or *cumulant generating function* corresponding to the exponential family

$$\exp\{r_{\boldsymbol{\alpha}}(\boldsymbol{x}) - A_P(\boldsymbol{x})\} P(d\boldsymbol{x}). \tag{16}$$

We will make the standard assumption that the domain of $A_P$ is open

**Assumption C.2** (Domain of log-normalizer). The subset $\Xi = \{\boldsymbol{\alpha} \in \mathbb{R}^d \mid A_P(\boldsymbol{\alpha}) < \infty\}$ is open.

Whenever $A_P$ is finite, (16) is a well-defined probability measure on $X$. $\Xi$ is known as the *natural parameter space* of the exponential family (16). When Assumption C.2 holds, the exponential family is said to be *regular*.

The log-normalizer $A_P$ possesses many nice properties: for instance, it is convex and infinitely differentiable on $\Xi$. Convexity can be seen by computing the Hessian of $A_P(\boldsymbol{\alpha})$

$$\nabla_{\boldsymbol{\alpha}}^2 A_P(\boldsymbol{\alpha}) = \frac{\mathbb{E}_P[(\boldsymbol{h}(\boldsymbol{x}) - \nabla_{\boldsymbol{\alpha}} A_P(\boldsymbol{\alpha}))(\boldsymbol{h}(\boldsymbol{x}) - \nabla_{\boldsymbol{\alpha}} A_P(\boldsymbol{\alpha}))^\top \exp\{r_{\boldsymbol{\alpha}}(\boldsymbol{x})\}]}{\mathbb{E}_P[\exp\{r_{\boldsymbol{\alpha}}(\boldsymbol{x})\}]} \tag{17}$$

and recognizing that it is positive semi-definite. Differentiability is addressed in the remark following Lemma C.11.

Now that we have introduced the dual of the maximum entropy problem, we are prepared to give a precise statement and proof of the maximum entropy principle

**Theorem C.3** (Kullback (1959)). *Suppose Assumptions C.1 and C.2 hold. Then there exists a probability measure $Q^* \in \mathcal{P}(P)$ with density $dQ^*/dP \propto \exp(r_{\boldsymbol{\alpha}^*}(\boldsymbol{x}))$. Moreover, $Q^*$ is the solution to the maximum entropy problem* (14) *and is unique up to $P$-null sets.*

Unlike the primal problem (14), the dual problem (15) is defined on finite-dimensional Euclidean space, which makes it simpler to analyze. We first argue by weak duality that the value of (14) is at least as large as (15). We then identify a vector $\boldsymbol{\alpha}^*$ and a distribution $Q^*$ for which the primal and dual objectives are equal. By weak duality, this implies that $Q^*$ is optimal for the primal problem.

*Proof of Theorem C.3.* We first rewrite the primal problem (14) in the form

$$\inf_q \psi(q) + g(\mathcal{A}q)$$

$$\psi(q) = \begin{cases} \int q(\boldsymbol{x}) \log(q(\boldsymbol{x})) P(d\boldsymbol{x}) & \text{if } q \geq 0 \\ +\infty & \text{else} \end{cases}, \quad g(\boldsymbol{y}_0, \boldsymbol{y}_1) = \begin{cases} 0 & \text{if } \boldsymbol{y}_0 = 1 \text{ and } \boldsymbol{y}_1 = \boldsymbol{h}^* \\ +\infty & \text{else} \end{cases},$$

$$\mathcal{A}(q) = \left( \int q(\boldsymbol{x}) P(d\boldsymbol{x}), \int \boldsymbol{h}(\boldsymbol{x}) q(\boldsymbol{x}) P(d\boldsymbol{x}) \right)$$

defined on the space of $X$-measurable functions $q$ for which $\mathbb{E}_P[|q(\boldsymbol{x})|] = \int |q(\boldsymbol{x})| P(d\boldsymbol{x}) < \infty$ and $\mathbb{E}_P[\|\boldsymbol{h}(\boldsymbol{x})\| |q(\boldsymbol{x})|] = \int \|\boldsymbol{h}(\boldsymbol{x})\| |q(\boldsymbol{x})| P(d\boldsymbol{x}) < \infty$. Here, $q$ represents the density of measure $Q$ with respect to $P$. $g(\mathcal{A}q)$ imposes the constraint that $Q$ is a probability measure and that the expectation of $\boldsymbol{h}$ under $Q$ is $\boldsymbol{h}^*$. And $\psi(q)$ is equal to the KL divergence between $Q$ and $P$.

Observe $\mathcal{A}$ is a bounded, linear map defined on this space. And $\psi$ and $g$ are convex. By Fenchel-Rockafellar duality (Borwein & Zhu, 2005, Theorem 4.4.2), weak duality holds for the maximum entropy problem and its dual (15).

Wainwright & Jordan (2008, Theorem 3.3) states that $\nabla_{\boldsymbol{\alpha}} A_P$ is a surjective mapping from $\Xi$ onto $\text{relint}(\mathcal{M})$. Hence, there exists $\boldsymbol{\alpha}^* \in \Xi$ for which $\nabla_{\boldsymbol{\alpha}} A_P(\boldsymbol{\alpha}^*) = \boldsymbol{h}^*$. The value of the dual at $\boldsymbol{\alpha}^*$ is

$$(\boldsymbol{\alpha}^*)^\top \boldsymbol{h}^* - A_P(\boldsymbol{\alpha}^*).$$

By differentiating the dual objective at $\boldsymbol{\alpha}^*$, we obtain,

$$0 = \nabla_{\boldsymbol{\alpha}}(\boldsymbol{\alpha}^\top \boldsymbol{h}^* - A_P(\boldsymbol{\alpha})) \implies \boldsymbol{h}^* = \frac{\int \boldsymbol{h}(\boldsymbol{x}) \exp\{r_{\boldsymbol{\alpha}^*}(\boldsymbol{x})\} P(d\boldsymbol{x})}{\int \exp\{r_{\boldsymbol{\alpha}^*}(\boldsymbol{x})\} P(d\boldsymbol{x})}.$$

In other words, the distribution $Q^* \in \mathcal{P}(P)$ defined such that $dQ^*/dP \propto \exp\{r_{\boldsymbol{\alpha}^*}(\boldsymbol{x})\}$ satisfies the moment constraint $\mathbb{E}_{Q^*}[\boldsymbol{h}(\boldsymbol{x})] = \boldsymbol{h}^*$. Moreover, the value of the primal objective at $Q^*$ is

$$\mathrm{D}_{\mathrm{KL}}(Q^* \parallel P) = (\boldsymbol{\alpha}^*)^\top \boldsymbol{h}^* - A_P(\boldsymbol{\alpha}^*),$$

which is equal to the value of the dual objective at $\boldsymbol{\alpha}^*$. By weak duality, we conclude $Q^*$ is the solution to the maximum entropy problem.

Uniqueness follows from the fact that the KL divergence $\psi$ is strictly convex. $\qquad\square$

## C.2. Estimating the Maximum Entropy Solution

Next, we discuss our estimator $\widehat{\boldsymbol{\alpha}}_N$ for the parameters $\boldsymbol{\alpha}^*$ of the maximum entropy solution. In particular, we provide verifiable conditions under which $\widehat{\boldsymbol{\alpha}}_N$ is well-defined, and we show that this estimator can be interpreted as the solution to a finite-sample version of the maximum entropy problem (4).

So far, the only assumptions we have made on the maximum entropy problem (14) are the relative interior condition on $\boldsymbol{h}^*$ (Assumption C.1) and the openness condition for the domain of $A_P$ (Assumption C.2). As we demonstrated in Appendix C.1, these conditions ensure that the solution to the maximum entropy problem exists and is unique. However, the solution to the dual problem need not be unique. Suppose, for example, that $\boldsymbol{h}$ is $d$-dimensional but has two identical components $\boldsymbol{h}(\boldsymbol{x})[i] = \boldsymbol{h}(\boldsymbol{x})[j]$. Then if $\boldsymbol{\alpha}^*$ is optimal for the dual problem, so is $\boldsymbol{\alpha}^* - te[i] + te[j]$ for all $t \in \mathbb{R}$, where

$e[i]$ and $e[j]$ denote the $i$ and $j$th standard basis vectors, respectively. Specifically, the set of optima for the dual problem is a hyperplane in $\mathbb{R}^d$. In order to estimate $\boldsymbol{\alpha}^*$, we want to ensure that the dual problem (15) also has a unique maximum.

As suggested by our example, in order to ensure that the dual optimum is unique, it suffices to eliminate linear redundancies among the statistics $\boldsymbol{h}(\boldsymbol{x})$.

**Assumption C.4** (Uniqueness of dual optimum). No linear combination of the components of $\boldsymbol{h}(\boldsymbol{x})$ is equal to a constant with $P$ probability one.

If Assumption C.4 holds, then the exponential family (16) is said to be *minimal*. An exponential family for which Assumption C.2 holds is minimal if and only if the log-normalizer $A_P(\boldsymbol{\alpha})$ is strictly convex on $\Xi$ (Wainwright & Jordan, 2008, Proposition 3.1). Once we have imposed Assumption C.4, Assumption C.1 is equivalent to $\boldsymbol{h}^* \in \text{int}(\mathcal{M})$, since $\text{relint}(\mathcal{M}) = \text{int}(\mathcal{M})$ as $\mathcal{M}$ is full-dimensional.

For non-trivial generative models, solving the dual problem (15) for $P = p_{\theta_{\text{base}}}$ is intractable since $A_{p_{\theta_{\text{base}}}}(\boldsymbol{\alpha})$ cannot be computed in closed-form. The estimator $\widehat{\boldsymbol{\alpha}}_N$ that we propose in (7) involves first drawing $N$ independent samples $\{\boldsymbol{x}_n\}_{n=1}^N$ from the base model $p_{\theta_{\text{base}}}$ and then solving the dual problem with the integral replaced by the empirical average from our samples. This is equivalent to solving the dual problem for $P$ equal to the empirical distribution of our samples $\frac{1}{N}\sum_{n=1}^N \delta_{\boldsymbol{x}_n}$, where $\delta_{\boldsymbol{x}}$ is the delta function at $\boldsymbol{x}$.

However, in order for $\widehat{\boldsymbol{\alpha}}_N$ to be well-defined, the interior point condition and uniqueness of the dual optimum must hold for the maximum entropy problem with $P = \frac{1}{N}\sum_{n=1}^N \delta_{\boldsymbol{x}_n}$. For this problem, these two conditions are straightforward to verify: (i) $\boldsymbol{h}^*$ lies in the interior of the convex hull of $\{\boldsymbol{h}(\boldsymbol{x}_n)\}_{n=1}^N$ and (ii) the empirical covariance matrix of $\{\boldsymbol{h}(\boldsymbol{x}_n)\}_{n=1}^N$ has full rank. For the example we provided at the beginning of the section, conditions (i) and (ii) are satisfied if and only if $\{\boldsymbol{h}(\boldsymbol{x}_n)\} = \{0, 1\}$ and $\boldsymbol{h}^* \in (0, 1)$.

It is possible for Assumptions C.1 and C.4 to hold for $p_{\theta_{\text{base}}}$ but not for $\frac{1}{N}\sum_{n=1}^N \delta_{\boldsymbol{x}_n}$. For our example, if $\{\boldsymbol{h}(\boldsymbol{x}_n)\} = \{0\}$ and $\boldsymbol{h}^* = 0$ (or $\{\boldsymbol{h}(\boldsymbol{x}_n)\} = \{1\}$ and $\boldsymbol{h}^* = 1$), then the maximum entropy solution exists and is equal to $Q^* = \frac{1}{N}\sum_{n=1}^N \delta_{\boldsymbol{x}_n}$, but every vector $\boldsymbol{\alpha} \in \mathbb{R}$ is optimal for the dual problem (15). We demonstrate in Appendix C.4 that the probability of this event approaches zero as the number of samples $N$ approaches infinity. However, we observe (e.g., Figure 3B) that when the base model $p_{\theta_{\text{base}}}$ lies far from the maximum entropy solution $p_{\boldsymbol{\alpha}^*}$, estimating $\boldsymbol{\alpha}^*$ with small variance requires many samples, and may even be computationally intractable.

### C.3. Connection Between the Relax and Reward Losses

In this section, we elucidate the connection between the relax and reward losses. We first introduce a problem corresponding to the relax loss that, similar to the maximum entropy problem (4), is defined on the space $\mathcal{P}(p_{\theta_{\text{base}}})$ of probability distributions that have a density with respect to $p_{\theta_{\text{base}}}$. When the generative model class $p_\theta$ is sufficiently expressive, the solution to this problem well approximates the minimizer of the relax loss. We then show that, under conditions, the solution to this related problem approaches the solution to the maximum entropy problem as $\lambda \to 0$. This confirms our intuition that when $\lambda \to 0$, minimizing the relax loss is equivalent to solving the calibration problem.

As in Appendix C.1, we let $X := (X, \mathcal{X})$ be a measurable space, $P$ be a probability measure defined on $X$, and $\boldsymbol{h} : X \to \mathbb{R}^d$ be a $X$-measurable function, and $\boldsymbol{h}^*$ be a target moment. We consider the problem

$$\inf_{Q \in \mathcal{P}(P)} \|\mathbb{E}_Q[\boldsymbol{h}] - \boldsymbol{h}^*\|^2 + \lambda D_{\text{KL}}(Q \parallel P), \text{ s.t. } \mathbb{E}_Q[\|\boldsymbol{h}\|] < \infty \tag{18}$$

In convex analysis (e.g., Hestenes, 1969; Powell, 1969; Boyd & Vandenberghe, 2004; Ben-Tal & Nemirovski, 2023), (18) is known as a penalty problem.

When $P = p_{\theta_{\text{base}}}$, then (18) agrees with the problem of minimizing the relax loss (2), except the domain of the problem is $\mathcal{P}(p_{\theta_{\text{base}}})$ rather than the class of generative models $p_\theta$. Suppose momentarily that the infimum of (18), denoted by $Q_\lambda$, is attained. The minimizer of the relax loss (2) will not in general be equal to $Q_\lambda$ since $Q_\lambda$ does not lie in the class of generative models. However, as we argued when we proposed the reward loss, we would expect $Q_\lambda$ and the minimizer of the relax loss to be close in KL distance when the class of generative models $p_\theta$ is sufficiently expressive.

Introducing the problem (18) is helpful insofar as, similar to the maximum entropy problem, we can obtain a closed-form expression for the solution $Q_\lambda$.

**Proposition C.5.** *Suppose Assumption C.2 holds. Then there exists a unique solution $\boldsymbol{\alpha}_\lambda$ to the fixed point equation*

$$\boldsymbol{\alpha} = -\frac{2}{\lambda}(\nabla_{\boldsymbol{\alpha}} A_P(\boldsymbol{\alpha}) - \boldsymbol{h}^*), \quad \boldsymbol{\alpha} \in \Xi.$$

*Moreover, $Q_\lambda$ defined by $dQ_\lambda/dP \propto \exp\{\boldsymbol{\alpha}_\lambda^\top \boldsymbol{h}(\boldsymbol{x})\}$ is the unique solution to* (18).

Our proof mirrors that for the maximum entropy principle (Theorem C.3). Namely, we invoke Fenchel-Rockafellar duality (Rockafellar, 1970) to relate the convex problem (18), defined on the space of probability densities with respect to $P$ with finite $\boldsymbol{h}$ moment, to its concave dual problem

$$\sup_{\boldsymbol{\alpha} \in \mathbb{R}^P} F_\lambda(\boldsymbol{\alpha}), \quad F_\lambda(\boldsymbol{\alpha}) = \lambda\left(-\frac{\lambda}{4}\|\boldsymbol{\alpha}\|^2 - A_P(\boldsymbol{\alpha}) + \boldsymbol{\alpha}^\top \boldsymbol{h}^*\right) \tag{19}$$

defined on Euclidean space. We then show that $\boldsymbol{\alpha}_\lambda$ is the unique solution to the dual problem, and we use this solution to construct a solution to the primal problem. Interestingly, $\boldsymbol{\alpha}_\lambda$ is the unique solution to the dual problem even when there is redundancy among the constraints $\boldsymbol{h}$ (i.e., Assumption C.4 does not hold).

*Proof of Proposition C.5.* We rewrite the primal problem (18) in the form

$$\inf_q \psi(q) + g(\mathcal{A}q),$$

$$\psi(q) = \begin{cases} \lambda \int q(\boldsymbol{x}) \log(q(\boldsymbol{x})) P(d\boldsymbol{x}) & \text{if } q \geq 0 \\ +\infty & \text{else} \end{cases}, \quad g(\boldsymbol{y}_0, \boldsymbol{y}_1) = \begin{cases} \|\boldsymbol{y}_1 - \boldsymbol{h}^*\|^2 & \text{if } \boldsymbol{y}_0 = 1 \\ +\infty & \text{else} \end{cases},$$

$$\mathcal{A}(q) = \left(\int q(\boldsymbol{x}) P(d\boldsymbol{x}), \int \boldsymbol{h}(\boldsymbol{x}) q(\boldsymbol{x}) P(d\boldsymbol{x})\right)$$

defined on the space of $X$-measurable functions $q$ for which $\mathbb{E}_P[|q(\boldsymbol{x})|] = \int |q(\boldsymbol{x})| P(d\boldsymbol{x}) < \infty$ and $\mathbb{E}_P[\|\boldsymbol{h}(\boldsymbol{x})\| |q(\boldsymbol{x})|] = \int \|\boldsymbol{h}(\boldsymbol{x})\| |q(\boldsymbol{x})| P(d\boldsymbol{x}) < \infty$. $q$ represents the density of measure $Q$ with respect to $P$. $g(\mathcal{A}q)$ is equal to $\|\mathbb{E}_Q[\boldsymbol{h}(\boldsymbol{x})] - \boldsymbol{h}^*\|^2$ if $Q$ is a probability measure and is infinite otherwise. $\psi(q)$ is equal to the KL divergence between $Q$ and $P$, scaled by $\lambda$.

As in the proof of Theorem C.3, $\mathcal{A}$ is a bounded, linear map defined on this space, and $\psi$ and $g$ are convex. By Fenchel-Rockafellar duality (Borwein & Zhu, 2005, Theorem 4.4.2), weak duality holds for the problem (18) and its dual (19).

By the remark following Lemma C.11, $F_\lambda$ is infinitely differentiable on $\Xi$, and taking two derivatives of $F_\lambda(\boldsymbol{\alpha})$ yields

$$\nabla_{\boldsymbol{\alpha}} F_\lambda(\boldsymbol{\alpha}) = \lambda\left(-\frac{\lambda}{2}\boldsymbol{\alpha} - \nabla_{\boldsymbol{\alpha}} A_P(\boldsymbol{\alpha}) + \boldsymbol{h}^*\right), \quad \nabla_{\boldsymbol{\alpha}}^2 F_\lambda(\boldsymbol{\alpha}) = \lambda\left(-\frac{\lambda}{2}\mathbb{I} - \nabla_{\boldsymbol{\alpha}}^2 A_P(\boldsymbol{\alpha})\right).$$

Since $\nabla_{\boldsymbol{\alpha}}^2 A_P(\boldsymbol{\alpha})$ is positive semi-definite, then the problem (19) is strongly concave. And by our assumption that $\Xi$ is open, $F_\lambda(\boldsymbol{\alpha})$ is equal to $-\infty$ for $\boldsymbol{\alpha}$ belonging to the boundary of $\Xi$. Together with strong concavity, this implies a unique maximizer $\boldsymbol{\alpha}_\lambda$ of $F_\lambda$ exists.

In particular, $\boldsymbol{\alpha}_\lambda$ is the unique $\boldsymbol{\alpha} \in \mathbb{R}^d$ that satisfies the fixed-point equation

$$\nabla_{\boldsymbol{\alpha}} F_\lambda(\boldsymbol{\alpha}) = \lambda\left(-\frac{\lambda}{2}\boldsymbol{\alpha} - \nabla_{\boldsymbol{\alpha}} A(\boldsymbol{\alpha}) + \boldsymbol{h}^*\right) = \boldsymbol{0} \implies \boldsymbol{\alpha} = -\frac{2}{\lambda}(\nabla_{\boldsymbol{\alpha}} A_P(\boldsymbol{\alpha}) - \boldsymbol{h}^*).$$

And the probability measure $Q_{\boldsymbol{\alpha}_\lambda} \propto \exp\{r_{\boldsymbol{\alpha}_\lambda}(\boldsymbol{x})\} P(d\boldsymbol{x})$ satisfies

$$\lambda \mathrm{D}_{\mathrm{KL}}\left(Q_{\boldsymbol{\alpha}_\lambda} \,\|\, P\right) + \|\mathbb{E}_{Q_{\boldsymbol{\alpha}_\lambda}}[\boldsymbol{h}(\boldsymbol{x})] - \boldsymbol{h}^*\|$$

$$= \lambda(\boldsymbol{\alpha}_\lambda^\top \nabla_{\boldsymbol{\alpha}} A_P(\boldsymbol{\alpha}_\lambda) - A_P(\boldsymbol{\alpha}_\lambda)) + \frac{\lambda^2}{4}\|\boldsymbol{\alpha}_\lambda\|^2$$

$$= \lambda\left(\boldsymbol{\alpha}_\lambda^\top\left(\boldsymbol{h}^* - \frac{\lambda}{2}\boldsymbol{\alpha}_\lambda\right) - A_P(\boldsymbol{\alpha}_\lambda)\right) + \frac{\lambda^2}{4}\|\boldsymbol{\alpha}_\lambda\|^2$$

$$= F_\lambda(\boldsymbol{\alpha}_\lambda).$$

By weak duality, this implies $Q_\lambda := Q_{\boldsymbol{\alpha}_\lambda}$ is optimal for the primal problem. Moreover, strict convexity of $\psi$ implies that the optimum of the primal problem is unique. $\square$

Next, we show that as the regularization parameter $\lambda \to 0$, then $Q_\lambda$ achieves the minimum possible Euclidean norm constraint violation i.e., Euclidean norm difference between $\mathbb{E}_{Q_\lambda}[\boldsymbol{h}]$ and $\boldsymbol{h}^*$. We also give a finite $\lambda$ bound on the constraint violation.

**Proposition C.6.** *The distribution $Q_\lambda$ satisfies*

$$\lim_{\lambda \to 0} \|\mathbb{E}_{Q_\lambda}[\boldsymbol{h}(\boldsymbol{x})] - \boldsymbol{h}^*\| = \inf_{\substack{Q \in \mathcal{P}(P) \\ D_{KL}(Q \,\|\, P) < \infty}} \|\mathbb{E}_Q[\boldsymbol{h}(\boldsymbol{x})] - \boldsymbol{h}^*\|.$$

*Moreover, we have the finite-sample bound on the Euclidean norm constraint violation of $Q_\lambda$*

$$\|\mathbb{E}_{Q_\lambda}[\boldsymbol{h}(\boldsymbol{x})] - \boldsymbol{h}^*\| \leq \inf_{Q \in \mathcal{P}(P)} \left\{ \sqrt{\lambda D_{KL}(Q \,\|\, P)} + \|\mathbb{E}_Q[\boldsymbol{h}(\boldsymbol{x})] - \boldsymbol{h}^*\| \right\}.$$

*Proof.* Fix $\varepsilon > 0$ and let $Q_\varepsilon$ be such that $\|\mathbb{E}_{Q_\varepsilon}[\boldsymbol{h}(\boldsymbol{x})] - \boldsymbol{h}^*\| \leq \inf_{Q \in \mathcal{P}(P)} \|\mathbb{E}_Q[\boldsymbol{h}(\boldsymbol{x})] - \boldsymbol{h}^*\| + \varepsilon$. Then by the optimality of $Q_\lambda$ for the objective (18),

$$\begin{aligned}
\|\mathbb{E}_{Q_\lambda}[\boldsymbol{h}(\boldsymbol{x})] - \boldsymbol{h}^*\|^2 &\leq \lambda D_{\mathrm{KL}}(Q_\lambda \,\|\, P) + \|\mathbb{E}_{Q_\lambda}[\boldsymbol{h}(\boldsymbol{x})] - \boldsymbol{h}^*\|^2 \\
&\leq \lambda D_{\mathrm{KL}}(Q_\varepsilon \,\|\, P) + \|\mathbb{E}_{Q_\varepsilon}[\boldsymbol{h}(\boldsymbol{x})] - \boldsymbol{h}^*\|^2.
\end{aligned} \tag{20}$$

Our choice of $Q_\varepsilon$ yields

$$\|\mathbb{E}_{Q_\lambda}[\boldsymbol{h}(\boldsymbol{x})] - \boldsymbol{h}^*\|^2 \leq \lambda D_{\mathrm{KL}}(Q_\varepsilon \,\|\, P) + \inf_{Q \in \mathcal{P}(P)} \|\mathbb{E}_Q[\boldsymbol{h}(\boldsymbol{x})] - \boldsymbol{h}^*\| + \varepsilon.$$

Taking $\lambda \to 0$ and then $\varepsilon \to 0$ yields the first result. Replacing $Q_\varepsilon$ with $Q \in \mathcal{P}(P)$ in (20) and taking the infimum over $Q$ yields the second result. $\qquad \square$

In the setting of Proposition C.9 where a solution to the maximum entropy problem exists, then a bound on the Euclidean norm constraint violation of $Q_\lambda$ is simply $\sqrt{\lambda D_{\mathrm{KL}}(Q^* \,\|\, P)}$. This implies that $\|\mathbb{E}_{Q_\lambda}[\boldsymbol{h}(\boldsymbol{x})] - \boldsymbol{h}^*\| = \mathcal{O}(\sqrt{\lambda})$.

When Assumptions C.1 and C.4 hold, we can obtain a faster rate of convergence of $\mathbb{E}_{Q_\lambda}[\boldsymbol{h}(\boldsymbol{x})]$ to $\boldsymbol{h}^*$, and we can show that $\boldsymbol{\alpha}_\lambda$ converges to the parameters $\boldsymbol{\alpha}^*$ of the maximum entropy distribution.

This result can seem counterintuitive, since as $\lambda \to 0$, the objective function of the relax problem Equation (18) approaches $\|\mathbb{E}_Q[\boldsymbol{h}] - \boldsymbol{h}^*\|^2$. This is minimized whenever $\mathbb{E}_Q[\boldsymbol{h}] = \boldsymbol{h}^*$ i.e., there are infinitely many minimizers. The key insight is that the optimization problem does *not* approach the degenerate problem

$$\inf_{Q \in \mathcal{P}(P)} \|\mathbb{E}_Q[\boldsymbol{h}] - \boldsymbol{h}^*\|^2 \text{ s.t. } \mathbb{E}_Q[\|\boldsymbol{h}\|] < \infty.$$

Rather, it approaches

$$\inf_{Q \in \mathcal{P}(P)} D_{\mathrm{KL}}(Q \,\|\, P) \text{ s.t. } \mathbb{E}_Q[\boldsymbol{h}] = \boldsymbol{h}^*, \ \mathbb{E}_Q[\|\boldsymbol{h}\|] < \infty,$$

which is the maximum entropy problem (14).

**Proposition C.7.** *Suppose Assumptions C.1, C.2, and C.4 hold, which imply that the maximum entropy solution $dQ^*/dP \propto \exp\{r_{\boldsymbol{\alpha}^*}(\boldsymbol{x})\}$ exists. Then $\boldsymbol{\alpha}_\lambda \to \boldsymbol{\alpha}^*$ as $\lambda \to 0$. In particular,*

*(i)* $\|\boldsymbol{\alpha}_\lambda - \boldsymbol{\alpha}^*\| = \mathcal{O}(\lambda)$

*(ii)* $\|\mathbb{E}_{Q_\lambda}[\boldsymbol{h}(\boldsymbol{x})] - \boldsymbol{h}^*\| = \mathcal{O}(\lambda)$

*(iii)* $|D_{KL}(Q_\lambda \,\|\, P) - D_{KL}(Q^* \,\|\, P)| = \mathcal{O}(\lambda)$.

*Proof.* Prior to proving (i)-(iii), we first establish $\|\boldsymbol{\alpha}_\lambda - \boldsymbol{\alpha}^*\| = o(1)$. From the proof of Proposition C.5, we know that $\boldsymbol{\alpha}_\lambda$ maximizes $\lambda^{-1} F_\lambda(\boldsymbol{\alpha}) = -\frac{\lambda}{4}\|\boldsymbol{\alpha}\|^2 - A_P(\boldsymbol{\alpha}) + \boldsymbol{\alpha}^\top \boldsymbol{h}^*$ for each $\lambda > 0$. And from (15), we know that $\boldsymbol{\alpha}^*$ maximizes $F_0(\boldsymbol{\alpha}) = -A_P(\boldsymbol{\alpha}) + \boldsymbol{\alpha}^\top \boldsymbol{h}^*$. Clearly, $F_\lambda(\boldsymbol{\alpha}) \to F_0(\boldsymbol{\alpha})$ pointwise as $\lambda \to 0$. Since each of $F_\lambda$ and $F_0$ is concave on $\Xi$, a classical result in convex analysis Rockafellar (1970, Theorem, 10.8) implies that the convergence $F_\lambda(\boldsymbol{\alpha}) \to F_0(\boldsymbol{\alpha})$ is uniform on closed, bounded subsets of $\Xi$ containing $\boldsymbol{\alpha}^*$.

Fix $\epsilon > 0$ such that the Euclidean ball of radius $\epsilon$ centered at $\boldsymbol{\alpha}^*$ is contained in $\Xi$. By Assumption C.4 $\nabla_{\boldsymbol{\alpha}}^2 A_P(\boldsymbol{\alpha})$ positive definite for every $\boldsymbol{\alpha} \in \Xi$, which implies $F_0$ is strictly concave. Hence, there exists a $\kappa$ such that for all $\|\boldsymbol{\alpha} - \boldsymbol{\alpha}^*\| = \epsilon$,

$$F_0(\boldsymbol{\alpha}) < \kappa < F_0(\boldsymbol{\alpha}^*).$$

This is because the left-hand side of the above inequality attains its maximum on the compact set $\|\boldsymbol{\alpha} - \boldsymbol{\alpha}^*\| = \epsilon$ and (ii) by strict concavity this maximum must be strictly less than the right-hand side. Moreover, by uniform convergence of $F_\lambda$ to $F_0$, there exists $\lambda_\epsilon > 0$ such that for all $\lambda < \lambda_\epsilon$ and all $\|\boldsymbol{\alpha} - \boldsymbol{\alpha}^*\| = \epsilon$

$$F_\lambda(\boldsymbol{\alpha}) < \kappa < F_\lambda(\boldsymbol{\alpha}^*). \tag{21}$$

Since $F_\lambda$ is also concave, (21) implies that the maximizer of $F_\lambda$, $\boldsymbol{\alpha}_\lambda$, must lie in the Euclidean ball of radius $\epsilon$ centered at $\boldsymbol{\alpha}^*$. This establishes $\|\boldsymbol{\alpha}_\lambda - \boldsymbol{\alpha}^*\| = o(1)$.

We are now prepared to prove (i). By Taylor expanding $\nabla_{\boldsymbol{\alpha}} A_P(\boldsymbol{\alpha})$ at $\boldsymbol{\alpha}_\lambda$ about $\boldsymbol{\alpha}^*$, we obtain

$$\nabla_{\boldsymbol{\alpha}} A_P(\boldsymbol{\alpha}_\lambda) = \boldsymbol{h}^* + \nabla_{\boldsymbol{\alpha}}^2 A_P(\boldsymbol{\alpha}^*)(\boldsymbol{\alpha}_\lambda - \boldsymbol{\alpha}^*) + \boldsymbol{r}_\lambda, \quad \|\boldsymbol{r}_\lambda\| = o(\|\boldsymbol{\alpha}_\lambda - \boldsymbol{\alpha}^*\|). \tag{22}$$

By Proposition C.5, $\boldsymbol{\alpha}_\lambda$ satisfies $\boldsymbol{\alpha}_\lambda = -\frac{2}{\lambda}(\nabla_{\boldsymbol{\alpha}} A(\boldsymbol{\alpha}_\lambda) - \boldsymbol{h}^*)$. Multiplying (22) by $-2/\lambda$ and substituting in this expression for $\boldsymbol{\alpha}_\lambda$ yields

$$\boldsymbol{\alpha}_\lambda = -\frac{2}{\lambda}\nabla_{\boldsymbol{\alpha}}^2 A_P(\boldsymbol{\alpha}^*)(\boldsymbol{\alpha}_\lambda - \boldsymbol{\alpha}^*) + \frac{1}{\lambda}\boldsymbol{r}_\lambda.$$

Solving for $\boldsymbol{\alpha}_\lambda - \boldsymbol{\alpha}^*$ yields

$$\boldsymbol{\alpha}_\lambda - \boldsymbol{\alpha}^* = -\left(\mathbb{I} + \frac{2}{\lambda}\nabla_{\boldsymbol{\alpha}}^2 A_P(\boldsymbol{\alpha}^*)\right)^{-1}\left(\boldsymbol{\alpha}^* + \frac{1}{\lambda}\boldsymbol{r}_\lambda\right)$$
$$= -\lambda\left(\lambda\mathbb{I} + 2\nabla_{\boldsymbol{\alpha}}^2 A_P(\boldsymbol{\alpha}^*)\right)^{-1}\boldsymbol{\alpha}^* + \tilde{\boldsymbol{r}}_\lambda \tag{23}$$

for $\tilde{\boldsymbol{r}}_\lambda = o(\|\boldsymbol{\alpha}_\lambda - \boldsymbol{\alpha}^*\|)$. And because $\|\boldsymbol{\alpha}_\lambda - \boldsymbol{\alpha}^*\| = o(1)$, then for all $\lambda$ sufficiently small, $\|\tilde{\boldsymbol{r}}_\lambda\| \leq \frac{1}{2}\|\boldsymbol{\alpha}_\lambda - \boldsymbol{\alpha}^*\|$. Taking the norm of both sides of (23) and rearranging yields

$$\|\boldsymbol{\alpha}_\lambda - \boldsymbol{\alpha}^*\| \leq 2\lambda\|\left(\lambda\mathbb{I} + 2\nabla_{\boldsymbol{\alpha}}^2 A_P(\boldsymbol{\alpha}^*)\right)^{-1}\boldsymbol{\alpha}^*\|$$

for all $\lambda$ sufficiently small. This proves (i).

For (ii), the relationship $\boldsymbol{\alpha}_\lambda = -\frac{2}{\lambda}(\nabla_{\boldsymbol{\alpha}} A(\boldsymbol{\alpha}_\lambda) - \boldsymbol{h}^*)$ yields

$$\|\mathbb{E}_{Q_\lambda}[\boldsymbol{h}(\boldsymbol{x})] - \boldsymbol{h}^*\| = \|\nabla_{\boldsymbol{\alpha}} A(\boldsymbol{\alpha}_\lambda) - \boldsymbol{h}^*\| \leq \frac{\lambda}{2}\|\boldsymbol{\alpha}_\lambda - \boldsymbol{\alpha}^*\| + \frac{\lambda}{2}\|\boldsymbol{\alpha}^*\| = \mathcal{O}(\lambda).$$

Lastly for (iii),

$$\begin{aligned}
D_{\text{KL}}(Q_\lambda \| P) &= \boldsymbol{\alpha}_\lambda^\top \mathbb{E}_{Q_\lambda}[\boldsymbol{h}(\boldsymbol{x})] - A_P(\boldsymbol{\alpha}_\lambda) \\
&= (\boldsymbol{\alpha}_\lambda^\top \boldsymbol{h}^* + \mathcal{O}(\lambda)) - \{A_P(\boldsymbol{\alpha}^*) + \nabla_{\boldsymbol{\alpha}} A_P(\boldsymbol{\alpha}^*)^\top(\boldsymbol{\alpha}_\lambda - \boldsymbol{\alpha}^*) + o(\|\boldsymbol{\alpha}_\lambda - \boldsymbol{\alpha}^*\|)\} \\
&= \boldsymbol{\alpha}_\lambda^\top \boldsymbol{h}^* - A_P(\boldsymbol{\alpha}^*) + \mathcal{O}(\lambda) \\
&= D_{\text{KL}}(Q^* \| P) + \mathcal{O}(\lambda).
\end{aligned}$$

$\square$

The convergence rate $\|\mathbb{E}_{Q_\lambda}[\boldsymbol{h}(\boldsymbol{x})] - \boldsymbol{h}^*\| = \mathcal{O}(\lambda)$ is standard for penalty methods (see e.g., Hestenes, 1969). In our argument, we additionally show that up to first order,

$$\|\mathbb{E}_{Q_\lambda}[\boldsymbol{h}(\boldsymbol{x})] - \boldsymbol{h}^*\| \leq c\lambda\{1 + \|\left(\lambda\mathbb{I} + 2\nabla_{\boldsymbol{\alpha}}^2 A_P(\boldsymbol{\alpha}^*)\right)^{-1}\|\}, \quad c \geq 0.$$

Hence, for finite $0 < \lambda \ll 1$, we would expect $\|\mathbb{E}_{Q_\lambda}[\boldsymbol{h}(\boldsymbol{x})] - \boldsymbol{h}^*\|$ to be small whenever the Fisher information matrix of the exponential family $\exp\{r_{\boldsymbol{\alpha}}(\boldsymbol{x}) - A_P(\boldsymbol{x})\}P(d\boldsymbol{x})$ at $\boldsymbol{\alpha} = \boldsymbol{\alpha}^*$ has a large minimum eigenvalue.

In Section 2.2 we derived the reward loss as the KL divergence of the model $p_\theta$ to the maximum entropy solution $p_{\alpha^*}$. The relax loss can also be viewed as a divergence to a tilt of the base model $p_{\theta_{\text{base}}}$, except that the tilt depends on the current model $p_\theta$. In particular, the stationary points of the relaxed loss are exactly the stationary points of the objective

$$D_{\text{KL}}\left(p_\theta \parallel p_{\theta_{\text{base}}}\right) + \frac{2}{\lambda}(\mathbb{E}_{p_{\text{sg}(\theta)}}[\boldsymbol{h}(\boldsymbol{x})] - \boldsymbol{h}^*)^\top \mathbb{E}_{p_\theta}[\boldsymbol{h}(\boldsymbol{x})]. \tag{24}$$

This can be seen by taking the gradient of (24). By identifying $\boldsymbol{\alpha} = -\frac{2}{\lambda}(\mathbb{E}_{p_{\text{sg}(\theta)}}[\boldsymbol{h}(\boldsymbol{x})] - \boldsymbol{h}^*)$ and $q_{\boldsymbol{\alpha}} \propto p_{\theta_{\text{base}}}(\boldsymbol{x}) \exp\{r_{\boldsymbol{\alpha}}(\boldsymbol{x})\}$, we observe that (24) is exactly equal to $D_{\text{KL}}\left(p_\theta \parallel q_{\boldsymbol{\alpha}}\right)$. $q_{\boldsymbol{\alpha}}$ *can be understood as our current best approximation to the solution of* (18). Unlike the solution of (18), though, $\mathbb{E}_{q_{\boldsymbol{\alpha}}}[\boldsymbol{h}(\boldsymbol{x})]$ is not equal to $\mathbb{E}_{p_{\text{sg}(\theta)}}[\boldsymbol{h}(\boldsymbol{x})]$. For sufficiently expressive class of generative models $p_\theta$, we would expect $\mathbb{E}_{q_{\boldsymbol{\alpha}}}[\boldsymbol{h}(\boldsymbol{x})]$ and $\mathbb{E}_{p_{\text{sg}(\theta)}}[\boldsymbol{h}(\boldsymbol{x})]$ to be approximately equal at the optimum.

### C.4. Consistency and Asymptotic Normality

In this section, we discuss the large sample behavior of the estimator $\widehat{\boldsymbol{\alpha}}_N$ for the parameters $\boldsymbol{\alpha}^*$ of the reward loss. Under Assumptions C.1, C.2, and C.4, we show that as $N \to \infty$ and $d$ remains fixed, then $\widehat{\boldsymbol{\alpha}}_N$ is close to $\boldsymbol{\alpha}^*$ with high probability. And under stronger conditions, we demonstrate that $\widehat{\boldsymbol{\alpha}}_N$ has a limiting normal distribution. The asymptotics of $\widehat{\boldsymbol{\alpha}}_N$ have previously been studied in the subject of empirical likelihood (Qin & Lawless, 1994; Kitamura & Stutzer, 1997; Owen, 2001).

We first aim to establish that $\widehat{\boldsymbol{\alpha}}_N$ is close to $\boldsymbol{\alpha}^*$ with high probability as $N \to \infty$ i.e., $\widehat{\boldsymbol{\alpha}}_N$ is *consistent* for $\boldsymbol{\alpha}^*$. Define the functions

$$A(\boldsymbol{\alpha}) := A_{p_{\theta_{\text{base}}}}(\boldsymbol{\alpha}), \quad A_N(\boldsymbol{\alpha}) := \log\left(\frac{1}{N}\sum_{n=1}^N \exp\{r_{\boldsymbol{\alpha}}(\boldsymbol{x}_n)\}\right),$$

where $A_P$ is defined in Appendix C.1. Observe that $A_N$ is random and depends on the independent samples $\{\boldsymbol{x}_n\}_{n=1}^N$ drawn from $p_{\theta_{\text{base}}}$. The dual problem corresponding to $p_{\theta_{\text{base}}}$ maximizes $\boldsymbol{\alpha}^\top \boldsymbol{h}(\boldsymbol{x}) - A(\boldsymbol{\alpha})$, whereas the dual problem corresponding to the distribution of samples $\{\boldsymbol{x}_n\}_{n=1}^N$ maximizes $\boldsymbol{\alpha}^\top \boldsymbol{h}(\boldsymbol{x}) - A_N(\boldsymbol{\alpha})$. By the Strong Law of Large Numbers (SLLN), for any $\boldsymbol{\alpha} \in \Xi$, $A_N(\boldsymbol{\alpha}) \to A(\boldsymbol{\alpha})$ with $p_{\theta_{\text{base}}}$ probability one. In order for our estimator $\widehat{\boldsymbol{\alpha}}_N$ to approach $\boldsymbol{\alpha}^*$, though, we need to argue that the dual objective corresponding to $\{\boldsymbol{x}_n\}_{n=1}^N$ *uniformly* approaches the dual objective corresponding to $p_{\theta_{\text{base}}}$ on some neighborhood containing $\boldsymbol{\alpha}^*$.

**Lemma C.8.** *For any closed, bounded subset $K$ of $\Xi$,*

$$\sup_{\boldsymbol{\alpha} \in K} |A_N(\boldsymbol{\alpha}) - A(\boldsymbol{\alpha})| \to 0$$

*with $p_{\theta_{\text{base}}}$ probability one.*

*Proof.* By the SLLN, we can construct a Borel set $\widetilde{N}$ of probability zero under $p_{\theta_{\text{base}}}$ such that on its complement $A_N(\boldsymbol{\alpha}) \to A(\boldsymbol{\alpha})$ holds for each $\boldsymbol{\alpha} \in \Xi \cap \mathbb{Q}^d$ (apply the SLLN for an individual $\boldsymbol{\alpha} \in \Xi \cap \mathbb{Q}^d$, then take a union over probability zero sets).

Rockafellar (1970, Theorem 10.8) states that if a sequence of finite convex functions defined on an open, convex set $C$ converges pointwise on a dense subset of $C$ to a limiting function, then the limiting function is convex on $C$, and the convergence is uniform on closed and bounded subsets of $C$. Applying this result to our setting, on the complement of $\widetilde{N}$

$$\sup_{\boldsymbol{\alpha} \in K} |A_N(\boldsymbol{\alpha}) - A(\boldsymbol{\alpha})| \to 0$$

for $K$ a closed and bounded subset of $\Xi$. $\qquad\square$

Once we have proven uniform convergence, our proof of consistency for $\widehat{\boldsymbol{\alpha}}_N$ is nearly identical to our proof that $\|\boldsymbol{\alpha}_\lambda - \boldsymbol{\alpha}^*\| = o(1)$ in Proposition C.7.

**Proposition C.9** (Consistency of $\widehat{\boldsymbol{\alpha}}_N$). *Suppose Assumptions C.1, C.2, and C.4 hold. For any $\epsilon > 0$,*

$$\mathbb{P}_{p_{\theta_{\text{base}}}}(\|\widehat{\boldsymbol{\alpha}}_N - \boldsymbol{\alpha}^*\| > \epsilon) \to 0 \quad \text{as } N \to \infty.$$

*Proof.* From Appendix C.1, we know that both $A$ and $A_N$ are convex functions. Moreover by Assumption C.4, $A$ is strictly convex.

From Lemma C.8, there exists a closed, bounded subset $K$ of containing $\boldsymbol{\alpha}^*$ on which $\sup_{\boldsymbol{\alpha} \in K} |A_N(\boldsymbol{\alpha}) - A(\boldsymbol{\alpha})| \to 0$ with $p_{\theta_{\text{base}}}$ probability one. Since $\Xi$ is open (Assumption C.2), $K$ can be chosen to have positive diameter. Fix $\epsilon > 0$ sufficiently small such that the Euclidean ball centered at $\boldsymbol{\alpha}^*$ of radius $\epsilon$ is contained in $K$. Just as in the proof of Proposition C.7, there exists some $\kappa \in \mathbb{R}$ such that for all $\|\boldsymbol{\alpha} - \boldsymbol{\alpha}^*\| = \epsilon$,

$$\boldsymbol{\alpha}^\top \boldsymbol{h}^* - A(\boldsymbol{\alpha}) < \kappa < (\boldsymbol{\alpha}^*)^\top \boldsymbol{h}^* - A(\boldsymbol{\alpha}^*).$$

Fix $\delta > 0$. By uniform convergence, there exists $N_{\epsilon,\delta} \in \mathbb{N}$ such that $\forall N \geq N_{\epsilon,\delta}$ and for all $\|\boldsymbol{\alpha} - \boldsymbol{\alpha}^*\| = \epsilon$,

$$\boldsymbol{\alpha}^\top \boldsymbol{h}^* - A_N(\boldsymbol{\alpha}) < \kappa < (\boldsymbol{\alpha}^*)^\top \boldsymbol{h}^* - A_N(\boldsymbol{\alpha}^*).$$

with probability at least $1 - \delta$ under $p_{\theta_{\text{base}}}$. And since the dual objective corresponding to $\{\boldsymbol{x}_n\}_{n=1}^N$ is concave, this implies that, on this event, its maximum occurs in the Euclidean ball of radius $\epsilon$.

In other words, we have proven that for every $\epsilon > 0, \delta > 0$, there exists $N_{\epsilon,\delta}$ such that for every $N \geq N_{\epsilon,\delta}$,

$$\mathbb{P}_{p_{\theta_{\text{base}}}} \left( \|\widehat{\boldsymbol{\alpha}}_N - \boldsymbol{\alpha}^*\| > \epsilon \right) \leq \delta.$$

$\square$

Next, we show that under stronger conditions on the problem, $\widehat{\boldsymbol{\alpha}}_N$ has a normal limiting distribution, and we derive its variance.

**Proposition C.10** (Asymptotic normality of $\widehat{\boldsymbol{\alpha}}_N$). *Suppose Assumptions C.1, C.2, and C.4 hold. Moreover, assume $2\boldsymbol{\alpha}^* \in \Xi$, for $\Xi$ defined in Appendix C.1. Then the estimator $\widehat{\boldsymbol{\alpha}}_N$ is asymptotically normal:*

$$\sqrt{N}(\widehat{\boldsymbol{\alpha}}_N - \boldsymbol{\alpha}^*) \xrightarrow{d} \mathcal{N}(\boldsymbol{0}, (\text{Var}_{p_{\boldsymbol{\alpha}^*}}[\boldsymbol{h}(\boldsymbol{x})])^{-1} \boldsymbol{\Sigma} (\text{Var}_{p_{\boldsymbol{\alpha}^*}}[\boldsymbol{h}(\boldsymbol{x})])^{-1}),$$

$$\boldsymbol{\Sigma} = \frac{\mathbb{E}_{p_{\theta_{\text{base}}}}[(\boldsymbol{h}(\boldsymbol{x}) - \boldsymbol{h}^*)(\boldsymbol{h}(\boldsymbol{x}) - \boldsymbol{h}^*)^\top \exp\{r_{2\boldsymbol{\alpha}^*}(\boldsymbol{x})\}]}{(\mathbb{E}_{p_{\theta_{\text{base}}}}[\exp\{r_{\boldsymbol{\alpha}^*}(\boldsymbol{x})\}])^2}.$$

Prior to stating the proof of Proposition C.10, we build some intuition by working out the asymptotic variance for the example we presented at the beginning of the section. Recall that the constraint function is $\boldsymbol{h}(\boldsymbol{x}) = \mathbb{1}\{\boldsymbol{x} > 0\}$, $\boldsymbol{h}^*$ is its target value, and $h_b = \mathbb{P}_{\theta_{\text{base}}}(\boldsymbol{x} > 0)$ is the expected value of $\boldsymbol{h}$ under $p_{\theta_{\text{base}}}$. By directly solving for $\widehat{\boldsymbol{\alpha}}_N$ in the expression (5) for the maximum entropy solution, we showed $\widehat{\boldsymbol{\alpha}}_N = \log(\frac{\boldsymbol{h}^*(1-\bar{\boldsymbol{y}}_N)}{(1-\boldsymbol{h}^*)\bar{\boldsymbol{y}}_N})$, where $\bar{\boldsymbol{y}}_N = \frac{1}{N}\sum_{n=1}^N \boldsymbol{y}_n$, $\boldsymbol{y}_n = \boldsymbol{h}(\boldsymbol{x}_n) \stackrel{d}{=} \text{Bernoulli}(h_b)$ for $\boldsymbol{x}_n \stackrel{i.i.d.}{\sim} p_{\theta_{\text{base}}}$. Next, we compute

$$\text{Var}_{p_{\boldsymbol{\alpha}^*}}[\boldsymbol{h}(\boldsymbol{x})] = \boldsymbol{h}^*(1 - \boldsymbol{h}^*)$$

$$\boldsymbol{\Sigma} = \frac{(\boldsymbol{h}^*)^2(1 - h_b) + (1 - \boldsymbol{h}^*)^2 \exp(2\boldsymbol{\alpha}^*)h_b}{(h_b \exp(\boldsymbol{\alpha}^*) + (1 - h_b))^2} = \frac{(\boldsymbol{h}^*)^2(1 - h_b) + \frac{(1-h_b)^2(\boldsymbol{h}^*)^2}{h_b}}{\left(\frac{1-h_b}{1-\boldsymbol{h}^*}\right)^2} = \frac{(\boldsymbol{h}^*)^2(1 - \boldsymbol{h}^*)^2}{h_b(1 - h_b)}.$$

Combining these two yields the asymptotic variance

$$\text{Var}_{p_{\boldsymbol{\alpha}^*}}[\boldsymbol{h}(\boldsymbol{x})]^{-2} \boldsymbol{\Sigma} = \frac{1}{h_b(1 - h_b)},$$

according to Proposition C.10. In other words, the estimator $\widehat{\boldsymbol{\alpha}}_N$ has greatest asymptotic variance when $h_b$ is close to either 0 or 1. Notice that we can compute the asymptotic variance of $\widehat{\boldsymbol{\alpha}}_N$ directly (i.e., without using Proposition C.10) by applying the delta method to $\bar{\boldsymbol{y}}_N$ and the function $z \mapsto \log(\frac{\boldsymbol{h}^*(1-z)}{(1-\boldsymbol{h}^*)z})$, in which case we obtain the same value.

The proof of Proposition C.10 relies on the technical result Lemma C.11, the statement and proof of which we defer to the end of the section.

*Proof of Proposition C.10.* Let $D_N$ be the set on which the strong duality holds for $P = \frac{1}{N}\sum_{n=1}^{N}\delta_{\boldsymbol{x}_n}$ and the dual optimum is uniquely achieved. From the proof of Proposition C.9, we can see $\mathbb{P}_{p_{\theta_{\text{base}}}}(D_N) \to 1$ as $N \to \infty$. Moreover, on the set $D_N$, $\widehat{\boldsymbol{\alpha}}_N$ is the unique root of

$$\frac{1}{N}\sum_{n=1}^{N}\psi(\boldsymbol{x}_n, \boldsymbol{\alpha}) = 0, \quad \psi(\boldsymbol{x}, \boldsymbol{\alpha}) := (\boldsymbol{h}(\boldsymbol{x}) - \boldsymbol{h}^*)\exp\{r_{\boldsymbol{\alpha}}(\boldsymbol{x})\}.$$

Also, from the proof of Proposition C.9, we know that Assumption C.4 implies $\text{Var}_{p_{\boldsymbol{\alpha}^*}}[\boldsymbol{h}(\boldsymbol{x})]$ is positive definite.

By Van der Vaart (2000, Theorem 5.21), if we can show $\psi(\boldsymbol{x}, \boldsymbol{\alpha})$ satisfies the Lipschitz condition

$$\|\psi(\boldsymbol{x}, \boldsymbol{\alpha}) - \psi(\boldsymbol{x}, \boldsymbol{\alpha}')\| \leq M(\boldsymbol{x})\|\boldsymbol{\alpha} - \boldsymbol{\alpha}'\| \tag{25}$$

for all $\boldsymbol{\alpha}, \boldsymbol{\alpha}'$ belonging to some neighborhood of $\boldsymbol{\alpha}^*$ and $\mathbb{E}_{p_{\theta_{\text{base}}}}[M(\boldsymbol{x})^2] < \infty$, then the previous facts imply that $\widehat{\boldsymbol{\alpha}}_N$ is asymptotically normal with variance

$$\underbrace{\mathbb{E}_{p_{\theta_{\text{base}}}}[(\boldsymbol{h}(\boldsymbol{x}) - \boldsymbol{h}^*)\boldsymbol{h}(\boldsymbol{x})^\top \exp\{r_{\boldsymbol{\alpha}^*}(\boldsymbol{x})\}]^{-1}(\mathbb{E}_{p_{\theta_{\text{base}}}}[\exp\{r_{\boldsymbol{\alpha}^*}(\boldsymbol{x})\}])}_{=\text{Var}_{p_{\boldsymbol{\alpha}^*}}[\boldsymbol{h}(\boldsymbol{x})]^{-1}}\boldsymbol{\Sigma}$$

$$\underbrace{(\mathbb{E}_{p_{\theta_{\text{base}}}}[\exp\{r_{\boldsymbol{\alpha}^*}(\boldsymbol{x})\}])(\mathbb{E}_{p_{\theta_{\text{base}}}}[(\boldsymbol{h}(\boldsymbol{x}) - \boldsymbol{h}^*)\boldsymbol{h}(\boldsymbol{x})^\top \exp\{r_{\boldsymbol{\alpha}^*}(\boldsymbol{x})\}]^{-1})^\top}_{\text{Var}_{p_{\boldsymbol{\alpha}^*}}[\boldsymbol{h}(\boldsymbol{x})]^{-1}}.$$

And so it remains only to establish the Lipschitz condition (25). First, we compute the derivative of $\psi$ with respect to $\boldsymbol{\alpha}$

$$\nabla_{\boldsymbol{\alpha}}\psi(\boldsymbol{x}, \boldsymbol{\alpha}) = (\boldsymbol{h}(\boldsymbol{x}) - \boldsymbol{h}^*)\boldsymbol{h}(\boldsymbol{x})^\top \exp\{r_{\boldsymbol{\alpha}}(\boldsymbol{x})\} = \nabla_{\boldsymbol{\alpha}}^2 \exp\{r_{\boldsymbol{\alpha}}(\boldsymbol{x})\} - \boldsymbol{h}^*(\nabla_{\boldsymbol{\alpha}}\exp\{r_{\boldsymbol{\alpha}}(\boldsymbol{x})\})^\top$$

Next, we appeal to Lemma C.11, which tells us that for all $\boldsymbol{\alpha}$ belonging to an open neighborhood of $\boldsymbol{\alpha}^*$, the derivatives of $\exp\{r_{\boldsymbol{\alpha}}(\boldsymbol{x})\}$ have norm dominated by a function $M(\boldsymbol{x})$ that is $p_{\theta_{\text{base}}}$-square integrable. Also, by the Mean Value Theorem, for all $\boldsymbol{\alpha}, \boldsymbol{\alpha}'$ belonging to this neighborhood,

$$\psi(\boldsymbol{x}, \boldsymbol{\alpha}) - \psi(\boldsymbol{x}, \boldsymbol{\alpha}') = \nabla\psi(\boldsymbol{x}, \tilde{\boldsymbol{\alpha}})(\boldsymbol{\alpha} - \boldsymbol{\alpha}')$$

for some $\tilde{\boldsymbol{\alpha}}$ on the line segment connecting $\boldsymbol{\alpha}$ to $\boldsymbol{\alpha}'$. By taking the norm on both sides and using $\|\nabla\psi(\boldsymbol{x}, \tilde{\boldsymbol{\alpha}})\| \leq M(\boldsymbol{x})$, we obtain the Lipschitz condition (25). □

**Lemma C.11.** *Under the assumptions of Proposition C.10, there exists an open neighborhood of $\boldsymbol{\alpha}^*$ on which all derivatives of $\exp\{r_{\boldsymbol{\alpha}}(\boldsymbol{x})\}$ with respect to $\boldsymbol{\alpha}$ are dominated by a $p_{\theta_{\text{base}}}$-square integrable function.*

*Proof.* In Proposition C.10, we assume $\mathbb{E}_{p_{\theta_{\text{base}}}}[\exp\{r_{2\boldsymbol{\alpha}^*}(\boldsymbol{x})\}] < \infty$; in other words, $2\boldsymbol{\alpha}^*$ is contained in the natural parameter space $\Xi$. Let $\varepsilon$ be defined such that the Euclidean ball of radius $\varepsilon$ centered at $2\boldsymbol{\alpha}^*$ is contained in $\Xi$. Fix any $\tilde{\boldsymbol{\alpha}}$ such that $\|\tilde{\boldsymbol{\alpha}} - \boldsymbol{\alpha}^*\| < \varepsilon/(2d)$, where $d$ is the dimension of the constraint $\boldsymbol{h}(\boldsymbol{x})$. Then by Cauchy-Schwarz

$$\exp\{\tilde{\boldsymbol{\alpha}}^\top\boldsymbol{h}(\boldsymbol{x})\} \leq \exp\{(\boldsymbol{\alpha}^*)^\top\boldsymbol{h}(\boldsymbol{x}) + \varepsilon/(2d)\|\boldsymbol{h}(\boldsymbol{x})\|\}. \tag{26}$$

Define the $2d$ vectors $(\boldsymbol{\beta}^{(\pm, l)})_{l=1}^d$ by $\boldsymbol{\beta}^{(+, l)} = \boldsymbol{e}[l]$, $\boldsymbol{\beta}^{(-, l)} = -\boldsymbol{e}[l]$, where $\boldsymbol{e}[l]$ denotes the $l$th standard basis vector. Then we can upper bound the second term using

$$\exp\{\|\boldsymbol{h}(\boldsymbol{x})\|\} \leq \prod_{l=1}^d \exp\{|\boldsymbol{h}_l(\boldsymbol{x})|\} \leq \prod_{l=1}^d (\exp\{\boldsymbol{h}_l(\boldsymbol{x})\} + \exp\{-\boldsymbol{h}_l(\boldsymbol{x})\}) \leq \sum_{l=1}^{2d} 2^d \exp\left\{d(\boldsymbol{\beta}^{(l)})^\top\boldsymbol{h}(\boldsymbol{x})\right\}.$$

Plugging this bound into (26) yields

$$\exp\{\tilde{\boldsymbol{\alpha}}^\top\boldsymbol{h}(\boldsymbol{x})\} \leq \sum_{l=1}^{2d} 2^d \exp\left\{(\boldsymbol{\alpha}^* + (\varepsilon/2)\boldsymbol{\beta}^{(l)})^\top\boldsymbol{h}(\boldsymbol{x})\right\}. \tag{27}$$

Squaring both sides of (27) yields

$$(\exp\{\tilde{\boldsymbol{\alpha}}^\top \boldsymbol{h}(\boldsymbol{x})\})^2 \leq 2^{2d} \sum_{l=1}^{2d} \sum_{k=1}^{2d} \exp\left\{(2\boldsymbol{\alpha}^* + (\varepsilon/2)(\boldsymbol{\beta}^{(l)} + \boldsymbol{\beta}^{(k)}))^\top \boldsymbol{h}(\boldsymbol{x})\right\}. \tag{28}$$

However, we notice $\|2\boldsymbol{\alpha}^* + (\varepsilon/2)(\boldsymbol{\beta}^{(l)} + \boldsymbol{\beta}^{(k)}) - 2\boldsymbol{\alpha}^*\| \leq \varepsilon$, so each term on the right-hand side of (28) has finite expectation under $p_{\theta_{\text{base}}}$. This implies $\exp\{r_{\boldsymbol{\alpha}}(\boldsymbol{x})\}$ is dominated by the right-hand side of (27), which is square integrable under $p_{\theta_{\text{base}}}$, for all $\|\boldsymbol{\alpha} - \boldsymbol{\alpha}^*\| < \varepsilon/(2d)$.

As for the derivatives of $\exp\{r_{\boldsymbol{\alpha}}(\boldsymbol{x})\}$, notice that the $k$th derivative with respect to $\boldsymbol{\alpha}$, $\nabla_{\boldsymbol{\alpha}}^{(k)} \exp\{r_{\boldsymbol{\alpha}}(\boldsymbol{x})\}$, is given by $\boldsymbol{h}(\boldsymbol{x})^{\otimes k} \exp\{r_{\boldsymbol{\alpha}}(\boldsymbol{x})\}$, where $\otimes$ denotes the tensor product. Moreover, by equivalence of norms, for any $\tau > 0$ there exists constants $c_k, c_{\tau,k} \geq 0$ such that $\|\boldsymbol{h}(\boldsymbol{x})^{\otimes k}\| \leq c_k \|\boldsymbol{h}(\boldsymbol{x})\|^k \leq c_{\tau,k} \exp\{\tau\|\boldsymbol{h}(\boldsymbol{x})\|\}$. So by choosing $\tau$ such that the Euclidean ball of radius $\varepsilon + 2d\tau$ centered at $2\boldsymbol{\alpha}^*$ is contained in $N(2\boldsymbol{\alpha}^*)$, our same argument yields a dominating function of the form (27) for $\|\boldsymbol{\alpha} - \boldsymbol{\alpha}^*\| < \varepsilon/(2d)$, with exponent $(\boldsymbol{\alpha}^* + (\varepsilon/2 + d\tau)\boldsymbol{\beta}^{(\ell)})^\top \boldsymbol{h}(\boldsymbol{x})$. $\qquad \square$

**Remark.** Under weaker assumptions (Assumptions C.1 and C.2), the proof of Lemma C.11 implies that the log-normalizer $A_P(\boldsymbol{\alpha})$ has derivatives of all orders on $\Xi$. Indeed, this is a consequence of Equation (27), which implies that for every $\boldsymbol{\alpha}$, there exists a neighborhood of $\boldsymbol{\alpha}$ contained in $\Xi$ on which the $k$th derivative of $\exp\{r_{\boldsymbol{\alpha}}(\boldsymbol{x})\}$ is uniformly $p_{\theta_{\text{base}}}$-dominated. This allows one to exchange differentiation and integration in the definition of $A_P(\boldsymbol{\alpha})$.

## D. Calibrating Continuous-time Diffusion Models

In this section, we provide details on how CGM can be applied to calibrate continuous-time diffusion models. First, in Appendix D.1 we give background on continuous-time diffusion models, including how we sample from $p_\theta$ and compute densities $p_\theta/p$ with respect to a dominating measure $p$. This enables us to employ CGM-relax and CGM-reward for calibrating a pre-trained diffusion model. In Appendix D.2, we discuss the setting where the calibration function depends only on the terminal time of the diffusion process. In this case, the solution to the maximum entropy problem is a diffusion process, and we provide a mathematical characterization of its drift. Finally, in Appendix D.3 we describe how the base diffusion model can be initialized to produce exact samples from a GMM or product of GMMs. This allows us to initialize the base model exactly in our synthetic experiments.

### D.1. Continuous-time Diffusion Models

A continuous-time diffusion model is the solution to the $k$-dimensional stochastic differential equation (SDE)

$$d\boldsymbol{x}(t) = \boldsymbol{b}_\theta(\boldsymbol{x}(t), t)dt + \sigma(t)d\boldsymbol{w}(t), \; \boldsymbol{x}(0) \sim p_{\text{init}}, \tag{29}$$

where $(\boldsymbol{w}(t))_{0 \leq t \leq 1}$ is a standard $k$-dimensional Brownian motion, $\boldsymbol{b}_\theta$ is a neural network drift function, $\sigma$ is a diffusion coefficient, and $p_{\text{init}}$ is a known distribution from which sampling is tractable. Oksendal (2013, Theorem 5.2.1) provides conditions on $\boldsymbol{b}_\theta$ and $\sigma$ that ensure there exists a unique solution to the SDE (29). We denote the solution, which is a probability distribution on continuous paths, by $p_\theta$, and we write $p_\theta(\boldsymbol{x}(t))$ for the distribution of the state at time $t$.

**Sampling from diffusion models.** To sample from $p_\theta$, we use the Euler-Maruyama method. Specifically, we discretize $[0, 1]$ into $T$ time bins $[0, 1/T], \ldots, [(T-1)/T, 1]$ and sample a path $(\widehat{\boldsymbol{x}}(t))_{0 \leq t \leq 1}$ according to $\widehat{\boldsymbol{x}}(0) \sim p_{\text{init}}$

$$\widehat{\boldsymbol{x}}(t + \Delta t) = \widehat{\boldsymbol{x}}(t) + \Delta t \boldsymbol{b}_\theta(\widehat{\boldsymbol{x}}(t), t) + \sigma(t)\sqrt{\Delta t}\boldsymbol{z}(t), \; 0 < \Delta t \leq 1/T \tag{30}$$

for each $t = 0, 1/T, \ldots, (T-1)/T$, where $\boldsymbol{z}(0), \ldots, \boldsymbol{z}((T-1)/T)$ are independent standard multivariate normal random variables. The Euler-Maruyama method with additive noise $\sigma(t)$ has strong order of convergence 1, meaning its error in approximating the solution to the SDE (29) is

$$\mathbb{E}_{p_\theta}[\|\widehat{\boldsymbol{x}}(t) - \boldsymbol{x}(t)\|] \leq C(T^{-1}), \quad 0 \leq t \leq 1$$

for $C$ a constant independent of $T$. In other words, as we increase the number of time bins $T$, we can expect our sample paths drawn according to the Euler-Maruyama scheme to more faithfully approximate samples from the distribution $p_\theta$.

**Computing densities.** In order to employ CGM-relax and CGM-reward, $p_\theta$ and $p_{\theta_{\text{base}}}$ must have densities with respect to one another, and it must be possible to compute these densities. Girsanov's Theorem (Cameron & Martin, 1944; Girsanov, 1960) provides conditions that guarantee these densities to exist and an expression for computing them.

**Theorem D.1** (Girsanov's Theorem). *Suppose the SDEs*

$$\nu_1(\boldsymbol{x}) : d\boldsymbol{x}(t) = \boldsymbol{b}_1(\boldsymbol{x}(t), t)dt + \sigma(t)d\boldsymbol{w}(t), \quad 0 \le t \le 1$$
$$\nu_2(\boldsymbol{x}) : d\boldsymbol{x}(t) = (\boldsymbol{b}_1(\boldsymbol{x}(t), t) + \sigma(t)\boldsymbol{b}_2(\boldsymbol{x}(t), t))dt + \sigma(t)d\boldsymbol{w}(t), \quad 0 \le t \le 1$$

*satisfy $\sigma(t) > 0$, $0 < t < 1$, have the same initial law $\nu_1(\boldsymbol{x}_0) = \nu_2(\boldsymbol{x}_0)$, and admit unique, strong solutions, $\nu_1$ and $\nu_2$. Suppose also*

$$\left[\frac{\nu_2(\boldsymbol{x})}{\nu_1(\boldsymbol{x})}\right]_t := \exp\left\{\sum_{i=1}^k \int_0^t \boldsymbol{b}_2(\boldsymbol{x}(t), t)[i]d\boldsymbol{w}^{\nu_1}(t)[i] - \frac{1}{2}\int_0^t \|\boldsymbol{b}_2(\boldsymbol{x}(t), t)\|^2 dt\right\} \tag{31}$$

*is a $\nu_1$-martingale, where $(\boldsymbol{w}^{\nu_1}(t))_{0 \le t \le 1}$ is a $k$-dimensional $\nu_1$-Brownian motion and $d\boldsymbol{w}^{\nu_1}(t)[i]$, $i = 1, \ldots, k$ denotes the Itô stochastic integral. Then the probability measure $\nu_2$ has a density with respect to $\nu_1$. In particular, for any bounded functional $\Phi$ defined on $C[0,1]^k$,*

$$\mathbb{E}_{\nu_2}[\Phi(\boldsymbol{x})] = \mathbb{E}_{\nu_1}\left[\Phi(\boldsymbol{x})\left[\frac{\nu_2(\boldsymbol{x})}{\nu_1(\boldsymbol{x})}\right]_1\right].$$

If $\|\sigma(t)^{-1}(\boldsymbol{b}_\theta(\boldsymbol{x}(t), t) - \boldsymbol{b}_{\theta_{\text{base}}}(\boldsymbol{x}(t), t))\|$ is bounded, $([p_\theta(\boldsymbol{x})/p_{\theta_{\text{base}}}(\boldsymbol{x})]_t)_{0 \le t \le 1}$ is a martingale with respect to $p_{\theta_{\text{base}}}$. Consequently, Girsanov's Theorem tells us that the probability density of $p_\theta$ with respect to $p_{\theta_{\text{base}}}$ is given by

$$\frac{p_\theta(\boldsymbol{x})}{p_{\theta_{\text{base}}}(\boldsymbol{x})} := \exp\left\{\sum_{i=1}^k \int_0^1 u_\theta(\boldsymbol{x}(t), t)[i]d\boldsymbol{w}^{p_{\theta_{\text{base}}}}(t)[i] - \frac{1}{2}\int_0^1 \|u_\theta(\boldsymbol{x}(t), t)\|^2 dt\right\},$$
$$u_\theta(\boldsymbol{x}(t), t) := \sigma(t)^{-1}(\boldsymbol{b}_\theta(\boldsymbol{x}(t), t) - \boldsymbol{b}_{\theta_{\text{base}}}(\boldsymbol{x}(t), t)) \tag{32}$$

This expression for the density of $p_\theta$ with respect to $p_{\theta_{\text{base}}}$ allows us to compute the KL divergence between the probability measures $p_\theta$ and $p_{\theta_{\text{base}}}$ according to

$$\text{D}_{\text{KL}}(p_\theta \| p_{\theta_{\text{base}}}) = \frac{1}{2}\int_0^1 \mathbb{E}_{p_\theta}\|u_\theta(\boldsymbol{x}(t), t)\|^2 dt.$$

The stochastic integral term vanishes since it has expectation zero.

When $(\widehat{\boldsymbol{x}}(t))_{0 \le t \le 1}$ is sampled from the Euler-Maruyama approximation to $p_{\theta_{\text{base}}}$, we approximate (32) by replacing the integrals with

$$\int_0^1 u_\theta(\widehat{\boldsymbol{x}}(t), t)[i]d\boldsymbol{w}^{p_{\theta_{\text{base}}}}(t)[i] \approx T^{-1/2}\sum_{t=0}^{T-1} u_\theta(\widehat{\boldsymbol{x}}(t/T), t/T)[i](\boldsymbol{z}((t+1)/T) - \boldsymbol{z}(t/T))$$
$$\int_0^1 u_\theta(\widehat{\boldsymbol{x}}(t), t)^2[i]dt \approx T^{-1}\sum_{t=0}^{T-1} u_\theta(\widehat{\boldsymbol{x}}(t/T), t/T)^2[i]$$

where $\boldsymbol{z}(0), \ldots, \boldsymbol{z}((T-1)/T)$ are the same random variables from (30). This same approximation to the density ratio (32) can be derived by writing out the density ratio of $\widehat{p}_\theta(\widehat{\boldsymbol{x}}(0), \widehat{\boldsymbol{x}}(1/T), \ldots, \widehat{\boldsymbol{x}}(1))$ and $\widehat{p}_{\theta_{\text{base}}}(\widehat{\boldsymbol{x}}(0), \widehat{\boldsymbol{x}}(1/T), \ldots, \widehat{\boldsymbol{x}}(1))$, where $\widehat{p}_\theta$ is the probability distribution defined by the Euler-Maruyama discretization of $\widehat{p}_\theta$.

**Efficient gradient computation.** CGM-relax and CGM-reward require computing gradients of the density ratio $\frac{p_\theta(\boldsymbol{x})}{p_{\text{stop-grad}(\theta)}(\boldsymbol{x})}$. By applying Girsanov's Theorem to compute the density ratio, differentiating the result, and substitut-

ing in our approximations to the integrals, we obtain

$$
\begin{aligned}
\nabla_\theta \frac{p_\theta(\boldsymbol{x})}{p_{\text{stop-grad}(\theta)}(\boldsymbol{x})} &= \sum_{i=1}^{k} \int_0^1 \nabla_\theta \sigma(t)^{-1} \boldsymbol{b}_\theta(\widehat{\boldsymbol{x}}(t), t)[i] d\boldsymbol{w}^{p_{\theta_{\text{base}}}}(t)[i] \\
&\approx T^{-1/2} \sum_{i=1}^{k} \sum_{t=0}^{T-1} \nabla_\theta \sigma(t/T)^{-1} \boldsymbol{b}_\theta(\widehat{\boldsymbol{x}}(t/T), t/Y)[i](\boldsymbol{z}((t+1)/T)[i] - \boldsymbol{z}(t/T)[i]) \\
&= T^{-1/2} \sum_{t=0}^{T-1} \sigma(t/T)^{-1} \sum_{i=1}^{k} \nabla_\theta \boldsymbol{b}_\theta(\widehat{\boldsymbol{x}}(t/T), t/Y)[i](\boldsymbol{z}((t+1)/T)[i] - \boldsymbol{z}(t/T)[i]).
\end{aligned} \tag{33}
$$

For high-dimensional diffusion models (e.g. Genie2 in our Section 4.1 experiments) memory constraints preclude the naive approach to computing Equation (33) by instantiating each term in memory and simultaneously back-propagating gradients through all terms at once. However, because the gradient is a sum across time, it can be computed in chunks. In practice, we divide $\{0, \ldots, T\}$ into $\lceil T/\texttt{chunk\_size} \rceil$ blocks of approximately equal size, where $\texttt{chunk\_size}$ is the largest chunk size that can fit into memory.

### D.2. Solution to the Maximum Entropy Problem

When the base model $p_{\theta_{\text{base}}}(\boldsymbol{x})$ constitutes a continuous-time diffusion model (29) satisfying certain regularity properties, and the constraint function $\boldsymbol{h}$ depends only on the path at time $t = 1$, there exists a closed-form solution to the maximum entropy problem (4).

Let $p$ be the law of an SDE having diffusion coefficient $\sigma$ and initial distribution $p'_{\text{init}}(\boldsymbol{x}(0))$; this is necessary for $p \ll p_{\theta_{\text{base}}}$ by Girsanov's Theorem (Theorem D.1). By the chain rule for the KL divergence, the objective for the maximum entropy problem defined on the full path measures is

$$
\begin{aligned}
&\text{D}_{\text{KL}}\left(p(\boldsymbol{x}) \parallel p_{\theta_{\text{base}}}(\boldsymbol{x})\right) \\
&= \text{D}_{\text{KL}}\left(p(\boldsymbol{x}(1)) \parallel p_{\theta_{\text{base}}}(\boldsymbol{x}(1))\right) + \mathbb{E}_{p(\boldsymbol{x}(0))}[\text{D}_{\text{KL}}\left(p(\cdot|\boldsymbol{x}(0)) \parallel p_{\theta_{\text{base}}}(\cdot|\boldsymbol{x}(0))\right)].
\end{aligned}
$$

The KL divergence is computed according to Girsanov's Theorem.

From here, by the maximum entropy principle applied to the marginal at time $t = 1$, the first term in the objective is lower bounded by

$$
\text{D}_{\text{KL}}\left(p(\boldsymbol{x}(1)) \parallel p_{\theta_{\text{base}}}(\boldsymbol{x}(1))\right) \geq \text{D}_{\text{KL}}\left(p_{\boldsymbol{\alpha}_0^*}(\boldsymbol{x}(1)) \parallel p_{\theta_{\text{base}}}(\boldsymbol{x}(1))\right)
$$

where $p_{\boldsymbol{\alpha}_0^*}(\boldsymbol{x}(1))$ is the solution to the maximum entropy problem in $k$-dimensional Euclidean space. Consequently, if we can show that there exists an SDE $p(\boldsymbol{x})$ satifying $p(\boldsymbol{x}(1)) = p_{\boldsymbol{\alpha}_0^*}(\boldsymbol{x}(1))$ and $p(\cdot|\boldsymbol{x}(1)) = p_{\theta_{\text{base}}}(\cdot|\boldsymbol{x}(1))$, then $p$ is the solution to the maximum entropy problem. This is the subject of the following result:

**Proposition D.2** (Maximum entropy solution for a diffusion model). *Suppose that the constraint function $\boldsymbol{h}$ depends only on the value of the path at time $t = 1$ and is bounded and continuous. Moreover, assume that $\boldsymbol{x}(0) \perp \boldsymbol{x}(1)$ under the base model $p_{\theta_{\text{base}}}(\boldsymbol{x})$.*

*Then the solution to the maximum entropy problem is a diffusion process*

$$
p^* : d\boldsymbol{x}(t) = \{\boldsymbol{b}_{\theta_{\text{base}}}(\boldsymbol{x}(t), t) + \sigma(t)\boldsymbol{u}^*(\boldsymbol{x}(t), t)\}dt + \sigma(t)d\boldsymbol{w}(t)
$$

*satisfying $p^*(\boldsymbol{x}(0)) = p_{\text{init}}$. The drift $\boldsymbol{u}^*(\boldsymbol{x}(t), t)$ admits the Feynman-Kac characterization*

$$
u^*(\boldsymbol{x}, t) = \sigma(t)\nabla_{\boldsymbol{x}} \log \mathbb{E}_{p_{\theta_{\text{base}}}}[\exp\{r_{\boldsymbol{\alpha}_0^*}(\boldsymbol{x}(1))\}|\boldsymbol{x}(t) = \boldsymbol{x}],
$$

*where $\boldsymbol{\alpha}_0^*$ are the parameters corresponding to the maximum entropy solution in $k$-dimensional Euclidean space (4) with base distribution $p_{\theta_{\text{base}}}(\boldsymbol{x}(1))$ and constraint function $\boldsymbol{h}(\boldsymbol{x})$.*

*Finally, $p^*(\boldsymbol{x})$ satisfies $p^*(\cdot|\boldsymbol{x}(1)) = p_{\theta_{\text{base}}}(\cdot|\boldsymbol{x}(1))$ and $p^*(\boldsymbol{x}(1)) = p_{\boldsymbol{\alpha}_0^*}(\boldsymbol{x}(1))$.*

We refer the reader to Domingo-Enrich et al. (2025, Theorem 1) for a proof. The result is a consequence of standard results in the theory of diffusion processes, specifically the Doob $h$-transform (Oksendal, 2013, Chapter 7).

The assumption that, under $p_{\theta_{\text{base}}}(\boldsymbol{x})$, the path at time $t = 0$ is independent of the path at time $t = 1$ is necessary to ensure that $p^*(\boldsymbol{x}(0))$ can be chosen to be equal to $p_{\text{init}}(\boldsymbol{x}(0))$. This is desirable because, by design, $p_{\text{init}}$ is a distribution from which sampling is tractable. However, when the independence assumption does not hold, $p^*(\boldsymbol{x}(0))$ cannot be chosen to be equal to $p_{\theta_{\text{base}}}(\boldsymbol{x}(0))$ (Denker et al., 2024, Appendix G.2).

Although, at first glance, this independence assumption may appear strong, Domingo-Enrich et al. (2025, Theorem 1) prove that diffusion models whose initial distribution is Gaussian noise and whose terminal distribution is the data distribution satisfies this property (i.e., variance-preserving SDEs). One example of a commonly used noise schedule satisfying this property is $\sigma(t) = t^{-1}$, which is singular at time $t = 0$.

Proposition D.2 tells us that when we fine-tune a diffusion model with CGM to satisfy a constraint on its terminal distribution $\mathbb{E}_{p_{\theta_{\text{base}}}(\boldsymbol{x}(1))}[\boldsymbol{h}(\boldsymbol{x})] = \boldsymbol{h}^*$, we expect the terminal distribution of the base model to change to satisfy the constraint, while the conditional path distribution given the endpoint should be preserved. In other words, seeking the distribution over paths that is closest in KL distance to the base model amounts to shifting the terminal distribution while leaving the conditional distributions unchanged.

### D.3. Exact Sampling from a Gaussian Mixture Model

In each of our synthetic data experiments, we initialize our base diffusion model $p_{\theta_{\text{base}}}$ such that $p_{\theta_{\text{base}}}(\boldsymbol{x}(1))$ is equal to a GMM. We achieve this by representing $p_{\theta_{\text{base}}}$ as the reversal of a forward diffusion process. A forward diffusion process draws samples from the target GMM density $\boldsymbol{x}(1) \sim p_{\text{target}}$ and then noises them according to the linear SDE

$$\overrightarrow{p} : d\boldsymbol{x}(t) = \frac{1}{2}\kappa(t)\boldsymbol{x}(t)dt + \sigma(t)d\boldsymbol{w}(t),\ 0 \leq t \leq 1. \tag{34}$$

When the diffusion coefficient is chosen such that $\sigma(t) = \sqrt{\kappa(t)}$ and the linear coefficient $(\kappa(t))_{0 \leq t \leq 1}$ satisfies $\kappa(t) \geq 0,\ \int_0^1 \kappa(t)dt = +\infty$, then $\overrightarrow{p}(\boldsymbol{x}(0)) \overset{d}{=} \mathcal{N}(\boldsymbol{0}, \mathbb{I})$. Such a noise schedule satisfies the independence assumption in Proposition D.2. We choose $\kappa(t) = t^{-1}$. Simply, (34) turns samples from $p_{\text{target}}$ into Gaussian noise. In practice, since the drift and diffusion coefficients defined by $\kappa(t)$ are unbounded (which violates the assumptions for existence and uniqueness of the solution to the SDE from Appendix D.1), we cap $\kappa(t)$ at some large $M$.

A foundational result in diffusion processes (Anderson, 1982) states that the reversal of (34) is another diffusion process that is given by

$$\overleftarrow{p} : d\boldsymbol{x}(t) = \left\{\sigma(t)^2 \nabla_{\boldsymbol{x}} \log \overrightarrow{p}(\boldsymbol{x}(t)) + \frac{1}{2}\kappa(t)\boldsymbol{x}(t)\right\}dt + \sigma(t)d\boldsymbol{w}(t),\ 0 \leq t \leq 1 \tag{35}$$

with $\overleftarrow{p}(\boldsymbol{x}(0)) \overset{d}{=} \overrightarrow{p}(\boldsymbol{x}(0))$. The probability distributions defined by (34) and (35) are equal in law. $\nabla_{\boldsymbol{x}} \log \overrightarrow{p}(\boldsymbol{x}(t))$ is called the *score* of the forward process (34).

Equation (35) is useful since it tells how to generate samples from $p_{\text{target}}$: first draw samples from $\overrightarrow{p}(\boldsymbol{x}(0)) \approx \mathcal{N}(\boldsymbol{x}(0) \mid \boldsymbol{0}, \mathbb{I})$, then solve the SDE (35) numerically using Euler-Maruyama, for example. However, for general target distributions $p_{\text{target}}$, the score of the forward process is intractable, which yields the backward diffusion process (35) also intractable.

In the case of a GMM, though, the score of the forward process is tractable. Indeed, for $p_{\text{target}}(\boldsymbol{x}(1)) = \sum \pi_i \mathcal{N}(\boldsymbol{x}(1) \mid \boldsymbol{\mu}_i, \boldsymbol{\Sigma}_i)$, we compute

$$\overrightarrow{p}(\boldsymbol{x}(t)) = \int \overrightarrow{p}(\boldsymbol{x}(t)|\boldsymbol{x}(1))p_{\text{target}}(\boldsymbol{x}(1))d\boldsymbol{x}(1)$$

$$= \sum \pi_i \int \mathcal{N}(\boldsymbol{x}(t)|m(t)\boldsymbol{x}(1), s(t)^2\mathbb{I})\mathcal{N}(\boldsymbol{x}(1) \mid \boldsymbol{\mu}_i, \boldsymbol{\Sigma}_i)d\boldsymbol{x}(1)$$

$$= \sum \pi_i \mathcal{N}(\boldsymbol{x}(t)|m(t)\boldsymbol{\mu}_i, s(t)^2\mathbb{I} + m(t)^2\boldsymbol{\Sigma}_i),$$

where $m(t)$ and $s(t)$ are defined by the forward diffusion process $(\kappa(t))_{0 \leq t \leq 1}$. For $\kappa(t) = t^{-1}$, we have $m(t) = t^{1/2}$ and $s(t) = (1-t)^{1/2}$. Using this expression for $\overrightarrow{p}(\boldsymbol{x}(t))$, we initialize $p_{\theta_{\text{base}}}(\boldsymbol{x})$ to the exact reversal of the forward process (34) according to (35).

# E. Comparison to Augmented Lagrangian Method

In this section, we compare CGM-relax and CGM-reward to a modified version of the augmented Lagrangian algorithm (Hestenes, 1969), which solves a general constrained optimization problem. This algorithm can be viewed as a generalization of the primal-dual algorithm proposed by Khalafi et al. (2026) for fine-tuning diffusion models. We demonstrate that on our synthetic data experiments, the AL algorithm performs no better than CGM-relax, and it introduces another hyperparameter (the dual parameter update frequency) to be chosen by the practitioner.

In particular, suppose one would like to solve

$$\min_{\boldsymbol{z}\in\mathbb{R}^p} f(\boldsymbol{z}), \quad \text{subject to } \boldsymbol{c}(\boldsymbol{z}) = \boldsymbol{c}^*,$$

where $f : \mathbb{R}^p \to \mathbb{R}$ is the objective function and $\boldsymbol{c} : \mathbb{R}^p \to \mathbb{R}^d$ is the constraint function. The AL method forms the "augmented Lagrangian" $\mathcal{L}_\lambda : \mathbb{R}^p \times \mathbb{R}^d \to \mathbb{R}$ with penalty parameter $\lambda \geq 0$ and dual variables $\boldsymbol{u} \in \mathbb{R}^d$, defined

$$\mathcal{L}_\lambda(\boldsymbol{z}, \boldsymbol{u}) = f(\boldsymbol{z}) + \boldsymbol{u}^\top (\boldsymbol{c}(\boldsymbol{z}) - \boldsymbol{c}^*) + \frac{\lambda}{2}\|\boldsymbol{c}(\boldsymbol{z}) - \boldsymbol{c}^*\|^2. \tag{36}$$

$\mathcal{L}_\lambda$ differs from the ordinary Lagrangian due the presence of the penalty term $\frac{\lambda}{2}\|\boldsymbol{c}(\boldsymbol{z}) - \boldsymbol{c}^*\|^2$. The AL algorithm alternates between minimizing $\mathcal{L}_\lambda$ with respect to $\boldsymbol{z}$ and updating the dual variables:

$$\boldsymbol{z}^{(k)} \leftarrow \arg\min_{\boldsymbol{z}\in\mathbb{R}^p} \mathcal{L}_\lambda(\boldsymbol{z}, \boldsymbol{u}^{(k)})$$
$$\boldsymbol{u}^{(k+1)} \leftarrow \boldsymbol{u}^{(k)} + \lambda(\boldsymbol{c}(\boldsymbol{z}^{(k)}) - \boldsymbol{c}^*). \tag{37}$$

Some variants of the AL method also update the penalty $\lambda$ according to some schedule. The augmented Lagrangian algorithm can be viewed as a proximal-point algorithm applied to the dual function (Rockafellar, 1976)

$$\boldsymbol{u}^{(k+1)} \leftarrow \arg\max_{\boldsymbol{u}\in\mathbb{R}^d} \left\{ d(\boldsymbol{u}) - \frac{1}{2\lambda}\|\boldsymbol{u} - \boldsymbol{u}^{(k)}\| \right\}, \ d(\boldsymbol{u}) = \min_{\boldsymbol{z}\in\mathbb{R}^p} f(\boldsymbol{z}) + \boldsymbol{u}^\top (\boldsymbol{c}(\boldsymbol{z}) - \boldsymbol{c}^*).$$

In the setting of the calibration problem (1), we identify $\boldsymbol{z} = \theta$, $f(\theta) = \mathrm{D}_{\mathrm{KL}}(p_\theta \| p_{\theta_{\mathrm{base}}})$, $c(\theta) = \mathbb{E}_{p_\theta}[\boldsymbol{h}(\boldsymbol{x})]$, $\boldsymbol{c}^* = \boldsymbol{h}^*$. However, the augmented Lagrangian does not admit a closed-form minimizer with respect to the model parameters $\theta$. Consequently, the AL algorithm as stated in Equation (37) cannot be applied. Instead, we propose alternating between performing a stochastic gradient update to the model parameters $\theta$ and updating the Lagrange multipliers $\boldsymbol{u}$. This introduces an additional algorithmic hyperparameter $\ell$ representing how frequently (i.e., after how many iterations) the dual variables are updated. We provide pseudocode for our implementation in Algorithm 3, which we refer to as 'CGM-AL'.

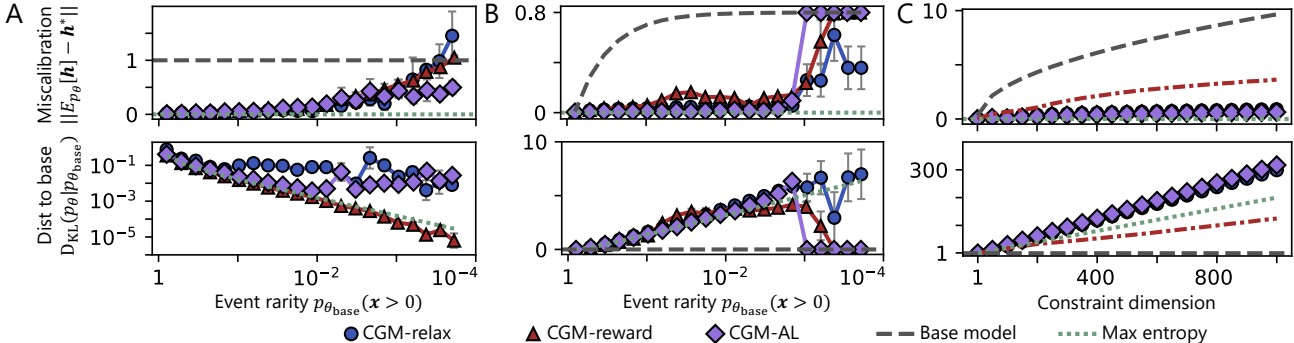

*Figure 7.* Comparison of CGM to the augmented Lagrangian (AL) algorithm on three synthetic data examples. **A**: Reweighting the probability of a rare mode in a 1D Gaussian mixture by a constant (multiplicative) factor. **B**: Reweighting the probability of a rare mode in a 1D Gaussian mixture to a fixed value. **C**: Reweighting the modes of a $k$-dimensional distribution, whose components are independent Gaussian mixtures. The number of constraints and KL to the base model increase as $k$ increases.

We compare CGM-AL against CGM-relax and CGM-reward on the rare event and increasing constraint dimension experiments described in Section 3. We also perform comparisons on the second set of rare event experiments that we later describe in Appendix F.1.

For CGM-AL, we select $\log(\lambda)$ by performing a grid search with the same number of points as for CGM-relax: for the rare event experiments, we use the same grid of values; for the increasing constraint dimension setting, we use $[0, 2]$. In addition to $\lambda$, we consider two potential values for the Lagrange multiplier update frequency, $\ell \in \{20, 100\}$.

From Figure 7, we observe that CGM-relax performs comparably to CGM-AL. In the first rare event experiment (A), CGM-AL better reduces the calibration error for small $p_{\theta_{\text{base}}}(\boldsymbol{x}(0) > 0)$ over CGM-relax, whereas the opposite is true for the second rare event experiment (B). For the increasing constraint dimension experiments, both algorithms perform comparably. We attribute the success of CGM-relax to our choice of $\lambda$ via grid search. Although the optimization landscape becomes ill-conditioned as $\lambda$ becomes is small, we select $\lambda$ to balance between constraint violation and KL distance to the base model $p_{\theta_{\text{base}}}$. Moreover, unlike the AL algorithm, CGM-relax does not require tuning a second hyperparameter, the Lagrange multiplier update frequency $\ell$. For these practical reasons, we prefer CGM-relax to CGM-AL, although we still view the latter as a viable option for generative model calibration.

---

**Algorithm 3** CGM-AL fine-tuning (augmented Lagrangian)

---

**Input**: $p_{\theta_{\text{base}}}, \boldsymbol{h}(\cdot), \boldsymbol{h}^*, M, \lambda$, and $\ell$

▷ Initialize model and dual variables
$p_\theta \leftarrow p_{\theta_{\text{base}}}, \boldsymbol{u} \leftarrow \boldsymbol{0}, t \leftarrow 0$
**while** not converged **do**
    ▷ Sample and compute weights
    $\boldsymbol{x}_1, \ldots, \boldsymbol{x}_M \overset{i.i.d.}{\sim} p_{\texttt{stop-grad}(\theta)}$
    $w_m \leftarrow p_\theta(\boldsymbol{x}_m)/p_{\texttt{stop-grad}(\theta)}(\boldsymbol{x}_m)$

    ▷ KL loss with LOO baseline
    $l_m \leftarrow \log p_{\texttt{stop-grad}(\theta)}(\boldsymbol{x}_m)/p_{\theta_{\text{base}}}(\boldsymbol{x}_m)$
    $l_m^{\text{LOO}} \leftarrow l_m - \frac{1}{M-1}\sum_{m'\neq m} l_{m'}$
    $\widehat{\mathcal{L}}^{\text{KL}} \leftarrow \frac{1}{M}\sum w_m l_m^{\text{LOO}}$

    ▷ Constraint violation loss
    $\boldsymbol{h}_m \leftarrow w_m(\boldsymbol{h}(\boldsymbol{x}_m) - \boldsymbol{h}^*)$
    $\widehat{\Delta \boldsymbol{h}} \leftarrow \frac{1}{M}\sum \boldsymbol{h}_m$
    $\widehat{\mathcal{L}}^{\text{viol}} \leftarrow \|\widehat{\Delta \boldsymbol{h}}\|^2 - \frac{1}{M}\widehat{\text{Var}}[\boldsymbol{h}_{1:M}]$,
        $\widehat{\text{Var}}[\boldsymbol{h}_{1:M}] = \frac{1}{M-1}\sum \|\boldsymbol{h}_m - \widehat{\Delta \boldsymbol{h}}\|^2$

    ▷ Augmented Lagrangian loss
    $\widehat{\mathcal{L}}^{\text{AL}} \leftarrow \widehat{\mathcal{L}}^{\text{KL}} + \boldsymbol{u}^\top \widehat{\Delta \boldsymbol{h}} + \frac{\lambda}{2}\widehat{\mathcal{L}}^{\text{viol}}$

    ▷ Primal update
    $\theta \leftarrow \text{gradient-step}(\theta, \nabla_\theta \widehat{\mathcal{L}}^{\text{AL}})$

    ▷ Dual update every $\ell$ iterations
    **if** $t \bmod \ell = 0$ **then**
        $\boldsymbol{u} \leftarrow \boldsymbol{u} + \lambda\,\texttt{stop-grad}(\widehat{\Delta \boldsymbol{h}})$
    **end if**
    $t \leftarrow t + 1$
**end while**

---

*Table 1.* Training configurations for experiments. Batch (sub-batch) indicates the number of samples per batch and the sub-batch size used to fit gradient computations into memory. $\mathcal{S}_V^L$ denotes the set of all sequences with vocabulary size $V$ and length $L$.

| Hyperparameter | Genie2 | ESM3-open | TarFlow | Gemma-2-9B-IT |
|---|---|---|---|---|
| Initial learning rate | $10^{-5}$ | $10^{-4}$ | $10^{-6}$ | $2 \times 10^{-6}$ |
| Batch (sub-batch) | $64\,(16)$ | $256\,(64)$ | $256\,(16)$ | $512\,(64)$ |
| Training steps | 100 | 100 | 50 | 200 |
| $x$ Space | $(\mathbb{R}^{100 \times 3})^{100}$ | $(\mathcal{S}_{4096}^{100})^{50}$ | $\mathbb{R}^{256 \times 256 \times 3}$ | $\mathcal{S}_{256,000}^{200}$ |
| Constraint dims ($k$) | 99 | 99 | 5 | 8 |
| Model parameters | 15M | 1.4B | 463M | 9B (110M trainable) |
| Training time (hrs) | 48 | 2.3 | 3 | 6 |

## F. Additional Experimental Details

In this section, we provide further details regarding our experiments with CGM-relax and CGM-reward, both on synthetic data (Section 3) and in the real-data case studies (Section 4). We provide explanations regarding the generative model classes $p_\theta$, CGM constraint functions $\boldsymbol{h}$ and targets $\boldsymbol{h}^*$, choice of CGM hyperparameters $\lambda$ and $N$, model architectures, and training procedures. We also include additional samples from our models before and after calibration.

We perform all experiments using Adam with default momentum hyperparameters $\beta = (0.9, 0.999)$ and a cosine decay learning rate schedule. Additional common training details are shown in Table 1. We train all models on a single H100 GPU.

### F.1. Synthetic Data Experiments

In this subsection, we provide additional details regarding the synthetic data experiments described in Section 3.

**Experimental setup.** In the diffusion generative model, we parameterize the drift function $\boldsymbol{b}_\theta$ as

$$\boldsymbol{b}_\theta(\boldsymbol{x}(t), t) = \sigma(t)^2 \{\nabla_{\boldsymbol{x}} \log \overrightarrow{p}(\boldsymbol{x}(t)) - u_\theta(\boldsymbol{x}, t)\} + \frac{1}{2}\kappa(t)\boldsymbol{x}(t).$$

$u_\theta$ is a neural network with two hidden layers of dimension 256 and SiLU activations. $\log \overrightarrow{p}(\boldsymbol{x}(t))$ is the analytical score of the forward process that we described in Appendix D.3. In addition to $\boldsymbol{x}(t)$, we feed as input to $u_\theta$ a sinusoidal time embedding of dimension 32. By initializing the weights of the output layer of $u_\theta$ to zero, we ensure that $p_\theta$ is initialized at $p_{\theta_{\text{base}}}$, the reversal of the forward diffusion process $\overrightarrow{p}$. We perform sampling using $10^2 + 1$ timesteps on a uniform time grid.

All results reported in Figures 3 and 7 are averages over 10 iterates, with two standard errors. For CGM-reward, we use $N = 10^5$ samples to approximate $\boldsymbol{\alpha}^*$. Next, we discuss the parameters unique to each experiment described in Section 3, namely the choice of $\lambda$ for CGM-relax, the number of training iterations, and the learning rate.

**Rare event**: For our rare event experiments, we choose $\log(\lambda)$ according to a 6-point linear search over $[-3, 2]$, and we report the value that achieves the smallest calibration error. We perform calibration for $5 \cdot 10^3$ training iterations. For CGM-relax we use constant stepsize $10^{-4}$, and for CGM-reward we use constant stepsize $10^{-3}$.

In Figure 3A, where we plot the KL to the base model on a log scale, we compute the $\pm 2$ standard errors using the delta method approximation $\log \hat{\mu} \pm 2\frac{\text{SE}}{\hat{\mu}}$. $\hat{\mu}$ is the sample mean and SE is the standard error estimate, computed over the 10 replicates.

**Increasing constraint dimension**: For our increasing constraint dimension experiments, we choose $\log(\lambda)$ according to a 10-point linear search over $[-3, 0]$. We choose the largest value of $\lambda$ for which the calibration error is reduced by 90%. We perform calibration for $2 \cdot 10^3$ training iterations. For both CGM-relax and CGM-reward we use initial stepsize $10^{-3}$ and final stepsize $10^{-6}$.

**Calibrating a rare event to a fixed proportion.** Lastly, we discuss a second set of rare event experiments, the results of which are reported in Figure 7B. Again, we consider reweighting a rare mode in a 1D GMM. The calibration constraint is

$$\boldsymbol{h}(\boldsymbol{x}) = \mathbb{1}\{\boldsymbol{x}(1) > 0\}, \ \ \boldsymbol{h}^* = 0.8.$$

Dissimilar to the rare event setting described in Section 3, $\boldsymbol{h}^*$ does *not* vary with $p_{\theta_{\text{base}}}(\boldsymbol{x}(1) > 0)$. Consequently, as $p_{\theta_{\text{base}}}(\boldsymbol{x}(1) > 0)$ decreases, the KL divergence from the maximum entropy distribution to the base model will increase. We vary $p_{\theta_{\text{base}}}(\boldsymbol{x}(1) > 0)$ from 0.8 (already calibrated) to approximately $10^{-4}$.

For CGM-relax, we choose $\log(\lambda)$ according to a 10-point linear search of $[-3, 0]$, and we choose the largest value of $\lambda$ for which the calibration error is reduced by 90%. We perform calibration using batch size $M = 10^2$ and $2 \cdot 10^3$ training iterations. For both CGM-relax and CGM-reward, we use initial stepsize $10^{-3}$ and final stepsize $10^{-6}$.

From Figure 7B, we observe that even for $p_{\theta_{\text{base}}}(\boldsymbol{x}(1) > 0)$ as small as $10^{-3}$, CGM successfully upweights the rare mode to its target value. For small $p_{\theta_{\text{base}}}(\boldsymbol{x}(1) > 0)$, this requires deviating far from the pre-trained model, approximately 5 nats at $p_{\theta_{\text{base}}}(\boldsymbol{x}(1) > 0) = 10^{-3}$. For $p_{\theta_{\text{base}}}(\boldsymbol{x}(1) > 0)$ smaller than $10^{-3}$, CGM-relax exhibits large variability across random seeds: when enough samples are drawn from the rare mode throughout training, it nearly reaches the maximum entropy distribution; when this is not the case, it does not move from the base model. CGM-reward does not move from the base.

### F.2. Calibrating Genie2

For our experiments with Genie2, we represent $p_\theta$ as a continuous-time diffusion model defined over three-dimensional protein backbone coordinates with drift function defined by the SE(3)-equivariant encoder-decoder architecture from Lin et al. (2024a).

Since Genie2 is trained as the reversal of a discrete-time noising process (a DDPM, see Ho et al., 2020), we first convert the discrete-time denoising diffusion model to a (continuous-time) diffusion model. We achieve this by redefining the final timestep $T$ of the original denoising process to be time 1 of the continuous-time process. To define the drift function, we take the DDPM transition mean defined at each time $t$ in the discrete-time process, divide it by $1/T = T$, and define the drift function to be equal to the resulting value in between times $t/T$ and $(t + 1)/T$. The diffusion coefficient is similarly defined by the DDPM transition standard deviation at each time $t$ in the discrete-time process, but is instead scaled by $T^{1/2}$. This approach of converting the DDPM into a continuous-time diffusion model ensures that when the SDE is solved under the Euler-Maruyama scheme using a grid of $T$ timesteps (i.e., the original time grid used to define the DDPM), one samples from the original DDPM.

We perform sampling using $10^2$ timesteps and a non-uniform time grid: we sample the first 50 steps on the interval $[0, 0.05]$ and the remaining steps on the interval $[0.05, 1]$. We point out that the original Genie2 model was trained with $10^3$ denoising steps; we find that reducing the number of sampling steps dramatically decreases the runtime of CGM calibration. Our sampling scheme is possible since we redefined the base generative model to be a continuous-time diffusion process. We computed self-consistency metrics for the base Genie2 model sampled on the original time grid (with $10^3$ steps) and on our proposed grid (with $10^2$ steps), we did not observe any difference in sample quality.

For CGM-relax, we calibrate to $k{=}99$ constraints on the bivariate CDF of alpha helix and beta strand content. And for CGM-reward, we calibrate to $k{=}15$ constraints using $N = 2.5 \times 10^4$ samples from $p_{\theta_{\text{base}}}$. Since sampling from the Genie2 base model is time intensive, the sampling cost to compute $\widehat{\boldsymbol{\alpha}}_N$ (with small variance) is a downside to CGM-reward. Results reported in Figure 4 are averages over 3 trials, with two standard errors. We report results for CGM-relax with additional values of $\lambda$ in Figure 8.

**Self-consistency RMSD and design failures.** To assess the quality of our generations, we compute the root mean-square deviation (RMSD) between $C_\alpha$ atoms resulting from (i) unfolding our generated structures into predicted amino sequences, (ii) refolding each of these predicted sequences into a protein structure, and (iii) aligning the predicted structures to the original structure. The self-consistency RMSD (scRMSD) is defined as the smallest RMSD between the given structure and one of the corresponding predictions. We use ProteinMPNN (Dauparas et al., 2022) for our inverse folding model and ESMFold (Lin et al., 2023) for our folding model; we compute scRMSD from 8 sequences. The pipeline we employ was developed by Lin et al. (2024b). Once we have determined the scRMSD of a generated structure, we classify it as a "design failure" if its scRMSD is greater than 2Å. Intuitively, designability is a binary measure of whether or not a structure could have been plausibly produced by folding an amino acid sequence.

**Secondary structure annotation.** As discussed in Section 4.1, we measure the diversity of a collection of protein structures

by computing the proportion of residues that lie in each of the three protein secondary structure types. For the CATH domains and Genie2, we perform annotations using the Biotite package (Kunzmann & Hamacher, 2018), which considers only $C_\alpha$ backbone atoms. For the CATH proteins (Sillitoe et al., 2021), we obtain the secondary structure distribution by annotating the domains collected and published by Ingraham et al. (2019).

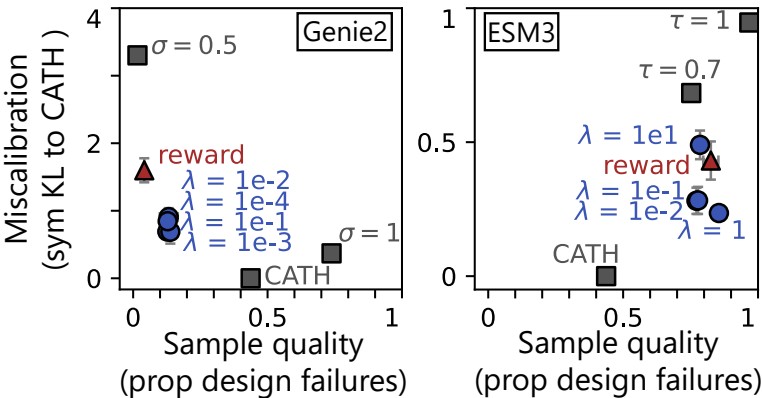

*Figure 8.* Results from varying the regularization hyperparameter $\lambda$ in CGM-relax for Genie2 and ESM3. We observe that CGM-relax is insensitive to the choice of $\lambda$, with the exception of $\lambda = 10^1$ for ESM3, in which case the fine-tuned model remains close to the base model.

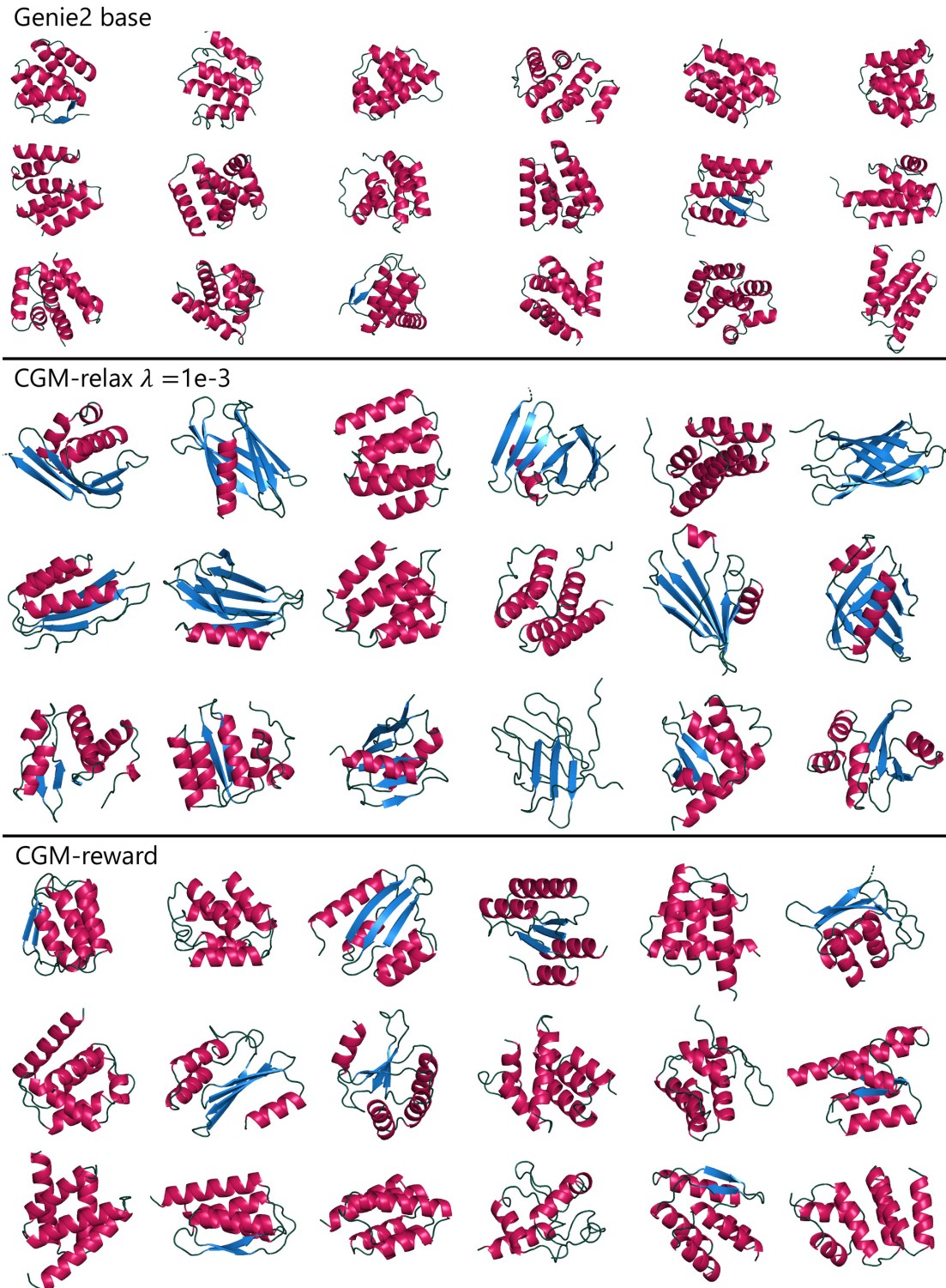

*Figure 9.* Random samples from the Genie2 model before calibration (top), after calibration using CGM-relax with 99 bivariate CDF constraints (middle), and after calibration using CGM-reward with 15 bivariate CDF constraints (bottom).

### F.3. Calibrating ESM3-open

In order to apply CGM to ESM3, we need to be able to sample from the model and to compute gradients of sample log-probabilities with respect to the model's parameters.

**Sampling method.** Following the method used by (Hayes et al., 2025), sampling is achieved by treating the model as a discrete time Markov chain that starts at a sequence of mask tokens and ends at a sequence of fully unmasked structure tokens. Each step $i$ of the chain consists of three steps and transitions from state $\boldsymbol{x}(i-1)$ to $\boldsymbol{x}(i)$.

1. Pick token indices $U(i)$ to unmask uniformly at random without replacement from the masked tokens of $\boldsymbol{x}(i-1)$.

2. Use the model $\pi_\theta$ to predict a categorical distribution $\pi_\theta^{(j)}(\cdot \mid \boldsymbol{x}(i-1))$ for $j \in U(i)$ for each of the newly unmasked tokens given the previous partially masked state.

3. Sample the values of those tokens from the predicted categorical distributions, resulting in $|U(i)|$ more unmasked tokens.

As implemented by (Hayes et al., 2025), we use $T = 50$ steps to sample 100-residue sequences and follow a cosine unmasking schedule. The cosine schedule determines the number of masked positions at each sampling step as

$$r(i) := \text{round}\left(100 \times \cos\left(\frac{\pi}{2}\frac{i}{T}\right)\right), \quad i = 0, \ldots, T.$$

Early sampling steps unmask few tokens per step, while later ones sample many at once. Intuitively, this let's the model sample more tokens in parallel once it has more information to predict the final sequence. Note that the number of tokens unmasked at step $i > 0$ is $|U(i)| = r(i-1) - r(i)$.

**Transition probabilities.** A Markov chain can be characterized by its initial state distribution, $\boldsymbol{x}(0) \sim \pi_0(\boldsymbol{x}(0))$ and its transition probabilities for going from one state to the next. The ESM3 sampling method starts fully masked, so has initial distribution $\pi_0(\boldsymbol{x}(0)) = \mathbf{1}\{\boldsymbol{x}(0) \text{ is fully masked}\}$. The transition probabilities follow from the sampling procedure and are

$$p_\theta(\boldsymbol{x}(i) \mid \boldsymbol{x}(i-1)) = C(i) \prod_{j \in U(i)} \pi_\theta^{(j)}(\boldsymbol{x}(i)[j] \mid \boldsymbol{x}(i-1)), \tag{38}$$

where $C(i)$ is a constant that accounts for randomly choosing which tokens to unmask. $C(i)$ does not depend on $\theta$ or the sampling trajectory $(\boldsymbol{x}(0), \boldsymbol{x}(1), \ldots, \boldsymbol{x}(T))$, since every unmasking order is equally likely. Note $U(i)$ can be computed from $\boldsymbol{x}(i-1)$ and $\boldsymbol{x}(i)$ by finding which tokens are masked in $\boldsymbol{x}(i-1)$ and not in $\boldsymbol{x}(i)$.

**Trajectory log-probability.** As in the neural SDE setting (Appendix D.1), the marginal likelihood of $\boldsymbol{x}_T$ is intractable, so we treat samples $\boldsymbol{x}$ as entire trajectories, $\boldsymbol{x} = (\boldsymbol{x}(0), \boldsymbol{x}(1), \boldsymbol{x}(2), \ldots, \boldsymbol{x}(T))$. Using the Markov property, the log-probability of a trajectory is

$$
\begin{aligned}
\log p_\theta(\boldsymbol{x}) &= \log\left(\pi_0(\boldsymbol{x}_0) \prod_{i=1}^{T} p_\theta(\boldsymbol{x}(i) \mid \boldsymbol{x}(i-1))\right) \\
&= \log \pi_0(\boldsymbol{x}_0) + \sum_{i=1}^{T} \log\left(C(i) \prod_{j \in U(i)} \pi_\theta^{(j)}(\boldsymbol{x}(i)[j] \mid \boldsymbol{x}(i-1))\right) \quad \text{(by Equation (38))} \\
&= \sum_{i=1}^{T} \log\left(C(i) \prod_{j \in U(i)} \pi_\theta^{(j)}(\boldsymbol{x}(i)[j] \mid \boldsymbol{x}(i-1))\right) \quad (\pi_0(\boldsymbol{x}(0)) = 1 \text{ by construction}) \\
&= \sum_{i=1}^{T} C(i) + \sum_{i=1}^{T} \sum_{j \in U(i)} \log \pi_\theta^{(j)}(\boldsymbol{x}(i)[j] \mid \boldsymbol{x}(i-1)).
\end{aligned}
$$

**Parameter gradients.** Now that we have defined the log-probability of $\boldsymbol{x}$, we can compute gradients with respect to $\theta$ as

$$\nabla_\theta \log p_\theta(\boldsymbol{x}) = \nabla_\theta \left( \sum_{i=1}^{T} C(i) + \sum_{i=1}^{T} \sum_{j \in U(i)} \log \pi_\theta^{(j)}(\boldsymbol{x}(i)[j] \mid \boldsymbol{x}(i-1)) \right)$$

$$= \sum_{i=1}^{T} \sum_{j \in U(i)} \nabla_\theta \log \pi_\theta^{(j)}(\boldsymbol{x}(i)[j] \mid \boldsymbol{x}(i-1)),$$

which conveniently is a sum over sampling steps. The decomposition of the gradient into a sum over sampling steps lets us compute parameter gradients using constant memory with respect to the number of sampling steps, which is critical for high-parameter-count models such as ESM3-open.

**Secondary structure annotation.** We use the ESM3 structure decoder and the ESM3 function `ProteinChain.infer_oxygen` to get heavy atom coordinates from sampled structure tokens. We then pass the coordinates to the Python package `PyDSSP` (Minami, 2023) to annotate secondary structure.

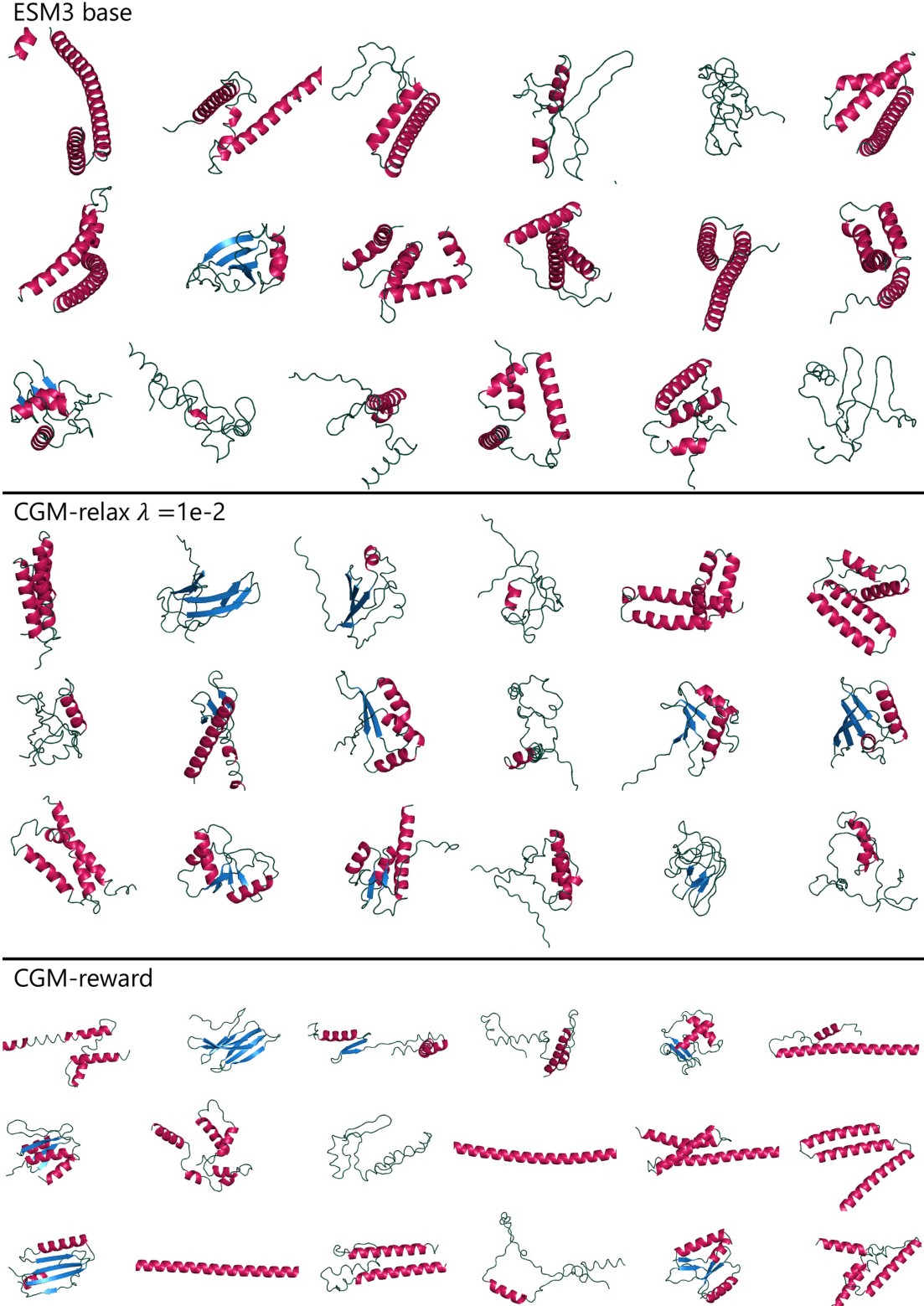

*Figure 10.* Random samples from the ESM3-open model before calibration (top), after calibration using CGM-relax with 99 bivariate CDF constraints (middle), and after calibration using CGM-reward with 15 bivariate CDF constraints (bottom).

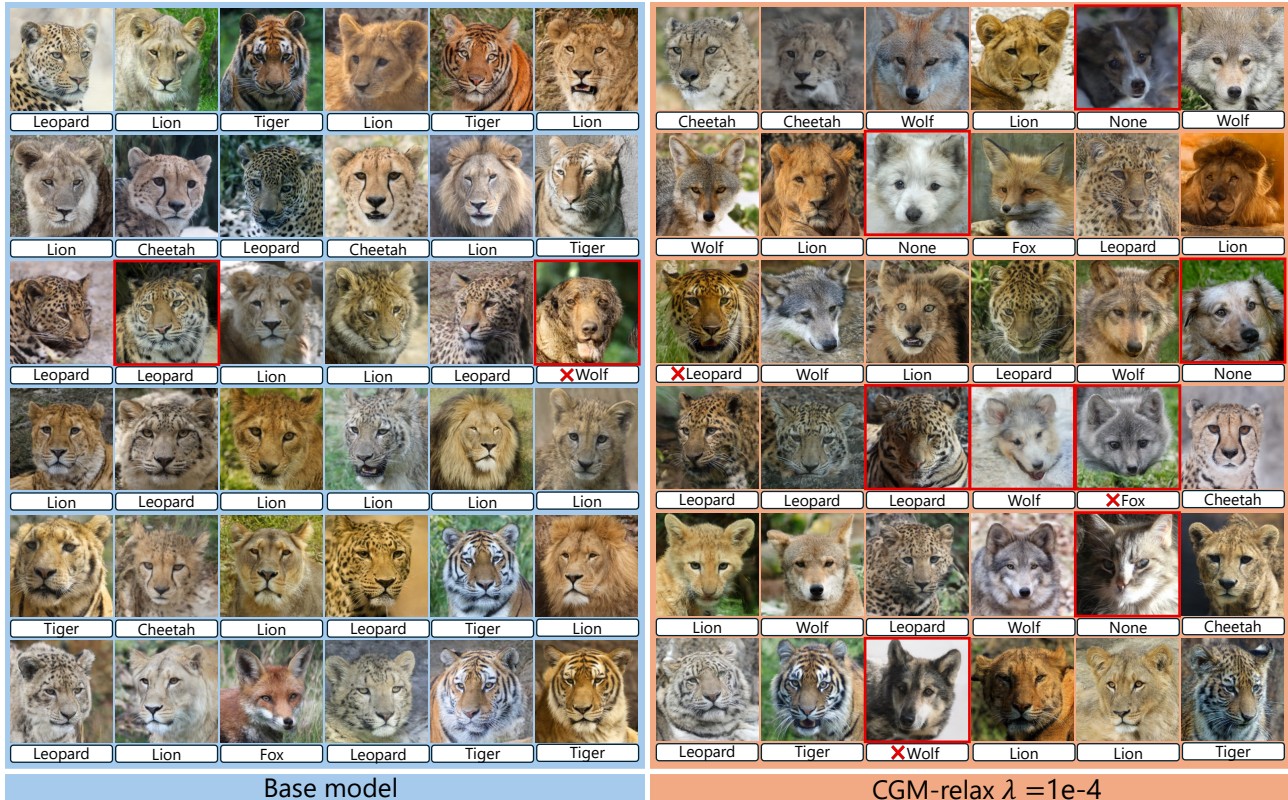

*Figure 11.* Random samples from the conditional TarFlow model trained on the AFHQ dataset (blue background) and the same model fine-tuned using CGM-relax ($\lambda$=$10^{-4}$) (orange background), with annotations by GPT o5-mini (white box). Red boxes denote poor-quality samples, and red crosses denote incorrect annotations. Although the model calibrated with CGM-relax produces animals with more balanced class proportions, it also produces fewer realistic samples.

### F.4. Calibrating TarFlow

As we described in Section 4.2, our goal when calibrating the TarFlow model (Zhai et al., 2025), trained conditionally on the Animal Faces HQ (AFHQ) dataset (Choi et al., 2020), is to generate more diverse samples from the wildlife class. By directly examining the AFHQ dataset, we identify six animals: {lion, tiger, wolf, fox, leopard, cheetah}; we do not further distinguish among these animals e.g., leopard versus snow leopard. Within the AFHQ training dataset, these animals are represented in the wildlife class with proportions {0.2615, 0.2254, 0.0897, 0.0933, 0.2003, 0.1290}, as annotated by GPT o5-mini.

Our motivation for choosing this problem was twofold. First, the quality of images generated by the base TarFlow model is high, such that a pre-trained classifier could attain high accuracy without fine-tuning. Second, we observe that the wildlife images generated by the base TarFlow model contained predominantly lions and leopards (Figure 11), and rarely contained foxes or wolves. From $5 \times 10^3$ samples annotated by GPT o5-mini, we computed animal proportions {0.3590, 0.1260, 0.0404, 0.0256, 0.2704, 0.1752}.

For image classification, we queried GPT o5-mini to classify each image according to the following prompt:

```
You are labeling animal photos.
Return JSON only: {"label": <one of the options>, "confidence": <0..1>}.
Choose exactly one from: lion, tiger, wolf, fox, leopard, cheetah or none.
```

Although we require the model to state its confidence when labeling the images, we do not use these confidence scores for fine-tuning. The calibration function $\boldsymbol{h}(\boldsymbol{x}) \in \mathbb{R}^5$ is a one-hot encoding of the first five classes. Since nearly all of the samples from the base TarFlow model are labeled as one of the six animals, we observe that choosing a six-dimensional constraint (i.e., adding cheetah) results in a poorly conditioned dual problem (7), since then the components of $\boldsymbol{h}$ are nearly linearly dependent. Our target is the uniform distribution over animals $\boldsymbol{h}^* = [.167, .167, .167, .167, .167]$.

We perform calibration for $\lambda \in \{10^{-2}, 10^{-3}, 10^{-4}, 10^{-5}\}$, and sample size $N = 5 \times 10^3$. We report results as the mean

over three replicates, with two standard errors. We assess the success of calibration using two metrics: the total variation (TV) distance of the distribution over animal proportions (using $5 \times 10^3$ annotated images) to the uniform distribution; and the FID, computed using only the samples belonging to the wildlife class in the AFHQ training dataset. We use $5 \times 10^4$ samples from the generative model to compute FID. It is important to note that the FID is an imperfect metric for assessing the quality of generated images since it will be lower for models whose animal class makeup is similar to that of the training distribution. To account for this, we evaluate CGM-relax on the maximum entropy reweighting of the training dataset to the uniform distribution over animal classes. In other words, we up or down weighted images belonging to a particular animal class in order to sample the six animals belonging to wildlife class with equal probability.

Our best model, calibrated using CGM-relax with $\lambda = 10^{-4}$, obtains class proportions $\{0.2248, 0.0854, 0.1750, 0.1566, 0.1668, 0.1086\}$, evaluated using $5 \times 10^3$ samples from the model; $0.0828$ of the samples were labeled as `None`. CGM-relax reduces the calibration error by nearly three times, from a TV distance of .306 to .101 (Figure 12). However, the FID score increases from 15.9 to 21.0. CGM-reward is unsuccessful at calibrating the base model; both miscalibration and FID is roughly the same as the base model. Since CGM-reward remains close to the base model, we evaluate FID on the original AFHQ training dataset.

In Figure 11, we provide random generations from both the pre-trained and the model calibrated with CGM-relax with $\lambda = 10^{-4}$. By examining samples from the calibrated model, we observe two axes along which sample quality worsens after calibration. First, some of the samples (those labeled as `None`) are dogs or cats, which lie outside the AFHQ wildlife class. Second, a greater proportion of samples depict blends of multiple animals.

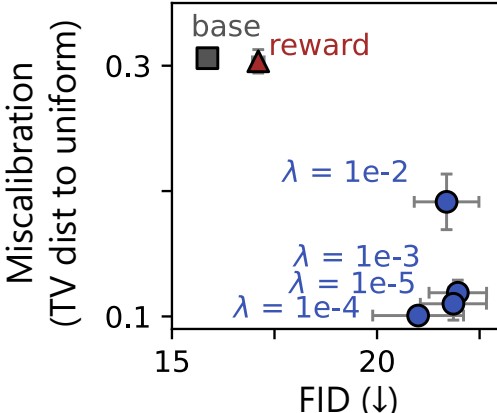

*Figure 12.* Calibrating TarFlow with CGM-relax reduces the TV distance of animal class labels to the uniform distribution by approximately three times. However, CGM-relax also produces fewer realistic samples, as measured by FID.

## F.5. Calibrating Gemma-2-9B-IT

### F.5.1. AUTOREGRESSIVE SAMPLING AND LOG-LIKELIHOODS

We sample in the standard autoregressive fashion with no temperature scaling. To compute sequence log-likelihoods, we consider the prompt as given and ignore tokens generated after the first end-of-sequence (EOS) token. Let $m$ be the length of the prompt and $n$ the index of the first EOS token. Then sequence $\boldsymbol{x}$ has log-probability

$$\log p_\theta(\boldsymbol{x}) = \sum_{i=m+1}^{n} \log p_\theta(\boldsymbol{x}(i) \mid \boldsymbol{x}(<i)).$$

For computational efficiency during training and evaluation, we set the maximum length of each story to be 200 tokens.

### F.5.2. PROMPT DEFINITION

We used the prompt "Write a short story (3-5 paragraphs) which only uses very simple words that a 3 year old child would likely understand. ONLY write the story without any additional text. Try to use characters with roughly EQUAL PROBABILITIES male or female. The story begins: "Once upon a time there was a <profession> named"", which is similar to the one used by Eldan & Li (2023) with the addition of the EQUAL PROBABILITIES sentence. As shown in Figure 13, we found that this addition decreased but did not eliminate gender imbalance compared to the original prompt.

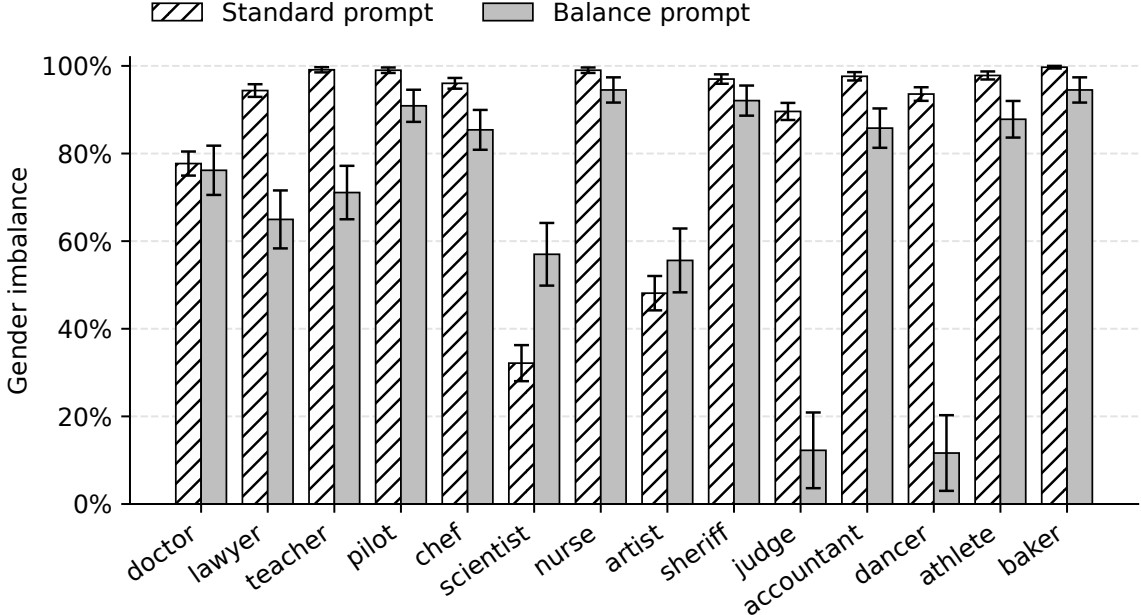

*Figure 13.* Gender imbalance for profession stories generated by Gemma-2-9B-IT with and without gender balance prompt. Values are computed using $2,048$ samples per profession. Error bars are two times standard error of the mean approximated using the binomial variance.

### F.5.3. GEMMA-2-9B-IT CONSTRAINT DEFINITION

To calibrate Gemma-2-9B-IT, we use a simple heuristic procedure to detect the gender of the story's character associated with the profession in the prompt. The procedure returns 1 for female, $-1$ for male, and 0 if the gender cannot be determined. Given a generated story, our procedure is as follows.

**(1) Pronoun at sentence two.** If the second sentence begins with a third-person singular pronoun, we assign gender based on that pronoun. This is common with our prompt template, e.g., "Once upon a time there was a doctor named Sam. **She**

was very kind...".

**(2) First-sentence scan.** If step (1) is inconclusive, we iterate through the words in the story's first sentence. If a title ("Mr.", "Mrs.", "Miss", "Ms.") appears, we assign the corresponding gender. Otherwise, we treat each word as a potential first name and query the `gender-guesser` package (Pérez et al., 2016). If the package classifies the token as "male", "mostly male", "female", or "mostly female", we assign the corresponding gender; otherwise we continue scanning.

**(3) No evidence.** If no gender is detected, we assign 0 (unknown).

We acknowledge the limitations of this simple approach but consider it sufficient for a proof of concept.

**Conditional constraint via sum-of-losses.** We wish to satisfy the conditional calibration constraints

$$\mathbb{E}[\boldsymbol{h}(\boldsymbol{x}) \mid \text{prompt}_i] = 0, \quad \text{for } i = 1, \dots, k$$

which encodes that the male and female labels should be balanced *for each* of the $k$ professions. We implement this as a sum of CGM losses $\sum_{i=1}^{k} \widehat{\mathcal{L}}_i$, where $\widehat{\mathcal{L}}_i$ is the reward or relax loss for the conditional generative model $p_\theta(\boldsymbol{x} \mid \text{prompt}_i)$. During every training batch, we sample 64 stories for each of the eight professions, resulting in a total batch size of 512.

## F.6. LoRA details

We used LoRA with rank 32, scaling factor $\alpha = 64$ and no LoRA dropout. LoRA was applied to the self-attention key, query, value, and output layers and to the MLP (gate, up, down) projection layers, resulting in $1.1 \times 10^8$ trainable parameters.

### F.6.1. GEMMA-2-9B-IT FIGURE DETAILS

Both panels of Figure 6 were created using five replicates per model with different Pytorch seeds, with points indicating the mean metric value across replicates. 2048 stories were sampled per profession for each replicate, resulting in $14 \times 2048 = 28672$ stories per model. Due to low variance, two-times-standard-error-of-the-mean error bars are smaller than most markers, so are not shown.

**Khalifa et al. (2021) baseline.** Khalifa et al. (2021) use the same method as CGM-reward to define an approximate target distribution $p_{\widehat{\boldsymbol{\alpha}}_N}$. Unlike CGM-reward, they minimize the forward KL

$$\begin{aligned}
D_{\text{KL}}\left(p_{\widehat{\boldsymbol{\alpha}}_N} \parallel p_\theta\right) &= \mathbb{E}_{p_{\widehat{\boldsymbol{\alpha}}_N}}\left[\log \frac{p_{\widehat{\boldsymbol{\alpha}}_N}(\boldsymbol{x})}{p_\theta(\boldsymbol{x})}\right] \\
&= \mathbb{E}_{p_{\text{stop-grad}(\theta)}}\left[\frac{p_{\widehat{\boldsymbol{\alpha}}_N}(\boldsymbol{x})}{p_{\text{stop-grad}(\theta)}(\boldsymbol{x})} \log \frac{p_{\widehat{\boldsymbol{\alpha}}_N}(\boldsymbol{x})}{p_\theta(\boldsymbol{x})}\right] \quad \text{(change of measure)} \\
&= \mathbb{E}_{p_{\text{stop-grad}(\theta)}}\left[-\frac{p_{\widehat{\boldsymbol{\alpha}}_N}(\boldsymbol{x})}{p_{\text{stop-grad}(\theta)}(\boldsymbol{x})} \log p_\theta(\boldsymbol{x})\right] + C \\
&= \mathbb{E}_{p_{\text{stop-grad}(\theta)}}\left[-\frac{p_{\theta_{\text{base}}}(\boldsymbol{x}) \exp\left\{\widehat{\boldsymbol{\alpha}}_N^\top \boldsymbol{x} - A_{p_{\theta_{\text{base}}}}(\widehat{\boldsymbol{\alpha}}_N)\right\}}{p_{\text{stop-grad}(\theta)}(\boldsymbol{x})} \log p_\theta(\boldsymbol{x})\right] + C \quad \text{(definition of } p_{\widehat{\boldsymbol{\alpha}}_N}) \\
&= K\mathbb{E}_{p_{\text{stop-grad}(\theta)}}\left[-\frac{p_{\theta_{\text{base}}}(\boldsymbol{x}) \exp\left\{\widehat{\boldsymbol{\alpha}}_N^\top \boldsymbol{x}\right\}}{p_{\text{stop-grad}(\theta)}(\boldsymbol{x})} \log p_\theta(\boldsymbol{x})\right] + C,
\end{aligned}$$

where $C$ and $K$ are constants that do not depend on $\theta$. $C$ can be ignored as it has no affect on parameter gradients, and $K$ can be absorbed into the learning rate. Similar to CGM-reward, gradients of this KL-divergence are estimated using Monte Carlo. For a fair comparison, we use the same $\widehat{\boldsymbol{\alpha}}_N$ and batch size to train Khalifa et al. (2021), CGM-reward, and CGM-relax.

**Distance from base-model (symmetrized KL) definition.** For each fine-tuned model, we sample $N = 2048$ stories $\{\boldsymbol{x}_i\}_{i=1}^N$ per profession, and compute log-probabilities $\log p_\theta(\boldsymbol{x}_i)$ and base-model log-probabilities $\log p_{\theta_{\text{base}}}(\boldsymbol{x}_i)$. We estimate the per-profession backward KL as

$$D_{\text{KL}}\left(p_\theta \parallel p_{\theta_{\text{base}}}\right) \approx \frac{1}{N} \sum_{i=1}^N \log \frac{p_\theta(\boldsymbol{x}_i)}{p_{\theta_{\text{base}}}(\boldsymbol{x}_i)}.$$

The forward KL uses importance sampling estimate

$$\mathrm{D}_{\mathrm{KL}}\left(p_{\theta_{\mathrm{base}}} \parallel p_{\theta}\right) \approx \frac{1}{N} \sum_{i=1}^{N} \frac{p_{\theta_{\mathrm{base}}}(\boldsymbol{x}_i)}{p_{\theta}(\boldsymbol{x}_i)} \log \frac{p_{\theta_{\mathrm{base}}}(\boldsymbol{x}_i)}{p_{\theta}(\boldsymbol{x}_i)}.$$

We add our estimates for the forward and backward KL for each profession to get the symmetrized KL, then report the average symmetrized KL across all eight professions.

**Gender imbalance definition.** For each model replicate, we compute the number of male (#male) and number of female (#female) characters in 2048 samples for each profession. The miscalibration for a single profession is defined as

$$\left| \frac{\#\mathrm{male} - \#\mathrm{female}}{\#\mathrm{male} + \#\mathrm{female}} \right|,$$

which takes maximum value 1 if all samples used the same gender and minimum value 0 if there are an equal number of each gender. The overall miscalibration values shown on the y-axis of Figure 6A were computed by taking the average miscalibration for the eight professions used for fine-tuning.

**Estimating KL for the max-entropy solution.** Using $N = 2048$ samples from the base-model for each profession, we compute an estimate $\widehat{\boldsymbol{\alpha}}$ of $\boldsymbol{\alpha}^*$ for each profession using the procedure outlined in Appendix C.2. We then compute importance weights for each sample $w_i = \exp(\widehat{\boldsymbol{\alpha}}^\top \boldsymbol{x}_i)$, which we use to compute normalized weights $\widetilde{w}_i = w_i / \sum_{j=1}^{N} w_j$. To estimate the forward and backward KL divergences, we treat base-model probabilities as $1/N$ and use

$$\mathrm{D}_{\mathrm{KL}}\left(p_{\theta_{\mathrm{base}}} \parallel p_{\widehat{\boldsymbol{\alpha}}}\right) \approx \frac{1}{N} \sum_{i=1}^{N} \log \frac{1/N}{\widetilde{w}_i}, \qquad \mathrm{D}_{\mathrm{KL}}\left(p_{\widehat{\boldsymbol{\alpha}}} \parallel p_{\theta_{\mathrm{base}}}\right) \approx \frac{1}{N} \sum_{i=1}^{N} \frac{\widetilde{w}_i}{1/N} \log \frac{\widetilde{w}_i}{1/N}.$$

This procedure was repeated five times for different sampling seeds to ensure variance from estimating $\boldsymbol{\alpha}$ had little effect on the outcome. The resulting error bars for the symmetrized KL are smaller than the marker size.

### F.6.2. EXAMPLE GENERATIONS

We provide example generations for four prompts before and after fine-tuning with CGM-relax with $\lambda = 0.1$.

*Listing 1.* Samples from Gemma-2-9B-IT

```
Once upon a time there was a doctor named Ben. Ben had a big car. Ben had a red hat. He drove to the park. The park had a swing. A girl
    was on the swing. The girl had a red ball. The girl threw the ball. The ball went far. Ben smiled. Ben waved. The girl waved
    back. Ben went home.

Once upon a time there was a lawyer named Ben. Ben liked to wear a big tie. Ben went to work in a tall building. He saw lots of people
    in the big building. Ben sat at a big desk. He read papers all day. One day a girl named Sue came to see Ben. Sue had a puppy.
    Sue's puppy was lost. Sue was sad. Ben helped Sue. He said, "Don't worry, Sue. We will find your puppy." Ben looked all over the
    big building. He asked all the people. Finally, Ben found the puppy under a chair. The puppy was sleepy. Ben gave the puppy
    back to Sue. Sue was happy. Ben was happy.

Once upon a time there was a teacher named Mary. Mary had many students. They liked to play games. One day, Mary had a new game. It was
    a red ball. Mary said, "Roll the ball!" The students rolled the ball. It was fun! They rolled the ball to each other. They
    laughed so much. They played with the ball all day. At the end of the day, Mary said, "Time to clean up." The students helped
    Mary put the ball away. They said, "Thank you, Mary!" They liked playing games with her. Mary waved goodbye. She saw the students
    go home. Mary was happy. She liked teaching the students.

Once upon a time there was a pilot named Ben. He had a big red plane. Ben loved to fly. He would fly high in the sky. He saw birds and
    clouds. Once, Ben saw a big, shiny star. It was yellow and so bright. Ben smiled. He liked his plane. He liked to fly. One day,
    Ben went to the park. He saw a little girl. Her name was Lily. Lily looked at the sky. "I want to fly!" she said. Ben smiled. "
    You can fly in my plane!" he said. Lily was so happy. She got in the plane. Ben took off. Lily saw the clouds up close. They
    were white and fluffy. She laughed. Ben flew Lily high, high in the sky. Lily saw trees and cars. They looked small from up high.
    Ben and Lily flew for a long time. Then, Ben landed the plane. Lily hopped out. "Thank you!" she
```

*Listing 2.* Samples from CGM-relax ($\lambda = 0.1$) model

```
nce upon a time there was a doctor named Sam. Sam was nice. Sam had a toy red car. Sam liked to play with the car. He played with it
    all day. One day, Ben knocked on Sam's door. "Come in!" said Sam. Ben was sad. "Zoom zoom!" Ben saw the car. Ben liked cars too.
    "Can I play?" asked Ben. Sam said yes. They played with the car. They went zoom zoom all over the room. They played until it was
    time for Ben to go home. Ben said "Bye Sam! Thank you!" Ben ran home. Sam was happy. Sam liked to make friends happy.

Once upon a time there was a lawyer named Sam. Sam had a big, red car. He liked to drive fast. He went to work in his car every day.
    At work, Sam helped people. He talked to them. He wrote things down. He made plans. People liked Sam. He was nice. He made them
    feel better. Sam liked his job. One day, Sam saw a dog on the street. The dog was small and sad. Sam stopped his car. He got out
    and looked at the dog. Sam gave the dog a cookie. The dog wagged its tail. Sam smiled. He took the dog home with him. He named
    the dog Spot. Sam and Spot were happy.
```

Once upon a time there was a teacher named Ray. Ray had lots of toys. Ray liked to play with the toys. One day, a girl named Lily came to Ray's room. Lily wanted to play, too. Ray said, "Come play!" Ray got a big red ball. Lily liked the red ball. They played with the ball all day. They rolled it and threw it. They had lots of fun. After they played, Lily said, "Thank you, Ray!" Ray said, "You're welcome, Lily!" Lily went home. Ray put the toys back. Then Ray sat down and read a book. The end.

Once upon a time there was a pilot named Mia. Mia flew a big plane. She flew high in the sky. Mia saw a bird. The bird was small. Mia waved to the bird. The bird flew away. Mia saw a cloud. The cloud was white. Mia touched the cloud. The cloud was soft. Mia flew back home. Mia's dog was happy. Mia gave her dog a hug. Mia's cat was sad. Mia gave her cat a pet. " Mia liked to fly. She liked her dog and cat too.

