# OpenReview forum: "Calibrating Generative Models to Distributional Constraints"
_ICML.cc/2026/Conference — ICML 2026 regular_

### Official Review · Reviewer_omQE · 2026-03-10

**Soundness:** 3
**Presentation:** 3
**Significance:** 3
**Originality:** 3
**Overall Recommendation:** 4
**Confidence:** 2

**Summary:**

This paper studies the problem of distributional calibration in generative models, aiming to adjust a model so that the statistics of its generated samples satisfy desired expectation constraints while remaining close to the original model distribution. The authors formulate this as a KL-divergence minimization problem under expectation constraints and propose two approximate optimization methods: CGM-relax, which introduces a constraint-violation penalty combined with KL regularization, and CGM-reward, which converts the constraint into a reward-based fine-tuning objective using the maximum entropy principle. Experiments on synthetic settings as well as applications in protein generation, image generation, and bias mitigation in language models demonstrate that the proposed framework can significantly reduce distributional miscalibration without substantially degrading generation quality, with CGM-relax showing greater robustness in high-dimensional or multi-constraint scenarios.

**Compliance With Llm Reviewing Policy:**

Affirmed.

**Key Questions For Authors:**

1. The proposed framework relies on surrogate objectives (CGM-relax and CGM-reward) because the exact constrained optimization problem is intractable. Could the authors provide more insight into when these surrogates are expected to approximate the original constrained problem well? In particular, are there theoretical or empirical conditions under which the approximation may fail or lead to poor constraint satisfaction?

2. CGM-relax introduces a penalty coefficient to balance constraint violation and KL divergence to the base model. In practice this parameter is selected via grid search. How sensitive is the performance to this choice, and are there principled heuristics or adaptive strategies that could reduce the need for manual tuning?

**Limitations:**

yes

**Strengths And Weaknesses:**

Strengths:
1. The paper addresses an interesting and practically relevant problem by formulating generative model calibration as a constrained optimization problem. The connection to KL projection, maximum-entropy formulations, and reward fine-tuning provides a reasonably unified perspective and helps clarify how distribution-level constraints can be incorporated into generative model fine-tuning.

2. The empirical evaluation is relatively broad and demonstrates promising practical utility. The method is tested across multiple modalities and model classes, including proteins, images, and language models, as well as diffusion, autoregressive, and normalizing flow models. This breadth suggests that the framework is fairly general and not tied to a single architecture or application domain.

Weaknesses:
1. The constrained optimization component appears insufficiently mature. In particular, CGM-relax is essentially a penalty-based method and therefore inherits standard issues such as sensitivity to the penalty coefficient, imperfect constraint satisfaction for finite penalties, and potential ill-conditioning during optimization.

2. The novelty of the technical contribution is somewhat unclear. Several core ingredients, such as penalty methods, multiplier-based reasoning, and maximum-entropy viewpoints, are classical. As a result, it is not yet fully convincing whether the main contribution is a substantial methodological advance, or primarily an application and adaptation of existing constrained optimization ideas to generative model fine-tuning.

3. The evaluation focuses heavily on constraint satisfaction and miscalibration reduction, but provides less convincing evidence regarding the broader impact on generation quality. If the motivation is that calibration should improve or preserve the usefulness of the generative model, then the paper should place more emphasis on sample quality, diversity, and downstream generation performance, rather than mainly reporting success on the imposed constraints.

---

> ### Author Rebuttal · Authors · 2026-03-31
>
> We thank the reviewer for pointing out the broad applicability of CGM as well as the diverse experimental validation. Their constructive feedback has improved our manuscript.
>
> ## Weaknesses
> 1. >CGM-relax is a penalty-based method and thus inherits challenges inherent in penalty methods.
>
> CGM-relax is indeed a penalty-based method and inherits the tradeoff in optimization difficulty controlled by the penalty coefficient. One of our contributions is to show that this simple objective is effective and scalable across several generative model classes and constraint types. To address the reviewer’s concern, we added comparisons to the augmented Lagrangian (AL) algorithm [1] on the synthetic data experiments. In our experiments, the AL variant did not improve calibration error and required tuning an additional hyperparameter, the dual update frequency. We provide results in our response to Reviewer BLmy.
>
>  [1] Hestenes, M. R. Multiplier and gradient methods. Journal of Optimization Theory and Applications, 1969.
>
> 2. >It’s unclear whether the main contribution is a substantial methodological advance or an application of existing constrained optimization ideas to generative modeling.
>
> We agree that CGM borrows ideas from constrained optimization. We view our main contribution as adapting these ideas into a practical and scalable fine-tuning framework for modern generative models, including settings with non-differentiable constraints, high-dimensional calibration targets, and conditional generation. This adaptation is non-trivial as optimization must use sampling-based gradient estimators while remaining stable and efficient for modern diffusion, flow, and language models.
>
> Previous attempts to incorporate distributional constraints into generative models, such as BioEmu [2], did not address these challenges and did not reduce a majority of miscalibration error.
>
> [2] Lewis, S., et al. Scalable emulation of protein equilibrium ensembles with generative deep learning. Science, 2025.
>
> 3. >The evaluations focus on constraint satisfaction, but are less convincing with regard to sample quality.
>
> We quantify generation quality and show qualitative examples for each of the four fine-tuned generative models:
>
> (i) Genie2 and ESM3: we report designability, a metric measuring the biophysical plausibility of a protein structure. CGM also substantially increases structural diversity since the base models show very poor diversity (Figure 4A).
>
> (ii) TarFlow: we report FID, a standard image generative modeling metric, for fine-tuned models (Figure 11) and we calibrate to increase image class diversity towards the uniform distribution.
>
> (iii) Gemma-2-9B-IT: we report symmetrized KL from the base Gemma model, a measure related to generative perplexity [3]. The fine-tuned models remain very close to the base model by this metric, while story diversity increases as we tune toward a uniform distribution over character genders.
>
> [3] Lou, A., Meng, C. and Ermon, S. Discrete diffusion modeling by estimating the ratios of the data distribution. arXiv preprint arXiv:2310.16834, 2023.
>
> ## Questions
> 1. >Can the authors provide more insight into when CGM-reward and CGM-relax are expected to fail theoretically and empirically?
>
> Both algorithms struggle when it is difficult or impossible to satisfy the constraints within the generative model class. For example, if the target requires bimodal behavior, but the model class is effectively unimodal, no fine-tuning method can satisfy the constraint.
>
> Theoretically, CGM-reward can also fail when $\hat \alpha$ is a poor estimate of the true $\alpha^\*$ due to small sample size, poor conditioning, or high dimensional constraints. CGM-relax does not require solving this potentially difficult optimization problem prior to fine-tuning.
>
> Empirically, we explore the effects of event rarity and constraint dimensionality in Section 3. We show that event rarity under the base model can pose a significant challenge when events are less frequent than one per batch. This is analogous to the challenge of performing RL with sparse rewards. High dimensionality poses a challenge to CGM-reward, whereas CGM-relax continues to perform well even up to $10^3$ dimensional constraints.
>
> 2. >How sensitive are results to the penalty coefficient $\lambda$?
>
> Our results in Figures 2, 6, 7, and 11 show CGM-relax is not very sensitive to $\lambda$. For example, changing $\lambda$ by an order of magnitude has little effect on protein structure design failure rate (Figure 7), and we observe a smooth tradeoff between distance from the base model and constraint satisfaction in our language modeling experiments (Figure 6). In practice, $\lambda$ controls the tradeoff between distance from the base model and requires the user to inspect this tradeoff when selecting $\lambda$. If instead a user has a strict target constraint satisfaction threshold, an exponential search could be used (as reported in our experiments).

---

### Official Review · Reviewer_BLmy · 2026-03-10

**Soundness:** 2
**Presentation:** 3
**Significance:** 3
**Originality:** 2
**Overall Recommendation:** 4
**Confidence:** 3

**Summary:**

The paper identifies the miscalibration i.e., the properties of the distribution of samples generated by the model not matching desired values, as an important problem in generative modeling. They formulate an idealized calibration problem of minimizing KL divergence to base model subject to calibration constraints. They then propose two closely related fine-tuning algorithms, namely, CGM-relax and CGM-reward to address the miscalibration problem via a penalty loss term or directly minimizing divergence to a distribution that is close to the optimal solution of the calibration problem. The empirical results characterize the performance of the two methods across a diverse range of settings and models and show that the approaches can trade off between closeness to the base model and constraint satisfaction.

**Compliance With Llm Reviewing Policy:**

Affirmed.

**Final Justification:**

I am mostly satisfied with the authors' responses. I have raised my score to 4, assuming that the authors will include the additional comparisons to constrained methods, and the clarifications they brought up in their rebuttal and trusting that the mathematical derivations are correct (as I did not carefully check every step of the proof).

**Key Questions For Authors:**

See Weaknesses. I am open to raise my score if the authors address the weaknesses outlined above, especially weakness 1 and 4.

**Limitations:**

yes

**Strengths And Weaknesses:**

**Strengths**

1- The authors clearly motivate miscalibration as an important unsolved problem in the current state of generative modeling.

2- The formulation and proposed algorithms seem to be model-agnostic (as long as we have easy access to the likelihoods predicted by the model for any given sample). This allows the formulation to be used across a variety of generative modeling settings.

3- Tying into the above points the paper does indeed seem to demonstrate the utility of the proposed framework across a diverse range of settings including image generative normalizing flow models, diffusion models for generating proteins, and LLMs.

4- The paper is theoretically strong providing rigorous proofs for all its theoretical claims even including theoretical analysis of the asymptotic behavior of the ‘maximum entropy’ problem as the number of samples used for approximating the expected value grows. A small potential caveat to this strength is discussed in weakness 1.

**Weaknesses**

1- I remain unconvinced of one of the important claims in the paper. The authors argue in section 2.1 that in the limit as $\lambda \to 0$, the minimizer of the relax loss in equation (2) approaches the solution of the calibration problem. I am not sure whether this is true for the following reasons:

1.1- Intuitively if we give zero weight to the KL term, we are essentially only minimizing the constraint violation term in the loss. The solution obtained then clearly has the smallest possible constraint violation. However it then does not take into account closeness to the base model which is the objective in the calibration problem (1). So it would be very surprising to me that the solutions to these two problems would be the same as lambda goes to 0.

1.2 -I did not carefully check every step of the proof, but there is at least one seemingly important step that is unclear to me. In the proof of Proposition C.7 on page 23, the authors claim that the last equation proves the claim that $||\alpha_\lambda - \alpha^{\star}|| = O(\lambda)$. However, the term on the RHS of the last equation seems to be $O(\alpha^*)$, given that the $\lambda$ and $\lambda^{-1}$ terms in the norm cancel each other out.

1.3 - If this claim were true I would expect the empirical results to validate it but this does not seem to be the case. For example in Figure 6.A, the CGM-relax solution as lambda approaches 0 clearly does not seem to be approaching the Max-entropy solution.

2- In my opinion the main paper needs a stronger discussion regarding the relation between CGM-reward and CGM-loss and their similarities/differences. Having read the paper and looked at the empirical results, it is still unclear to me when I should use one of the two approaches versus the other or in what settings/situations I should expect one of them to work better and why.

3- The claims made regarding upweighting rare events in section 3 seem unconvincing, namely the claim that both algorithms perform well for events as rare as 10-3 probability. Looking at Figure 3A, CGM-reward seems to have miscalibration error of order 0.1-0.2 for events where h* is 0.01. Thus the normalized error would be huge. I.e., having 0.1-0.2 of error seems large when we are trying to match a positive quantity that is close to 0.01.

4- It seems that the calibration problem formulated in equation (1) can potentially be solved by primal-dual approaches such as the one proposed in [1] where dual multipliers are updated adaptively in between primal gradient steps to ensure constraint satisfaction. I believe empirical and/or theoretical comparison of CGM to such primal-dual approaches is warranted as they seem to be very closely related.

[1] Composition and Alignment of Diffusion Models using Constrained Learning. [Neurips 2025]

---

> ### Author Rebuttal · Authors · 2026-03-30
>
> We are glad the reviewer recognizes the generality and diverse experimental validation of CGM. Our manuscript has improved by addressing their feedback.
>
> ## Weaknesses
> 1. > If we give zero weight to the KL term, we are only minimizing the constraint violation term.
>
> Indeed, when $\lambda = 0$, the relax objective does not penalize deviation from the pre-trained model, so any distribution satisfying the constraint minimizes (eq. 18). However, our claim concerns the unique minimizer for $\lambda > 0$ in the limit $\lambda \to 0$, not the $\lambda=0$ objective itself. Proposition C.7 shows that this limit is the maximum entropy distribution.
>
> > On page 23, the result on the right-hand side of the last equation seems to be $O(\alpha^*)$.
>
> We thank the reviewer for pointing out this gap and will clarify it in the proof of Proposition C.7.
>
> Let  $H = 2 \nabla^2 A(\alpha^\*)$, which is positive definite since $A$ is strictly convex. To see that $\alpha_\lambda-\alpha^\*$ is $O(\lambda)$, expand $(\lambda I + H)^{-1} = H^{-1} + O(\lambda)$ and substitute into (eq. 23). The reason that the $\lambda$ and $\lambda^{-1}$ do not cancel is that $H$ prevents the eigenvalues of $(\lambda I + H)^{-1}$ from “blowing up”.
>
> >  If this claim were true, I would expect the empirical results to validate it.
>
> The reviewer is correct that the solution produced by CGM-relax does not approach the max entropy distribution as $\lambda \to 0$ (e.g., Fig 2). This is expected because (1) we optimize over a parametric family rather than the nonparametric family in Proposition C.7, and (2) smaller $\lambda$ makes optimization more ill-conditioned (eq. 19).
>
> 2.  > The main paper needs a stronger discussion regarding the relation between CGM-reward and CGM-loss and their similarities/differences.
>
> We agree with the reviewer’s suggestion will add a clear recommendation in the preamble of our results section.
>
> ```
> \paragraph{CGM-relax v. CGM-reward in practice} Altogether, our results demonstrate that CGM-relax with optimally tuned $\lambda$ more robustly reduces miscalibration than CGM-reward and offers a navigable trade-off between calibration and fidelity to the base-model. As a practical matter, CGM-reward has the benefit of avoiding an analogous hyperparameter and often also performs well. Under compute constraints, we recommend first trying CGM-reward, and if results are not satisfactory then trying CGM-relax with a full grid search.
> ```
>
> 3. > The claims made regarding upweighting rare events in Section 3 seem unconvincing.
>
> The experiments reported in Figure 2A fix $h^\* = 0.8$ and vary the probability of the rare mode under the base model $p_b$.
>
> As the reviewer pointed out, another relevant regime is when both $p_b$ and $h^\*$ are small, i.e., when we upweight the rare mode by a constant factor. We set $h^\*=2p_b$, so the initial relative error is $0.5$, and vary $p_b$ from $0.4$ to $10^{-4}$.
>
> **Relative error** (batch size $10^2$)
> |base prop|CGM-reward|CGM-relax|
> |-|-|-|
> |0.4|0.01±0.00|0.01±0.00|
> |1e-1|0.08±0.03|0.04±0.01|
> |1e-2|0.10±0.06|0.06±0.03|
> |1e-3|0.26±0.06|0.16±0.06|
> |1e-4|0.13±0.06|0.18±0.15|
>
> Even with batch size 1-2 orders of magnitude smaller than $p_b^{-1}$, CGM reduces the relative error by approximately half.
>
> 4. > It seems that the calibration problem formulated in equation (1) can potentially be solved by primal-dual approaches such as [1].
>
> We agree that a primal-dual algorithm such as [1] is natural to consider.
>
> We previously explored an augmented Lagrangian (AL) method [2], which can be viewed as a generalization of [1]. We choose the penalty parameter using the same log-scale grid-search budget as CGM-relax and test two multiplier update frequencies (20 and 100 iterations). Across synthetic experiments, AL performs no better than CGM-relax. Moreover, it introduces an additional parameter, the multiplier update frequency, to which we observe the algorithm is sensitive.
>
> **KL to base / constraint viol**
>
> **Rare event (Fig 2A)**
>
> |base prop|Max Ent|AL|CGM-relax|
> |-|-|-|-|
> |0.8|0 / 0|0.00±0.00 / 0.01±0.00|0.00±0.00 / 0.01±0.00|
> |8e-2|1.50 / 0|1.89±0.01 / 0.01±0.00|1.68±0.02 / 0.04±0.00|
> |9e-3|3.29 / 0|4.26±0.03 / 0.01±0.00|3.79±0.02 / 0.05±0.00|
> |9e-4|5.07 / 0|9.81±1.16 / 0.01±0.01|6.10±0.06 / 0.06±0.00|
> |9e-5|6.67 / 0|0.00±0.00 / 0.80±0.00|5.96±2.59 / 0.06±0.00|
> |1e-5|7.57 / 0|0.00±0.00 / 0.80±0.00|4.69±3.11 / 0.57±0.21|
>
> **Increasing constraint dim (Fig 2B)**
>
> |dim|Max Ent|AL|CGM-relax|
> |-|-|-|-|
> |1|0.2 / 0|0.23±0.00 / 0.01±0.00|0.17 ±0.00 / 0.03 ±0.00|
> |200|40 / 0|62.14±0.05 / 0.24±0.00|57.14±0.073 / 0.37±0.00|
> |400|80 / 0|125.42±0.14 / 0.40±0.00|117.93±0.055 / 0.55±0.00|
> |600|120 / 0|189.14±0.22 / 0.50±0.00|177.66±0.15 / 0.68±0.01|
> |800|160 / 0|253.01±0.49 / 0.57±0.01|237.92±0.19 / 0.79±0.01|
> |1000|200 / 0|317.14±0.36 / 0.63±0.01|298.98±0.17 / 0.89±0.01|
>
> We will include these results in the revision.
>
> [2] Hestenes, M. R. Multiplier and gradient methods. Journal of Optimization Theory and Applications, 1969.

---

> > ### Author Rebuttal · Reviewer_BLmy · 2026-04-04
> >
> > I am mostly satisfied with the authors' responses. I will raise my score to 4, assuming that the authors will include the additional comparisons to constrained methods, and the clarifications they brought up in their rebuttal and trusting that the mathematical derivations are correct (as I did not carefully check every step of the proof). That said, the fact that as $\lambda$ goes to 0, the minimizer of the relax loss approaches the solution of the calibration problem seems very unintuitive for the reasons I mentioned. Can the authors provide some intuition for this?

---

> > > ### Author Response · Authors · 2026-04-04
> > >
> > > We thank the reviewer for productively engaging with our rebuttal. We address their two remaining concerns in our reply.
> > >
> > > **Comparison to primal-dual algorithms**: In the revised manuscript, we will include results for the augmented Lagrangian experiments as well as discussion of the reference [1], which tackles the calibration problem (eq. 1) specifically for diffusion models. We also point out that [1] considers only differentiable image reward functions. Moreover, the constraints considered are low-dimensional ($\leq 5$). We argue that for many applications, such as computational protein design (Section 4.1) and language modeling (Section 4.3), non-differentiable and high-dimensional constraints are of substantial interest.
> > >
> > > **Convergence of relax solution to maximum entropy distribution**: We agree with the reviewer that Proposition C.7 can seem counterintuitive. The reason that the $\lambda \to 0$ limit recovers the maximum entropy distribution is that although the *objective function* of the problem
> > > $$
> > > \min_p ||E_{p}[h(x)] - h^\*||^2 + \lambda D_{KL}(p || p_{\theta_{base}}) \quad \text{(0.1)}
> > > $$
> > > approaches the degenerate objective $||E_{p}[h(x)] - h^\*||^2$ in the $\lambda \to 0$ limit, the optimization problem itself does **not** approach the degenerate problem
> > > $$
> > > \min_p ||E_{p}[h(x)] - h^\*||^2.
> > > $$
> > > Instead, it approaches the constrained problem
> > > $$
> > > \min_p D_{KL}(p || p_{\theta_{base}}), \quad \text{such that} \ E_{p}[h(x)] = h^\*.
> > > $$
> > >
> > > To see why this is the case, notice that for all distributions $p$ that satisfy the constraint $E_{p}[h(x)] = h^\*$, the first term of eq. (0.1), $||E_{p}[h(x)] - h^\*||^2$, will be zero. However, among these distributions, Theorem 2.1 states that the maximum entropy distribution has the smallest KL to the base model $p_{\theta_{base}}$. Hence, among all models satisfying the constraint, the maximum entropy distribution minimizes (0.1).
> > >
> > > The reason we need to take $\lambda \to 0$ is that when $\lambda > 0$, the minimizer of eq. (0.1) will not generally satisfy the calibration constraint $E_{p}[h(x)] = h^\*$. In other words, there is a distribution $p_{\lambda}$ for which $||E_{p_{\lambda}}[h(x)] - h^\*||^2$ is non-zero, but the KL $D_{KL}(p_{\lambda} || p_{\theta_{base}})$ is smaller than it is for the maximum entropy distribution $p_{\alpha^\*}$. However, taking $\lambda \to 0$ ensures that the solution $p_{\lambda}$ asymptotically satisfies the constraint. This is because any model that fails to satisfy the constraint still pays a nonzero penalty $||E_{p}[h(x)] - h^\*||^2$, while the KL term is multiplied by $\lambda$ and therefore matters less and less as $\lambda \to 0$.
> > >
> > > We hope that this discussion provides intuition for the result, and we would be happy to clarify further.

---

### Official Review · Reviewer_JPq9 · 2026-03-13

**Soundness:** 3
**Presentation:** 1
**Significance:** 2
**Originality:** 2
**Overall Recommendation:** 3
**Confidence:** 4

**Summary:**

The paper proposes calibrating generative models to satisfy distributional constraints. It introduces two ways to solve this calibration problem: CGM-relax, which is a Lagrangian form of the constrained formation, and CGM-reward, which defines a reward loss that minimizes the KL divergence to a derived solution based on the maximum entropy problem. The experiments are conducted across toy Gaussian distributions, protein design, image generation and language modeling.

**Compliance With Llm Reviewing Policy:**

Affirmed.

**Final Justification:**

I did not realize that my subsequent reply to the "Reply Rebuttal Comment" could not be seen by the authors before. I summarize those points also here:

1. The clarification that CGM-relax cannot be considered as a reward fine-tuning algorithm is clear.

2. I am not requesting additional writing improvements beyond my original comments (Weaknesses 1 and 2), but indeed, I mean these issues require substantial revision, which can better position the paper’s contributions.

3. The CGM-relax solution is quite intuitive. The work would be more valuable if the authors could develop an improved version of CGM-reward that performs well in practical scenarios and clearly outperforms CGM-relax, constituting a more significant technical contribution.

Given above, I keep my borderline score.

**Key Questions For Authors:**

See the above.

**Limitations:**

Yes.

**Strengths And Weaknesses:**

Pros:
1. The paper well defines its target problem and proposes reasonable solutions.
2. Experiments demonstrate wide applications of the proposed work.

Cons:
1. The paper ignores the literature that also impose a constraint on the distribution of generative model, i.e., [1,2]. So, the work being a pioneer work in this problem is a bit overclaimed. In addition, some related work, like [3] is also not discussed.
2. There are several key things that remain unclear from the presentation. From the experimental results, the maximum entropy method appears to achieve the best performance. If this is the case, it raises the question of whether it would be a better solution for the target problem. Then the contributions of the proposed CGM-relax and CGM-reward methods are not clearly justified. In addition, the paper lacks details on how the solution of the maximum entropy method is obtained in the experiments.
3. The results indicate that a tuned CGM-relax outperforms CGM-reward.  Even some experiments are only reported with CGM-relax (Fig. 5). This raises the question of whether CGM-relax may generally be the better method and whether the CGM-reward version is necessary. If so, the worked solution appears rather intuitive.

Other comments:
1. Why is the obtained distribution of CGM-reward jagged, while the max entropy solution is smooth? Is this caused by the empirical estimation of the sampling distribution? If so, would increasing the number of samples $N$ lead to a smoother result?

References
[1] Liu, X., Tong, X., & Liu, Q. (2021). Sampling with trusthworthy constraints: A variational gradient framework. Advances in neural information processing systems, 34, 23557-23568.
[2] Yao, Y., Pan, Y., Li, J., Tsang, I., & Yao, X. (2024). PROUD: PaRetO-gUided diffusion model for multi-objective generation. Machine Learning, 113(9), 6511-6538.
[3] Uehara, M., Zhao, Y., Hajiramezanali, E., Scalia, G., Eraslan, G., Lal, A., ... & Biancalani, T. (2024). Bridging model-based optimization and generative modeling via conservative fine-tuning of diffusion models. Advances in Neural Information Processing Systems, 37, 127511-127535.

---

> ### Author Rebuttal · Authors · 2026-03-30
>
> We appreciate the reviewer’s constructive feedback and are glad they recognize CGM as a reasonable solution to the calibration problem, as well as the breadth of the experimental validation. Our manuscript has been improved by incorporating their feedback.
>
> ## Weaknesses
> > The paper ignores the literature that also impose a constraint on the distribution of generative model.
>
> The goal of CGM is to impose distributional constraints by _fine-tuning_ a pre-trained generative model, as opposed to applying an _inference-time_ method when sampling from a pre-trained model. The references cited by the reviewer are inference-time methods.
>
> We define inference-time methods to be those that take a pre-trained generative model and modify the samples during generation time in order to satisfy the constraint (e.g., references [1,2]). On the other hand, fine-tuning methods modify the weights of the model and do not change the sampling procedure.
>
> Both fine-tuning and inference-time methods play important roles in generative modeling. For the goal of generating high-reward samples, for example, inference-time methods like best-of-N sampling are suitable when one has a large computational budget at sampling time. On the other hand, reward fine-tuning algorithms like direct preference optimization (DPO) [4] are preferable when one wishes to share the fine-tuning cost across examples (i.e., amortization) by training ahead of time.
>
> While there has been significant progress (e.g., [1, 2, 5]) in inference-time methods for distributional constraints, _we are not aware of previous fine-tuning methods that solve this problem for arbitrary likelihood-based generative models_. We thank the reviewer for raising this point; we will clarify the distinction between fine-tuning and inference-time methods in the revised manuscript.
>
> [3] does not consider distributional constraints, but instead seeks to optimize a reward.  Furthermore this reward is required to be differentiable, whereas in CGM the constraint function $h(x)$ can be nondifferentiable. This is the case in all of our applications.
>
> [4] Rafailov, R., et al. Direct Preference Optimization: Your Language Model is Secretly a Reward Model. NeurIPS, 2023.
>
> [5] Kolloff, C. et al. Minimum-Excess-Work Guidance. arXiv preprint arXiv:2505.13375, 2025.
>
> > The maximum entropy method appears to achieve the best performance. If this is the case, it raises the question of whether it would be a better solution for the target problem.
>
> Related to the previous response, the maximum entropy solution can be viewed as an inference time method. It involves first drawing samples from the pre-trained generative model and then reweighting them using (eq. 5). It does not provide a fine-tuned model.
>
> As for computation, the maximum entropy solution is a theoretical optimum characterized by a single vector $\alpha^\*$ having dimension equal to that of the calibration constraint. We approximate $\alpha^\*$ by first drawing $N$ samples from the pre-trained model and then solving a convex optimization problem (implemented using cvxpy). However, in some cases (see e.g., Figure 3B) the vector $\alpha^\*$  is not practically computable, even with up to $N = 10^5$ samples. It can even be undefined when the conditions of Theorem 2.1 are not met. In these cases, it is not an option as an inference-time method.
>
> We will provide additional discussion of the maximum entropy solution in our revised manuscript.
>
> > This raises the question of whether CGM-relax may generally be the better method and whether the CGM-reward version is necessary.
>
> As the reviewer pointed out, CGM-reward is generally less robust than CGM-relax, particularly in the setting of high-dimensional constraints. Nonetheless, we chose to present CGM-reward because of its connection to reward fine-tuning algorithms, which are widely used for fine-tuning language models, diffusion models, and flow models. The poor performance of CGM-reward for high-dimensional constraints, even when $\alpha^\*$ is fixed to its oracle value (Figure 3B), is suggestive that reward fine-tuning algorithms may face similar limitations.
>
> As a practical matter, we recommend one to first try CGM-reward because it does not require hyperparameter tuning, and to then try CGM-relax with a grid-search over lambda if CGM-reward does not work well.
>
> Note also that we have included results for both CGM-relax and CGM-reward in all experimental settings (see Appendix E, Figure 11 for TarFlow results). We have also ablated CGM-relax for numerous $\lambda$ values in all experiments.
>
> > Why is the obtained distribution of CGM-reward jagged, while the max entropy solution is smooth?
>
> In Figure 2A, we choose to visualize the solutions obtained by CGM-relax and CGM-reward by plotting histograms of samples. We will clarify that these are histograms in our revision.

---

> > ### Author Rebuttal · Reviewer_JPq9 · 2026-04-03
> >
> > Thanks for the detailed responses.
> >
> > It is now clearer that the key distinction between the maximum entropy approach and the proposed method lies in the difference between inference-time and fine-tuning-based strategy, which is however not well clarified in the original submission.
> >
> > I understand that [1,2] presents an inference-time method, which differs from the proposed fine-tuning method. But I argue that the framework of calibrating generative models with distribution constraint formulated in the paper cannot be a novel contribution, thus reducing the significance of this work.
> >
> > It seems CGM-relax can be understood as reward fine-tuning, which also maintains a pre-defined lambda, if the constraint gap is considered as a reward.
> >
> > The writing of the paper, including literature and contribution clarification, requires substantial improvement.
> >
> > An additional point: The CGM-reward method automatically derives the balance factor while considering the constraint. This aspect could represent a meaningful contribution if the authors can clearly demonstrate its practical advantages in real-world scenarios and or address its less robustness issues in high-dimensional constraints.

---

> > > ### Author Response · Authors · 2026-04-03
> > >
> > > We thank the reviewer for their response to our rebuttal. We have addressed their remaining concerns in our response.
> > >
> > > **Significance**: The reviewer argues that CGM lacks significance because there exist prior *inference-time* algorithms that aim to solve the distributional calibration problem. However, there are challenges to developing robust fine-tuning algorithms that do not arise for inference-time algorithms [1, 2]. For example, in CGM we propose unbiased, low variance gradient estimators for updating the model parameters. We demonstrate these estimators are effective even when calibrating generative models with high-dimensional latent variables (e.g., diffusion models). Inference-time algorithms do not update the model parameters.
> > >
> > > As promised in our rebuttal, we will provide additional discussion in the main text of inference-time versus fine-tuning methods for distributional calibration. Since the reviewer commented that the manuscript requires “substantial improvement” in terms of its presentation, we would appreciate specific feedback on additional areas for improvement.
> > >
> > > **CGM-relax as a reward fine-tuning algorithm**: The CGM-relax objective cannot be cast as a reward fine-tuning algorithm. Reward-fine-tuning objectives are of the form $D_{KL}(p_{\theta} || p_{\theta_{base}}) - E\_{p\_{\theta}}[r(x)]$, where $r$ is the reward to be maximized. Importantly, since the constraint function $h$ acts on the full distribution, as opposed to individual samples, the CGM-relax objective cannot be rewritten in this form. The CGM-reward objective, on the other hand, can be viewed as a reward fine-tuning algorithm with the reward equal to the weighted residual $r(x) = (\alpha^\*)^\top(h(x) - h^*)$.
> > >
> > > **Advantages of CGM-reward**: The reviewer is correct that the maximum entropy distribution is characterized by a vector $\alpha^\*$ that solves the calibration problem (eq. 4) and does not require hyperparameter search. However, $\alpha^*$ is not available to the practitioner and must be estimated. As we pointed out in our rebuttal, we do this by first drawing $N$ samples from the pre-trained model $p_{\theta_{base}}$ and then solving the convex program (eq. 7). In both our rebuttal and Section 3 of the manuscript, we have clearly described that for high-dimensional constraints, the estimate of $\alpha^\*$ is degenerate with high probability even when $N$ is as large as $N = 10^5$. We would be happy to further clarify why CGM-reward fails for high-dimensional constraints.

---

### Decision · Program_Chairs · 2026-04-30

**Decision:**

Accept (regular)

**Comment:**

This paper studies an important and fairly general problem in generative modeling, namely how to calibrate a generator to satisfy desired distribution-level constraints while remaining close to the original model. Reviewers appreciated both the clean optimization perspective and the two concrete methods, CGM-relax and CGM-reward, and they also viewed the evaluation breadth across multiple settings as a strength. The main remaining concerns were more about positioning, presentation, and practical guidance on when the two variants should be preferred than about any fatal methodological weakness. One reviewer remained at weak reject, but even there the emphasis after rebuttal was on framing and clarity rather than on a broken core idea. Given the final scores and the conceptual strength of the contribution, I lean toward acceptance.